# CD201⁺ fascia progenitors choreograph injury repair

Donovan Correa-Gallegos[1,6], Haifeng Ye[1,6], Bikram Dasgupta[1,6], Aydan Sardogan[1], Safwen Kadri[1], Ravinder Kandi[1], Ruoxuan Dai[1], Yue Lin[1], Robert Kopplin[1], Disha Shantaram Shenai[1], Juliane Wannemacher[1], Ryo Ichijo[1], Dongsheng Jiang[1], Maximilian Strunz[2], Meshal Ansari[2], Illias Angelidis[2], Herbert B. Schiller[2,3], Thomas Volz[4], Hans-Günther Machens[5] & Yuval Rinkevich[1✉]

Optimal tissue recovery and organismal survival are achieved by spatiotemporal tuning of tissue inflammation, contraction and scar formation[1]. Here we identify a multipotent fibroblast progenitor marked by CD201 expression in the fascia, the deepest connective tissue layer of the skin. Using skin injury models in mice, single-cell transcriptomics and genetic lineage tracing, ablation and gene deletion models, we demonstrate that CD201⁺ progenitors control the pace of wound healing by generating multiple specialized cell types, from proinflammatory fibroblasts to myofibroblasts, in a spatiotemporally tuned sequence. We identified retinoic acid and hypoxia signalling as the entry checkpoints into proinflammatory and myofibroblast states. Modulating CD201⁺ progenitor differentiation impaired the spatiotemporal appearances of fibroblasts and chronically delayed wound healing. The discovery of proinflammatory and myofibroblast progenitors and their differentiation pathways provide a new roadmap to understand and clinically treat impaired wound healing.

Wound healing is a conserved process that requires tight spatiotemporal tuning of consecutive phases[1]. In the inflammatory phase, local signals trigger the innate immune system around the wound[2,3]. Timely wound healing requires the transition of the inflammatory phase into a proliferation phase in which specialized fibroblasts called myofibroblasts[4] produce the wound extracellular matrix (ECM). The final remodelling phase of wound healing is characterized by the lasting action of myofibroblasts in the restored tissue, in which further ECM production forms scar tissue.

The coordinated and timely initiation, persistence and resolution of each of the phases is key to effective healing. Protracted inflammatory phase or delayed proliferative phase can lead to chronic wounds observed in conditions such as ageing, diabetes, surgical or infectious wounds, vascular and heart diseases, or even cancer[1,3,5]. By contrast, a protracted remodelling phase leads to excessive tissue contraction and fibrosis, as observed in Dupuytren's contractures, hypertrophic scars and keloid lesions[1]. In sum, impaired wound healing imposes an enormous burden. Therefore, better understanding the choreography of tissue recovery is a matter of tremendous clinical and research interest.

We previously showed that wound fibroblasts in the dorsal skin originate from a single embryonic cell lineage that expresses the engrailed 1 (*En1*) gene during development, termed *En1*-past fibroblasts[6,7] (EPFs), and more recently we showed that EPFs from the fascia possess a higher scar-forming potential[8].

Recent single-cell transcriptomics (scRNA-seq) studies have further unravelled fibroblast heterogeneity[9–17]. For example, proinflammatory fibroblasts and myofibroblasts are systematically detected across keloid lesions[18], aged skin[19], Dupuytren's nodules[20] and atopic dermatitis[15]. Moreover, distinct fibroblast subtypes respond differently to the same signals[21–23], depending on their spatial locations. Nevertheless, we lack a precise map of the wound pinpointing where fibroblast progenitors reside and where differentiation signals are most effective.

Here we perform longitudinal stromal-enriched scRNA-seq combined with in vivo genetic lineage tracing, cell ablation and gene deletion approaches to generate a high-resolution pedigree study of all fibroblasts during skin wound healing. We identify a fibroblast progenitor in the subcutaneous fascia that controls the spatiotemporal features of wound healing by undergoing sequential differentiation into specialized fibroblastic cell types.

## Fibroblast heterogeneity in wounds

To reveal the biological transition of fibroblasts during wound healing, we combined scRNA-seq with genetic lineage tracing of EPFs by crossing mice expressing Cre recombinase under the *En1* promoter (*En1^cre*) with the fluorescent Rosa26–mTmG (*R26^mTmG*) reporter mouse line (Extended Data Fig. 1a). The resulting *En1^cre R26^mTmG* line tags EPFs permanently with the GFP. We then performed bilateral, full-thickness excisional wounds on the back skin and, at relevant days post-injury (dpi), we sequenced the mesenchymal-enriched fraction from the inflammatory (1 and 3 dpi), proliferation (5 and 7 dpi) and remodelling phases of wound healing (14 and 27 dpi) (Extended Data Fig. 1b). The

[1]Institute of Regenerative Biology and Medicine (IRBM), Helmholtz Munich, Munich, Germany. [2]Member of the German Centre for Lung Research (DZL), Comprehensive Pneumology Center (CPC) and Institute of Lung Health and Immunity (LHI), Helmholtz Munich, Munich, Germany. [3]Institute of Experimental Pneumology, Ludwig-Maximilians University Hospital, Munich, Germany. [4]Klinikum rechts der Isar, Department of Dermatology, School of Medicine, Technical University of Munich, Munich, Germany. [5]Klinikum rechts der Isar, Department of Plastic and Hand Surgery, School of Medicine, Technical University of Munich, Munich, Germany. [6]These authors contributed equally: Donovan Correa-Gallegos, Haifeng Ye, Bikram Dasgupta. ✉e-mail: yuval.rinkevich@helmholtz-muenchen.de

initial sequencing approach was unbiased towards EPFs, and therefore included both GFP[+] and GFP[−] fibroblasts.

We detected seven distinct fibroblast clusters (Extended Data Fig. 1c). Clusters 0 to 2 were present in both uninjured and injured skin at all timepoints, whereas clusters 3 to 6 were only found in injured tissue (Extended Data Fig. 1d). To annotate previously known populations, we scored the expression of a curated set of published markers for resident papillary, reticular and fascia fibroblasts, and for classical myofibroblasts. Cluster 0 scored highest for the papillary profile, marked by *Sparc*[17], *Dcn* and *Lum*. Cluster 2 scored highest for the reticular profile, marked by *Cxcl12*[17], *Cygb*[17], and *Mgp*. Cluster 1 scored the highest for the fascia profile, marked by *Sca1*[8,24] (also known as *Ly6a*) and *Plac8*[17], as well as *Pi16* and *Dpt*, both previously reported as universal fibroblast markers[25]. We also identified *Cd201* (also known as *Procr*), a marker of progenitor mesenchymal cells in spleen[26] and muscle[27], as a specific fibroblast marker of the fascia (Extended Data Fig. 1e,f).

Next, we examined the wound-exclusive clusters (clusters 3 to 6). Cluster 6 was prominent during the inflammatory phase (1–3 dpi) (Extended Data Fig. 1c,d) and was marked by *Pdpn*, *Ccl2* and *Cxcl1* expression (Extended Data Fig. 1f). Gene ontology (GO) overrepresentation analysis revealed an immunomodulatory specialization of cluster 6 (hereafter called 'proinflammatory fibroblasts', Extended Data Fig. 1g). Clusters 3 and 5 peaked during the proliferation phase (5–7 dpi) (Extended Data Fig. 1d), with a high myofibroblast profile score, differing only by the expression of α-smooth muscle actin (αSMA, encoded by the *Acta2* gene) (Extended Data Fig. 1e,f). The *Acta2*[high] cluster 5 (hereafter referred to as 'myofibroblasts') also expressed the myofibroblast markers *Postn* and *Lrrc15*[28], as well as *Runx2*. The *Acta2*[low] cluster 3, representing the immature cells (hereafter called 'proto-myofibroblast'[4]), expressed *Tnc*, *Stat3* and *Pcsk5* (Extended Data Fig. 1f). The proto-myofibroblast expression profile included hypoxia-related GO terms, whereas myofibroblasts expressed terminal differentiation terms (Extended Data Fig. 1g). Cluster 4 (hereafter referred to as as SFRP2[+] fibroblasts) was marked by *Sfrp2* (which is also detected in human[29]), Cthrc1 and Fstl1 (Extended Data Fig. 1f). GO terms analysis indicated that all clusters except proinflammatory fibroblasts specialized in matrix production and tissue contraction (Extended Data Fig. 1g).

Next, we sought to trace the most likely origin of myofibroblasts by using partition-based graph abstraction (PAGA) for trajectory inference, which calculates the connections (for example, differentiation steps) between clusters. This initial unbiased analysis revealed a complex interconnection between all fibroblasts and corroborated previous observations that papillary fibroblasts do not contribute to the wound fibroblast pool[24] (Fig. 1a), and that the most likely fate of reticular fibroblasts is the SFRP2[+] fibroblast (Fig. 1b). Furthermore, we identified the main myofibroblast source to be the proto-myofibroblast, which in turn branched off from the proinflammatory fibroblast. Notably, the proinflammatory fibroblast cluster derived exclusively from the fascia cluster, suggesting that proinflammatory, proto-myofibroblasts and myofibroblasts follow a sequential lineage trajectory that emerges from the fascia fibroblasts (Fig. 1b). As EPFs are the predominant lineage in wounds[6], we aimed to place this lineage in the context of this new differentiation trajectory. We sorted the GFP[+] EPFs in our scRNA-seq dataset and reanalysed them with a higher clustering resolution (Extended Data Fig. 2a). EPFs were present across all homeostatic populations as well as in the proinflammatory, proto-myofibroblast and myofibroblast clusters (Extended Data Fig. 2b–d).

Our analysis also revealed an additional intermediate fascia EPF cluster that is present in injured samples but is absent from uninjured skin (Extended Data Fig. 2c,d). These injured fascia EPFs express the naive fascia markers *Pi16* and *Plac8*, while simultaneously upregulating the proinflammatory fibroblast markers *Pdpn*, *Ccl2* and *Cxcl1* (Extended Data Fig. 2e). This reinforces the notion that the three injury-related cell clusters (proinflammatory, proto-myofibroblast and myofibroblast) are most probably derived from fascia-resident EPFs.

Based on our initial trajectory analysis and previous evidence of the contribution of fascia fibroblast into the myofibroblast pool[8], we recalculated the sequential trajectory sprouting from the fascia cluster, passing through the proinflammatory and proto-myofibroblast states and ending in the myofibroblast state (Fig. 1c) for further study.

## Genetic programmes along the trajectory

We next explored the genetic programmes of each fibroblast cluster from our trajectory. To do so, we mapped the transcription factor 'programme', their respective target genes ('regulon') and GO terms analysis of the same revealed discrete functions for each fibroblast cluster (Extended Data Fig. 3a).

The source of the trajectory, fascia fibroblasts, maintained a *Fos*, *Jun* and *Egr1* programme, which controlled a 263-gene regulon that promotes cell survival and growth (Extended Data Fig. 3b), fitting its role as a surveillance system for injury repair.

The proinflammatory fibroblasts expressed a 116-gene regulon controlled by the *Nfe2l2* and *Bach1* programme, which directs oxygen sensing (Extended Data Fig. 3c), indicating that, besides immunomodulation, oxygen sensing is an important function acquired by fascia fibroblasts when becoming proinflammatory fibroblasts.

For the second step in the trajectory, the proto-myofibroblast programme comprised *Stat3*, hypoxia-inducible factor-1-alpha (*Hif1a*), *En1*, *Ets2*, *Cebpb*, *Rora* and *Fosl1*, which controls a 251-gene regulon involved in migration, oxygen sensing, adhesion and collagen production (Extended Data Fig. 3d). These programme changes reveal a progressive phenotype switch from immunomodulatory activities from the proinflammatory fibroblasts towards matrix-oriented functions, mimicking the transition from the inflammatory to proliferation phases of wound healing.

Meanwhile in the last step of the trajectory, the myofibroblast maintained a *Tcf4*, *Runx1* and *Runx2* programme that controlled a 21 genes regulon that directs terminal differentiation processes (Extended Data Fig. 3e). Together, our inferred trajectory and discrete phenotypes indicate that fascia fibroblast differentiation into proinflammatory, proto-myofibroblasts and myofibroblasts mirror the transition between the different wound healing phases (Extended Data Fig. 3f).

## Spatiotemporal differentiation of CD201[+] cells

We then sought to frame the cell differentiation steps of our trajectory within the wound healing phases in vivo. First, we confirmed the temporal expression patterns of our cluster-specific markers (Extended Data Fig. 4a) in mouse skin wounds at 3 and 7 dpi: two critical timepoints marking the transition from the inflammatory to proliferation phases and from the proliferation to remodelling phases of wound healing, respectively. PDPN[+] proinflammatory and phosphorylated activated STAT3[+] (pSTAT3[+]) proto-myofibroblasts peaked during the first transition at 3 dpi and declined at later timepoints (Extended Data Fig. 4b,c), whereas RUNX2[+] myofibroblasts were prominent at the second transition into the remodelling phases of wound healing at 7 dpi (Extended Data Fig. 4d).

We next analysed the spatial distribution of proinflammatory, proto-myofibroblast and myofibroblast clusters in three distinct wound compartments (Extended Data Fig. 4e). At 3 dpi, PDPN[+] proinflammatory fibroblasts were confined to the wound bed and upper wounds (Extended Data Fig. 4f), whereas pSTAT3[+] proto-myofibroblasts showed no preference among these areas (Extended Data Fig. 4g). By contrast, at 7 dpi, RUNX2[+] myofibroblasts clearly localized in upper wound compartments, with minimal presence in deep wound areas (Extended Data Fig. 4h). This gradual spatial bottom-up differentiation of fibroblasts

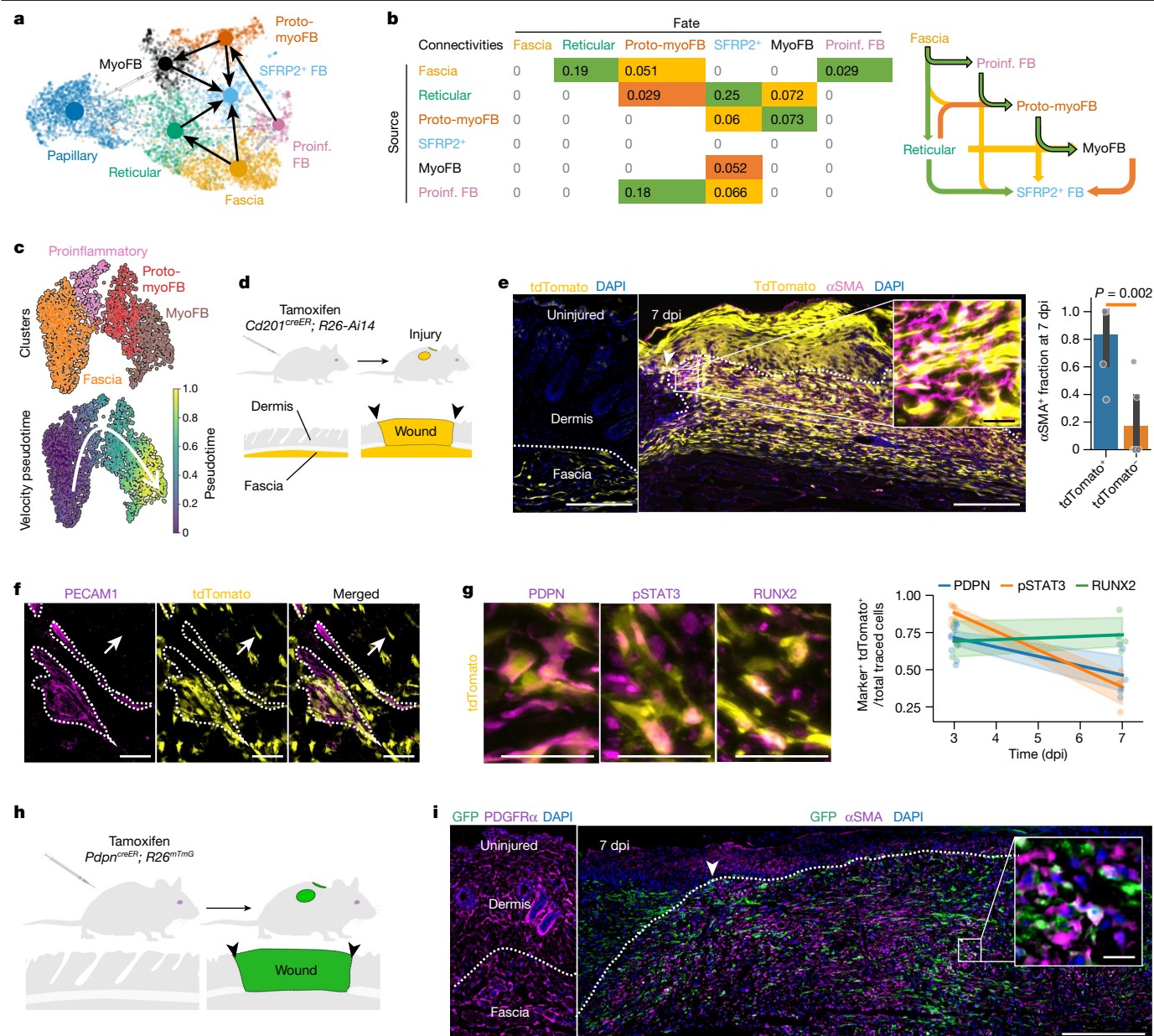

**Fig. 1 | CD201+ fascia fibroblasts differentiate into specialized states during wound healing. a**, Uniform manifold approximation and projection (UMAP) analysis of all fibroblast clusters embedded with PAGA connectivities for trajectory inference. FB, fibroblast; MyoFB, myofibroblast; proinf., proinflammatory. **b**, Left, PAGA connectivity values with potential sources (rows) and fates (column) colour-coded as most (green), intermediate (yellow) and least probable (orange). Right, schematic of the potential trajectories highlighting the fascia-to-myofibroblast trajectory. **c**, UMAP analyses of fibroblasts from the fascia-to-myofibroblast trajectory, colour-coded for individual clusters (top) and velocity pseudotime score (bottom). **d**, Genetic lineage system to trace the fate of CD201+ fascia fibroblasts during wound healing. **e**, Representative histology of uninjured (left) *Cd201^creER^ R26^Ai14^* and 7 dpi wound (middle) immunolabelled for αSMA. Right, the contribution of CD201+ fibroblasts to the myofibroblast pool was determined for 6 wounds

from 3 mice. Two-tailed Student's *t*-tests. The dotted line delimits the fascia or wound region. Arrowheads indicate the original injury site. The inset shows an expanded view of the outlined region. **f**, Representative whole-mount immunostaining for the endothelial marker PECAM1 in uninjured *Cd201^creER^ R26^Ai1^* fascia. Arrows indicate labelled cells dispersed away from the adventitial space. **g**, Representative high-magnification images showing the expression of PDPN, pSTAT3 or RUNX2 in traced cells (left) and quantification at 3 and 7 dpi (right). Values obtained from at least three biological replicates. **h**, Genetic lineage system to trace the fate of proinflammatory fibroblast. **i**, Representative histology of uninjured skin immunolabelled for PDGFRα (left) and a 7-dpi wound (right) of *Pdpn^creER^ R26^mTmG^* immunolabeled for αSMA. The dotted line delimits the fascia or wound region. Arrowheads indicate the original injury site. The inset shows an expanded view of the outlined region. Scale bars: 500 μm (**f** and **e,i**, main images) and 50 μm (**g** and **e,i**, expanded views).

towards upper wound compartments indicates a spatial distribution of guiding signals promoting the sequential differentiation of fascia fibroblasts (Extended Data Fig. 4i).

To validate that fascia progenitors differentiate into all three distinct fibroblast states in vivo, we performed genetic lineage tracing using an inducible CreER recombinase mouse line driven by the expression of

the fascia marker CD201 (*Cd201^creER^*). In our scRNA-seq dataset, *Cd201* expression enriched the fascia fraction 1.7-fold versus the dermal fraction in relation to the pan-fibroblast marker *Pdgfra* (60% fascia and 39% dermal), whereas *Sca1* showed a lower enrichment of 1.24-fold (44.6% fascia and 55.4% dermal) (Extended Data Fig. 5a), indicating that *Cd201* is a better candidate to specifically target fascia fibroblasts.

We then crossed the *Cd201^creER* line with the fluorescent reporter line *R26^Ai14*, in which tamoxifen administration induces the enduring expression of tdTomato in cells that express CD201 at the time of exposure in the *Cd201^creER R26^Ai14* offspring (Fig. 1d).

In uninjured skin, our system almost exclusively labelled fibroblasts from the fascia compartment and not endothelial, epithelial or immune cells (Fig. 1e and Extended Data Fig. 5b–d). Similarly, tdTomato+ cells in 7 dpi wounds were predominantly fibroblasts and not endothelial, epithelial, or immune cells (Extended Data Fig. 5b,c). Furthermore, tdTomato+ cells contributed to 83% of all αSMA+ myofibroblasts (Fig. 1e). Although CD201 has been reported as a marker of haematopoietic stem cells[30], we observed no tdTomato expression on immune cells in the bone marrow or skin wounds (Extended Data Fig. 5b,c), discarding their potential contribution to the tdTomato+ pool in skin wounds. We also observed that tdTomato+ fibroblasts in the uninjured fascia spread along the connective tissue and were not restricted to the perivascular space (Fig. 1f).

Next, we tested whether CD201+ fascia fibroblasts differentiate into proinflammatory, proto-myofibroblasts and myofibroblasts upon injury. During the progression from inflammatory–proliferation to proliferation–remodelling transitions, CD201+ progenitors consistently downregulated proinflammatory and proto-myofibroblast markers: 71.5 down to 46.4% for PDPN and 88.2 down to 38.4% for pSTAT3 from 3 to 7 dpi. By contrast, the expression of the mature myofibroblast marker RUNX2 increased up to 73.5% in the CD201 fibroblast lineage at 7 dpi (Fig. 1g and Extended Data Fig. 6a). The temporal shift in marker gene expression of CD201+ fascia fibroblasts indicates that their transition into proinflammatory and to myofibroblasts occurs in synchrony to the progression of the wound healing phases.

To validate that the CD201+ fibroblast differentiation into myofibroblasts is preceded by the intermediate proinflammatory state, we generated a complementary system in which the expression of CreER recombinase is controlled by the proinflammatory marker *Pdpn* (*Pdpn^creER*). These mice were crossed with the *R26^mTmG* reporter mouse line, enabling lineage tracing from the proinflammatory state onwards (Fig. 1h). In contrast to our *Cd201^creER* fascia tracing system, *Pdpn^creER R26^mTmG* mice show no GFP expression in any skin fibroblast under homeostatic conditions (Fig. 1i). At 7 dpi, αSMA+ GFP+ (proinflammatory-derived) myofibroblasts were detected abundantly in upper wounds (Fig. 1i). Furthermore, co-expression of PDPN, pSTAT3 and RUNX2 markers in GFP+ cells at relevant times after injury further indicate that proinflammatory fibroblasts follow the same sequential trajectory into mature myofibroblasts (Extended Data Fig. 6b). Given that PDPN marks lymphatic vessels[31], we determined the relative contribution of LYVE1+ lymphatic cells to the GFP+ pool. We observed that only one-quarter of all GFP+ cells were lymphatic cells and only one-third of all lymphatic cells were tagged with GFP+ (Extended Data Fig. 6c), making most of the labelled cells descendants of the proinflammatory fibroblasts. These results confirm that proinflammatory fibroblasts transition into myofibroblasts during skin wound healing and that our *Cd201^creER* and *Pdpn^creER* transgenic systems faithfully trace their conversion steps.

In sum, in silico trajectory inference, spatiotemporal expression patterns and complementary genetic lineage tracing methods, detail a spatiotemporal coordination of fibroblast differentiation taking place during wound healing, which arises from CD201+ progenitors differentiating into proinflammatory fibroblasts within wound beds that finally mature into myofibroblasts in the upper wound region.

## Targeting the trajectory

We next tested the relevance of the multi-step differentiation of fascia into myofibroblast for wound healing. As myofibroblasts are contractile cells, we first correlated tissue changes to fascia fibroblast differentiation in a mouse back-skin fascia ex vivo model (Fig. 2a). During culture and similar to mammalian wounds, the translucent fascia explants gradually contracted down to 50% of their original area until they formed an opaque sphere of scar-like tissue (Fig. 2b).

We also confirmed that ex vivo fascia fibroblasts transitioned into proinflammatory fibroblasts within 1–3 days post culture (4.6-fold increase in PDPN+ cells at day 3 over day 1; Fig. 2c). By contrast, differentiation of proto-myofibroblasts and myofibroblasts coincided with tissue contraction after day 3 (9.5-, 4.2- and 3.2-fold increases in pSTAT3+, RUNX2+ and αSMA+ cells at day 6, respectively; Fig. 2c). Cell ablation hindered explant contraction (Fig. 2d), demonstrating that differentiation of fascia fibroblasts differentiation into contractile states mediates tissue contraction. By contrast, heat-inactivated or low-serum medium had minimal effects on tissue contraction (Fig. 2d), indicating that fascia fibroblast differentiation is intrinsically regulated.

We then performed genetic cell ablation of PDPN+ proinflammatory fibroblasts in vivo by crossing our *Pdpn^creER* mice with the *R26^DTA* line, in which Cre-mediated recombination enables expression of the diphtheria toxin protein, leading to cell death. Thus, upon exposure to tamoxifen in *Pdpn^creER R26^DTA* mice, PDPN+ proinflammatory fibroblasts die before transiting into contractile proto- and myofibroblasts. We used the splinted wound model, which better mimics the human healing process by mitigating the action of other contractile tissues such as muscle. Splinted wounds in control mice (*Pdpn^creER R26^WT* and *Pdpn^creER R26^mTmG*) showed a clear wound closure between 7 and 14 dpi, whereas the ablation of proinflammatory fibroblasts caused a significant delay in wound closure and contraction in *Pdpn^creER R26^DTA* mice, whose wounds did not fully close by 14 dpi (Fig. 2e). In the experimental group, 7 dpi wounds showed a minimal wound bed with poor matrix content compared with control mice (Fig. 2f), indicative of a null wound healing progression. Notably, ablation of proinflammatory fibroblasts caused a significant decrease in the number of contractile αSMA+ myofibroblasts in 7 dpi wounds (Fig. 2f). Together, our results indicate that fascia-derived proinflammatory fibroblasts have a pivotal role in the proper progression of the wound healing phases.

## Retinoic acid gates the first transition step

The dynamic action of biochemical and biomechanical signals ensues the progression of the wound healing phases. To explore the connections between CD201+ progenitor differentiation and relevant signals, we scored the expression of genes from seven signalling pathways in our scRNA-seq dataset throughout the fascia-to-myofibroblast trajectory. Classical pathways associated with myofibroblast differentiation[4]—and thus, the proliferation and remodelling phases—such as TGFβ, Wnt and ECM mechanotransduction peaked at the myofibroblast state. Inflammatory-related signals, such as chemokines peaked at the proinflammatory fibroblast state (Fig. 3a).

Notably, the expression of retinoic acid (RA) pathway genes peaked in the transition from CD201+ progenitors to proinflammatory fibroblasts (Fig. 3a). Genes encoding RA-synthesizing enzymes, *Aldh1a3* and *Rdh10*, were overexpressed together with the chemokine genes *Ccl11*, *Ccl2*, *Ccl7*, *Ccl8* and *Cxcl1* (Fig. 3b). Indeed, ALDH1A3, CCL2 and CXCL1 proteins were expressed in CD201+-derived proinflammatory fibroblasts (tdTomato+PDPN+) in 3 dpi wounds (Fig. 3c–e). The RA-degrading enzyme *Cyp26b1* was overexpressed in proto-myofibroblasts (Fig. 3b) and its protein product was localized in the upper wound margins, where differentiation into myofibroblasts occurs (Fig. 3f).

The spatial link between RA synthesis and degradation, and fibroblast differentiation suggests that a RA gradient, originating within the wound bed and decreasing in the upper wound regions, enables the differentiation of progenitor fibroblasts to myofibroblasts. To test this, we either added exogenous RA or increased the endogenous RA levels using a CYP26B1 inhibitor in fascia explants. Both treatments prevented tissue contraction (Fig. 3g), indicating that RA is actively metabolized in culture and that high RA concentrations prevent fascia

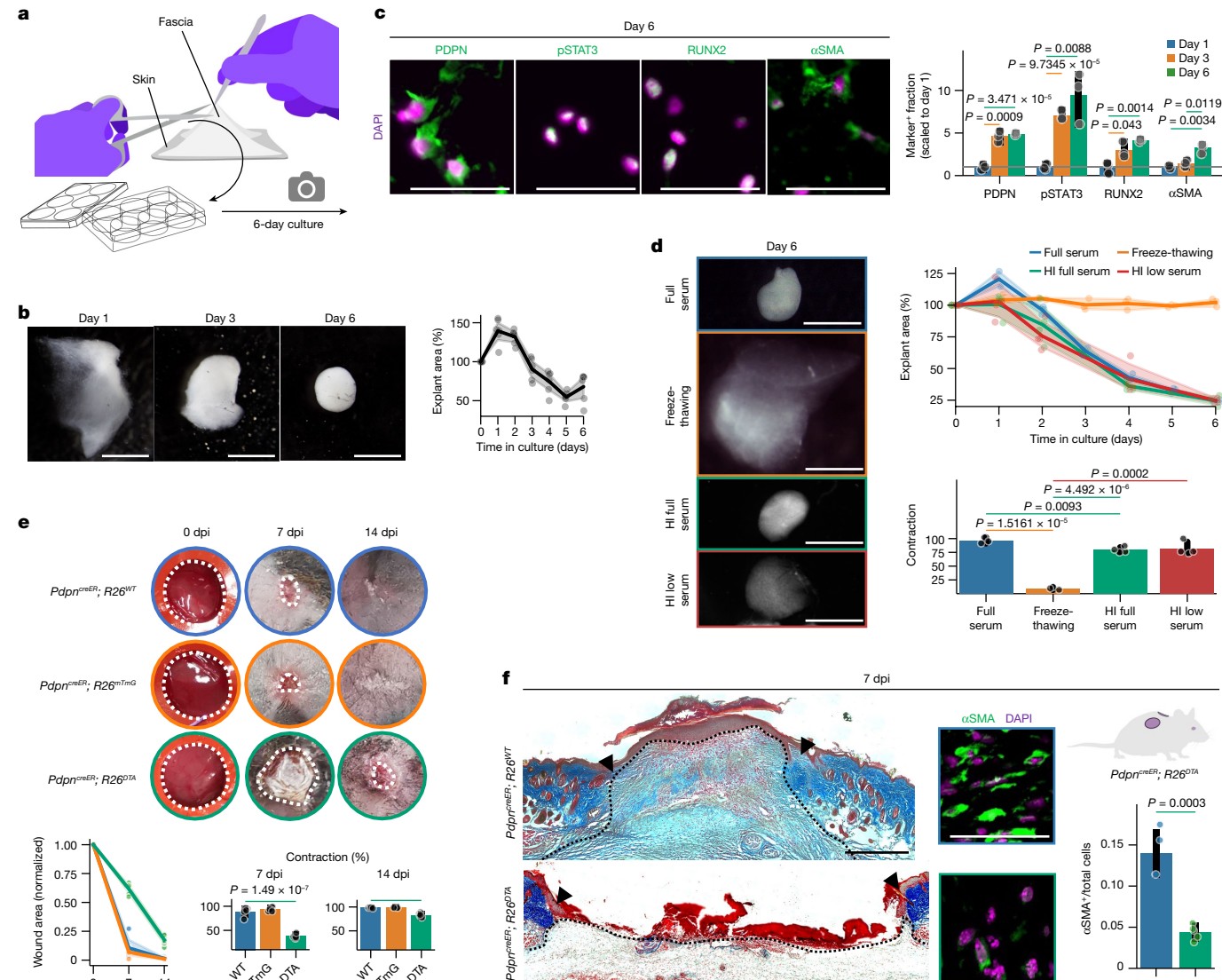

**Fig. 2 | Targeting fascia differentiation impairs tissue contraction and wound closure. a**, The fascia explant culture method. **b**, Representative bright-field images of fascia explants at indicated timepoints of culture (left) and area versus time contraction measurements (right). *n* = 5 from 3 biological replicates. **c**, Representative micrographs of fascia explants immunolabelled for PDPN, pSTAT3, RUNX2 or αSMA (left) and quantification of expressing cell fraction at 1, 3 and 6 days of culture normalized to day 1 values (right). *n* = 3 biological replicates. **d**, Representative images of fascia explants at 6 days of culture (left), contraction dynamics (top right) and total contraction values (bottom right) showing the effects of cell ablation (freeze–thawing) and different serum conditions. *n* = 3 (full serum and freeze–thawing) and 4 (heat-inactivated high and low serum) biological replicates. HI, heat-inactivated. Colours of the image outlines indicate treatments in the graphs. **e**, Representative photographs of splinted wounds on PDPN-related transgenic lines at indicated timepoints after injury (top), wound area quantification (bottom left) and contraction percentage at the indicated dpi (bottom right), showing that ablation of proinflammatory fibroblasts delays wound closure. *n* = 6 wounds from 3 biological replicates. WT, wild type. **f**, Masson's trichrome staining of control (top left) and ablated (bottom left) 7 dpi wounds. The dotted line delimits the wound region. Arrowheads indicate the original injury site. High-magnification images of control (top middle) and ablated (bottom middle) 7 dpi wounds immunolabelled for αSMA. Right, myofibroblast ratio in wounds. *n* = 4 (control) and 5 wounds from 3 biological replicates. Colours of the image outlines indicate treatments in the graph. Two-tailed *t*-tests. Scale bars: 2 mm (**b**,**d**), 500 μm (**f**, left) and 50 μm (**c** and **f**, middle).

tissue contraction. Similarly, agonists for each of the three RA receptors phenocopied the contraction obstruction, and a pan-RA receptor antagonist significantly increased the overall tissue contraction (Extended Data Fig. 7a), showing that the RA signalling acts through its canonical transcriptional activity.

In fascia explants, treatment with an agonist of RARγ, the most highly expressed receptor in fibroblasts (Fig. 3b), increased the number of PDPN⁺ proinflammatory fibroblasts and CCL2 expression, and significantly decreased the number of RUNX2⁺ and αSMA⁺ myofibroblasts after 6 days of culture (Fig. 3h), indicating that RA overactivation promotes the proinflammatory state and prevents their conversion to myofibroblasts, leading to impaired tissue contraction. Notably, the pan-RAR antagonist did not increase myofibroblast numbers (Fig. 3h), indicating that other signals cause the transition from proinflammatory fibroblast to myofibroblast.

To explore the potential role of RA overactivation in sustaining the proinflammatory state, and thus the inflammatory phase, we treated skin wounds with RARγ agonist. Although subcutaneous exposure to the agonist had no significant effect on the wound closure rate (Extended Data Fig. 7b), treated wounds had a stronger and protracted inflammatory phase, revealed by a significant increase of monocyte and macrophage numbers in wounds at 3 and 7 dpi (Fig. 3i and Extended

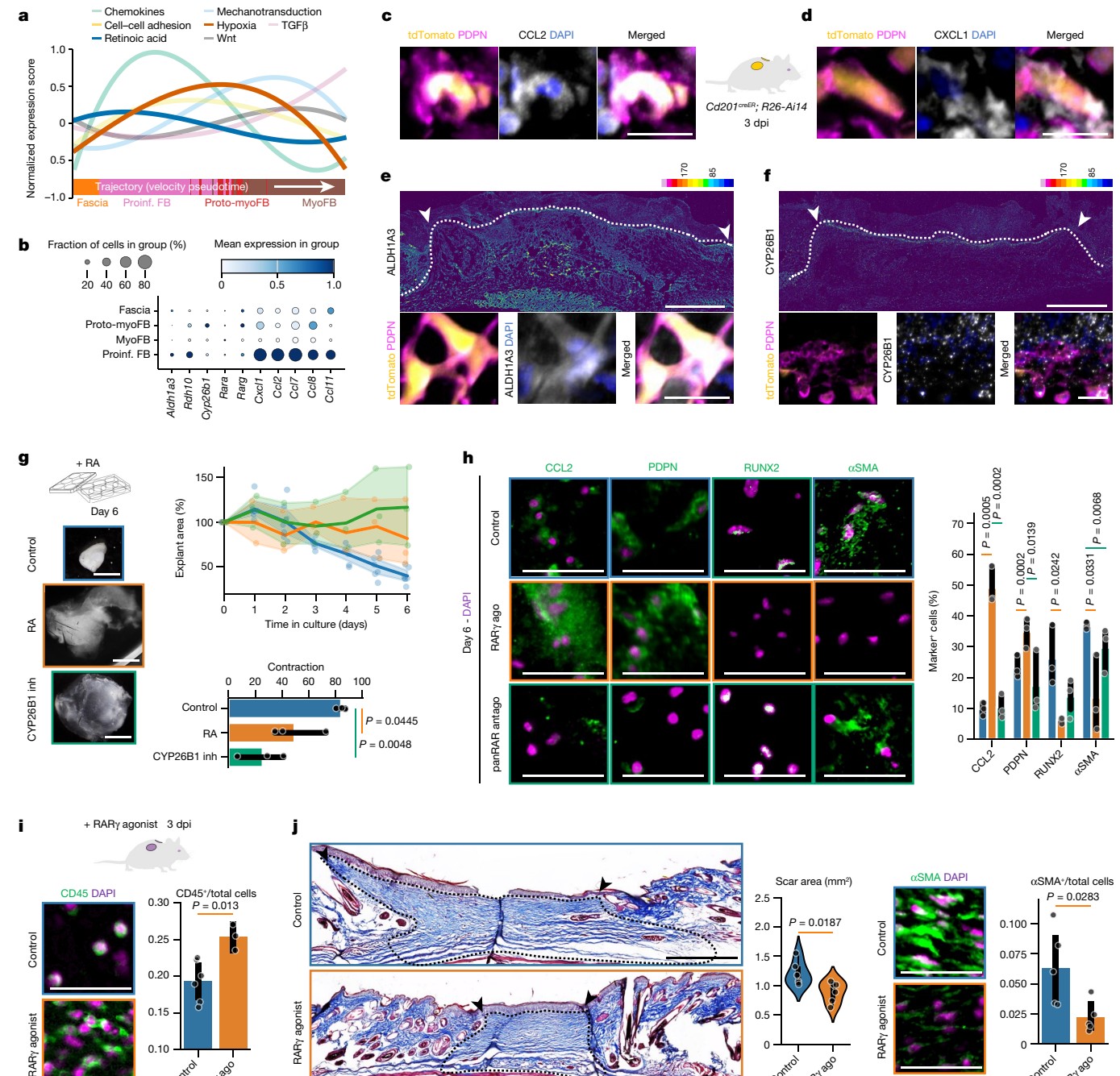

**Fig. 3 | A RA gate supports the proinflammatory state and limits myofibroblast differentiation. a**, Expression scores of signalling pathway genes along the fascia-to-myofibroblast trajectory. **b**, Expression of chemokine- and RA-related genes. **c**,**d**, Representative micrographs of fascia-traced fibroblasts using the *Cd201^creER^R26^Ai14^* system, showing the co-expression with PDPN and CCL2 (**c**) or CXCL1 (**d**). **e**,**f**, Top, representative immunolabelling for ALDH1A3 (**e**) and CYP26B1 (**f**) using '16_colours' pseudo-colouring representing expression intensity. Bottom, magnification of fascia-traced cells showing co-expression with PDPN and ALDH1A3 (**e**) or CYP26B1 (**f**). **g**, Bright-field images of control, RA-treated or CYP26B1 inhibitor (inh)-treated fascia explants (left), explant area over time (top right), and total contraction (bottom right). *n* = 3 biological replicates. Colours of the image outlines indicate treatments in the graphs. **h**, Representative micrographs from fascia explants immunolabelled for CCL2, PDPN, RUNX2 or αSMA (left) and the percentage of cells expressing the markers (right) in control, RARγ agonist (ago) or pan-RAR antagonist (antago) treatments. Colours of the image outlines indicate treatments in the graph. *n* = 3 biological replicates. **i**, High-magnification images of control (top left) and treated (bottom left) 3 dpi wounds immunolabelled for the immune cell marker CD45, and CD45^+^ cell ratio (right). *n* = 6 control and 3 treated wounds from 3 biological replicates. Colours of the image outlines indicate treatments in the graph. **j**, Low-magnification images of Masson's trichrome-stained control (top left) or treated (bottom left) 14 dpi wounds, and quantification of scar area (middle). The dotted line delimits the analysed wound area. Arrowheads indicate the original injury borders. *n* = 6 control and treated wounds from 3 biological replicates. High-magnification images of control (top) and treated (bottom) wounds immunolabelled for αSMA and positive cell ratio in scars (right). N = 5 wounds from 3 biological replicates. Colours of the image outlines indicate treatments in the graph. Two-tailed *t*-tests. Scale bars: 2 mm (**g**), 500 μm (**e**,**f**, top and **j**, left) and 50 μm (**c**,**d**,**h**,**i**, and **e**,**f**, bottom and **j**, right).

Data Fig. 7c,d). Conversely, treated wounds had significantly fewer αSMA[+] myofibroblasts associated with smaller scars at 14 dpi (Fig. 3j). These observations confirm that in vivo RA activity has a role in the proinflammatory fibroblast state and its overactivation limits the subsequent transition to myofibroblasts, reducing scar formation as result.

To expand on the effects of overactivation of RA signalling observed in vivo, namely monocyte recruitment and myofibroblast differentiation prevention, we cultured CD201[+] fascia fibroblasts in inflammatory or proliferation phase-simulating media in combination with exogenous RA (Extended Data Fig. 7e). Inflammatory-simulating medium increased the expression of *Ccl2* and *Cxcl1* chemokine genes, whereas the proliferation-simulating medium only increased the expression of *Acta2* (Extended Data Fig. 7e). Addition of RA to the inflammatory medium further increased *Pdpn* and *Ccl2* expression and decreased *Acta2* and *Cxcl1* expression (Extended Data Fig. 7f). Addition of RA to proliferation medium caused a concentration-dependent decrease in *Acta2* expression (Extended Data Fig. 7g), in line with the effects limiting myofibroblast differentiation observed in vivo and ex vivo.

Together, our results indicate that RA has a supporting role during the inflammatory phase of wound healing by promoting a monocyte and macrophage recruiting phenotype in the fascia-derived proinflammatory fibroblasts via expression of the monocyte-chemoattractant CCL2. Even though overactivation of the pathway effectively limits the amount of myofibroblasts, downregulation of the RA signal is insufficient to trigger the transition to the contractile states, and thus other signals are required.

## Hypoxia gates the second transition step

Previous analyses highlighted the relevance of hypoxia during the transition from proinflammatory fibroblast to proto-myofibroblast, preceding classical myofibroblast-inductive signals such as TGFβ and ECM mechanotransduction mediated by YAP–TAZ[4] (Fig. 3a and Extended Data Fig. 1g). Notably, HIF1α—the master regulator of hypoxia—is part of the transcriptional programme of proto-myofibroblasts (Extended Data Fig. 3d). Furthermore, *Hif1a* transcriptional activity directly correlates with its expression in proto- and myofibroblasts (Fig. 4a) and HIF1α[+] cells were indeed detected within CD201[+]- and PDPN[+]-derived lineages as well as in fascia explants at times when proinflammatory fibroblasts transition to proto-myofibroblasts (Fig. 4b).

Inhibition of HIF1α activity in explants impaired tissue contraction (Extended Data Fig. 8a), increased numbers of PDPN[+] proinflammatory fibroblasts and decreased numbers of αSMA[+] myofibroblasts (Extended Data Fig. 8b). Notably, HIF1α inhibition prevented tissue contraction in the presence of either TGFβ or YAP–TAZ activators; comparatively, exogenous RA was bypassed by the action of the stronger TGFβ signal (Extended Data Fig. 8c,d). Furthermore, HIF1α inhibition decreased the activation (via phosphorylation) of SMAD2, indicating a minimal TGFβ signal activity upon treatment (Extended Data Fig. 8e). To expand on the molecular connections between hypoxia and TGFβ pathways, we examined the correlation between gene expression of the HIF1α regulon and the inferred hypoxia signal activity in our scRNA-seq dataset (Extended Data Fig. 8f). Directly regulated genes included external modulators of TGFβ (*Lgals1*[32] and *Serpine1*[33]), as well as internal modulators such as *Eno1*[34], *Ldha*[35], *Pkm*[36], *Ddit4*[37] and *En1*[38], which were all enriched in the proto- and myofibroblast clusters (Extended Data Fig. 8g,h). These data indicate that HIF1α acts as an upstream regulator of both YAP–TAZ mechanotransduction and TGFβ pathways.

In vivo chemical inhibition of HIF1α significantly delayed wound closure (Extended Data Fig. 8i) and these wounds had a marginal wound bed, similar to inflammatory phase wounds, whereas control wounds were completely closed and exhibited rich wound beds (Extended Data Fig. 8j). Notably, 3 dpi treated wounds showed similar leukocyte infiltration to control wounds (Extended Data Fig. 8k), confirming that HIF1α has a role only during the later contraction-related phases of wound

healing. Furthermore, HIF1α inhibition in *Pdpn*[creER]*R26*[mTmG] wounds significantly decreased proinflammatory-derived proto-myofibroblast (GFP[+]pSTAT3[+]) and myofibroblast (GFP[+]RUNX2[+]) numbers at 7 dpi (Fig. 4c), confirming that HIF1α activity is necessary for the transition from proinflammatory fibroblast to proto- and myofibroblast states, which in turn induces wound closure, tissue contraction and scar formation.

To unambiguously demonstrate the role of HIF1α during fascia-to-myofibroblast differentiation and its effect in the progression to contraction phases of wound healing, we made mice deficient in HIF1α in fascia or proinflammatory fibroblasts by crossing *Cd201*[creER] and *Pdpn*[creER] mice to a line harbouring *loxP* sites flanking exon 2 of the HIF1α gene (*Hif1a*[flox]), which results in a null allele in response to tamoxifen-induced recombination[39]. Wounds on mice with single allele deletion in CD201[+] fascia progenitors (*Cd201*[creER]*Hif1a*[fl/+]) showed equivalent wound closure dynamics to wild-type mice, in which 91.7% closure of the initial wound size takes place around 9 dpi (Fig. 4d). However, bi-allele deletion of *Hif1a* in CD201[+] fascia fibroblasts (*Cd201*[creER]*Hif1a*[fl/fl]) resulted in a significant delay in wound healing (Fig. 4d), with a closure rate of 72.7% at 9 dpi. We also observed impaired wound closure in mice that were deficient in HIF1α in PDPN[+] proinflammatory fibroblasts (*Pdpn*[creER]*Hif1a*[fl/fl]), which had a wound closure rate of 62.5% at 9 dpi (Fig. 4d). At the same timepoint, wounds of mice deficient in HIF1α in CD201[+] fascia and PDPN[+] proinflammatory fibroblasts exhibited an immature wound bed (Fig. 4e), similar to wounds in the inflammatory phase. In both systems, wound beds did not contract and develop (Fig. 4e), revealing a limited action of contractile proto-myofibroblast and myofibroblast states, which severely impairs the progression to the final phases of wound healing.

We previously showed that fascia fibroblasts transfer pre-made ECM into wounds, which contracts and contributes to the scar matrix[8]. To understand how the transition from immunomodulatory (proinflammatory fibroblast) to contractile and matrix-modifying states (proto-myofibroblast and myofibroblast) affects ECM dynamics in vivo, we labelled the fascia ECM prior to injury (Extended Data Fig. 9a). At 3 dpi, transferred pre-made fascia ECM covered the wound bed, where PDPN[+] proinflammatory fibroblasts were preferably allocated on thinner, looser fibres, whereas pSTAT3[+] proto-myofibroblasts were associated with thicker, woven fibre bundles (Extended Data Fig. 9b). These associations support the notion that differentiation of CD201[+] progenitors into ECM-modifying states (proto-myofibroblast and myofibroblast) mediates matrix malleability and tissue contraction. To test this, we blocked the differentiation into proto-myofibroblast and myofibroblasts with HIF1α inhibitor in mice with labelled fascia ECM. Again, control 3 dpi wounds showed fibre bundles with a condensed organization, whereas treated mice retained slackened and weakly organized fibres (Fig. 4f). We assessed the contraction of the traced matrix by measuring its porosity and complexity using fractal analysis. This analysis revealed a significant reduction in the complexity and an increase in porosity in the HIF1α-inhibition group compared with controls (Fig. 4g), indicative of a relaxed fibre organization compared wuth the more condensed conformation in controls.

Together, our comprehensive scRNA-seq, ex vivo and in vivo chemical inhibition experiments, genetic lineage tracing, gene deletion and matrix fate map studies showed that in mouse, HIF1α gates the second differentiation step from proinflammatory fibroblast to proto-myofibroblast and myofibroblast that controls the timing of the progression of the inflammatory and contraction (proliferation and remodelling) phases of wound healing.

## Trajectory in human skin pathologies

To explore whether fascia progenitor differentiation into myofibroblasts is conserved in humans, we merged scRNA-seq datasets of PDGFRA[+] fibroblasts from fibrotic conditions hallmarked by the

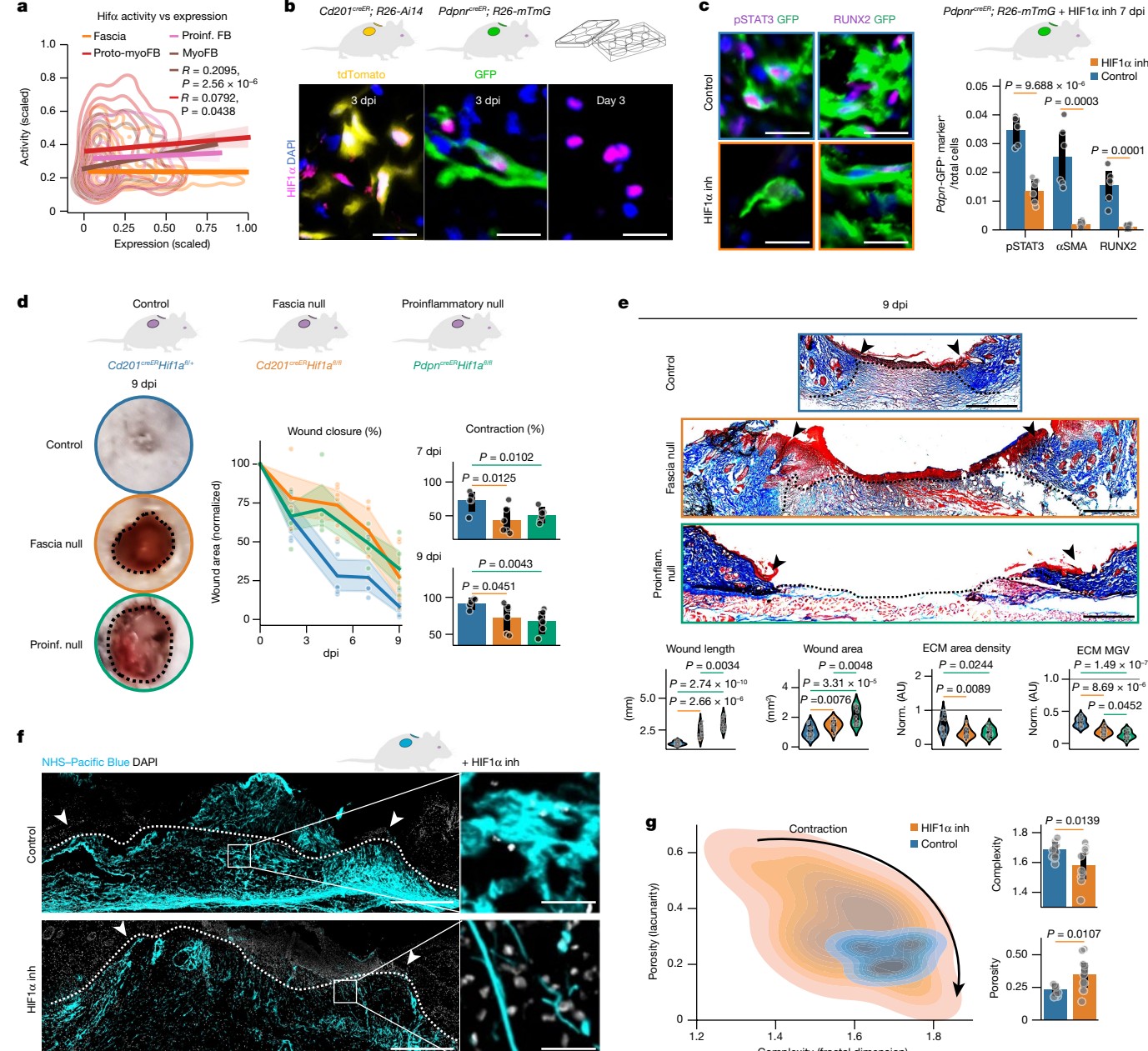

**Fig. 4 | HIF1α mediates the transition to proto-myofibroblasts and tissue contraction. a**, HIF1α activity versus expression correlations in the different fibroblast clusters from the scRNA-seq data. Pearson's *R* coefficient is shown. **b**, Representative micrographs of wounds in *Cd201*^creER *R26*^Ai14 and *Pdpn*^creER *R26*^mTmG mice, and fascia explant at indicated timepoints after injury or culture showing the expression of HIF1α. **c**, Representative high-magnification images of control and HIF1α inhibitor-treated *Pdpn*^creER *R26*^mTmG wounds showing the expression of pSTAT3 or RUNX2 in GFP-traced cells (left) and marker-expressing cell ratios (right). *n* = 6 wounds from 3 biological replicates. **d**, Representative photographs of control, fascia or proinflammatory HIF1α null wounds at 9 dpi (left). Wound area quantification over time (middle), and contraction at 7 dpi (top right) and at 9 dpi (bottom right). Colours of the image outlines indicate treatments in the graph. *n* = 6 wounds from 3 biological replicates for each

genotype. **e**, Masson's trichrome staining (top) and wound maturation-related measurements (bottom) of 9 dpi wounds from indicated genotypes. *n* = 12 images from 3 biological replicates per each genotype. **f,g**, Fascia ECM fate mapping in 3 dpi wounds treated with HIF1α inhibitor. **f**, Representative low-magnification (left) and high-magnification (right) images of control or HIF1α inhibitor-treated wounds. **g**, Lacunarity and fractal dimension density plots to assess porosity (lacunarity) and shape complexity (fractal dimension) differences in labelled ECM from control or HIF1α inhibitor-treated wounds (left) and individual comparisons (right). *n* = 11 images from 3 biological replicates for each condition. The dotted line delimits the wound region. Arrowheads indicate the original injury site. All *P* values (except in **a**) were obtained from two-tailed *t*-tests. Scale bars: 500 μm (**e** and **f**, left) and 50 μm (**b**,**c**, and **f**, right).

action of myofibroblasts, such as Keloid lesions[18], hypertrophic scars[40] and scleroderma[41]. We also included psoriatic skin, an autoimmune disease in which fibroblasts acquire a proinflammatory phenotype[42] (Extended Data Fig. 10a,b). We then used transfer learning to annotate the human fibroblast supercluster using our mouse dataset as reference. As well as the three homeostatic populations, we detected

the proinflammatory fibroblast-like, proto-myofibroblast-like and myofibroblast-like populations in the human skin pathologies (Fig. 5a). Comparing fibroblast compositions between skin pathologies, we noted that fascia fibroblasts were four times more abundant in hypertrophic scars, and papillary fibroblasts were twice as common in scleroderma samples than other conditions. Proinflammatory fibroblasts

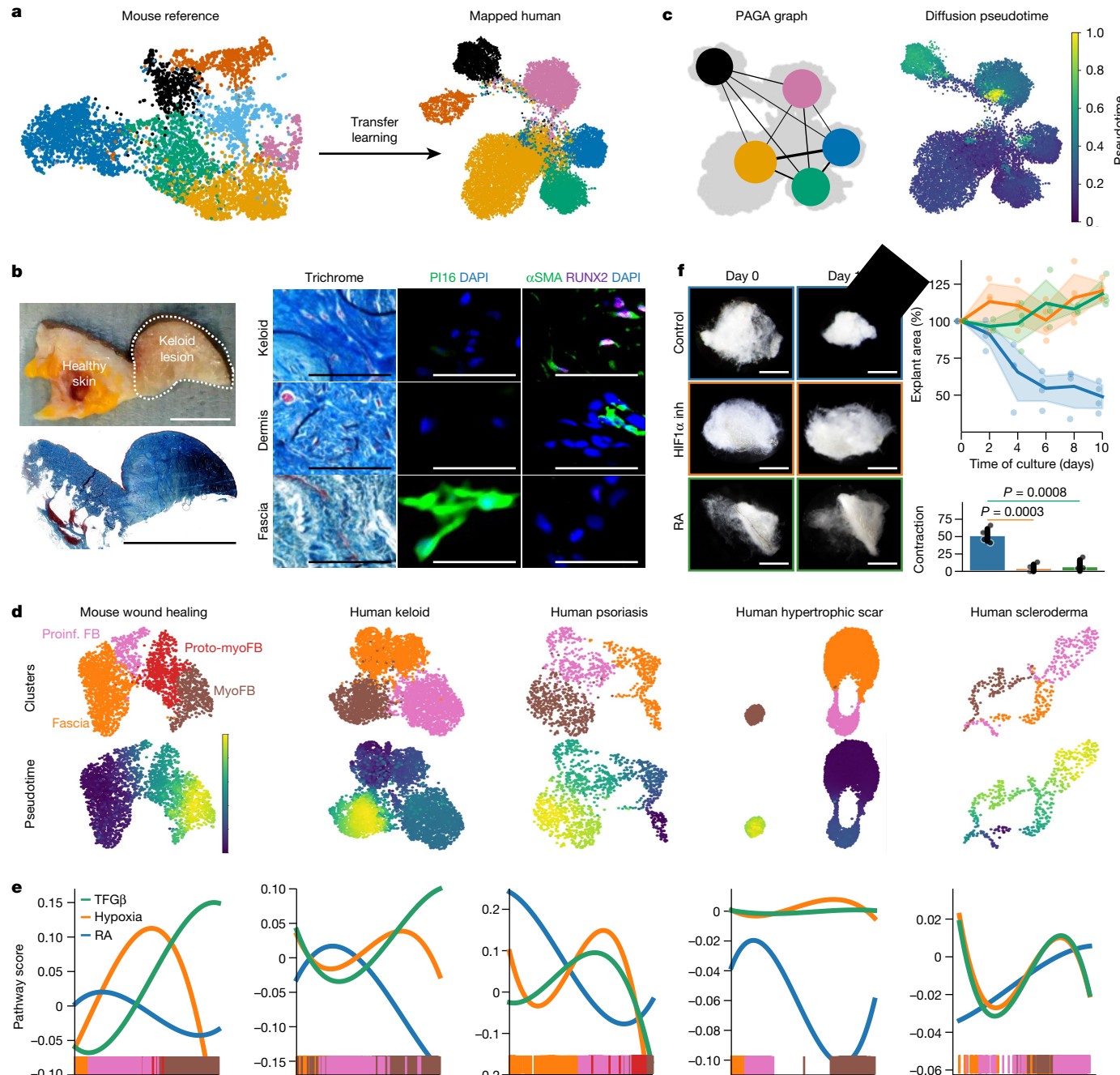

**Fig. 5 | Fascia-to-myofibroblast trajectory in human skin pathologies.**
**a**, 'Transfer learning' for the mapping of human fibroblasts using the mouse atlas as a reference. UMAP representation of mouse (left) and human fibroblasts (right). **b**, Photograph (top left) and low-magnification image of trichrome-stained section (bottom left) of a human keloid lesion. Right, high-magnification micrographs of indicated regions of healthy and keloid tissue that were trichrome-stained or immunolabelled for the fascia marker PI16 or the myofibroblast markers αSMA and RUNX2. **c**, PAGA connectivities embedded onto the UMAP graph of the human fibroblast supercluster (left) and the diffusion pseudotime arrangement (right). **d**, UMAP analyses of subclusters involved in the fascia-to-myofibroblast trajectories in mouse wound healing and indicated human skin pathologies (top) and their pseudotime ordering (bottom), showing that fascia fibroblasts must transition to proinflammatory fibroblasts before becoming myofibroblasts. **e**, RA, hypoxia and TGFβ signalling pathway activities across the inferred trajectories in all datasets, showing the conserved connections of RA to the proinflammatory state and hypoxia to the myofibroblast state. **f**, HIF1α inhibitor and RA treatments in human fascia explants replicate mouse experiments. Representative bright-field photographs of control, RA-, or HIF1α inhibitor-treated fascia explants at indicated timepoints after culture (left), explant area over time (top right) and total contraction (bottom right). Colours of the image outlines indicate treatments in the graph. *n* = 4 technical replicates for each condition. The dotted line delimits the keloid lesion. Two-tailed *t*-tests. Scale bars: 2 mm (**e**), 500 μm (**b**, left) and 50 μm (**b**, right).

and myofibroblasts were present in all conditions. Proinflammatory fibroblasts were more prominent in keloids and scleroderma, whereas myofibroblasts were more abundant in psoriatic skin. Notably, proto-myofibroblasts were only present in hypertrophic scars and

psoriatic skin samples (Extended Data Fig. 10c). We next explored the conservation of the population-specific markers by comparing the expression of the top markers of the human clusters to their mouse counterpart. The most conserved marker for papillary fibroblasts

across mouse and human was *CD9*, and *APOE*, *CFH* and *IGFBP7* marked the reticular population in both species (Extended Data Fig. 10d). Conserved markers for proinflammatory fibroblasts included *CCL2*, *CXCL2* and *NFKBIA*, and *COL12A1* marked the proto-myofibroblast cluster (Extended Data Fig. 10d,e). The fascia clusters had *PI16* as the most specific marker in both human and mouse datasets (Extended Data Fig. 10d,e). Immunolabelling of *PI16* in human keloid and healthy dermis and hypodermis corroborated that *PI16* protein expression was restricted to fascia connective tissue in the hypodermis and was absent in dermis or keloid lesions (Fig. 5b). By contrast, myofibroblasts showed the highest conservation of marker genes, including *POSTN*, several collagen genes and *RUNX2* (Extended Data Fig. 10d,e). As *RUNX2* is not a classical myofibroblast marker, we confirmed its presence within keloid lesions in αSMA$^+$ cells and its absence from healthy tissue (Fig. 5b). These results strongly indicate that homeostatic and wound fibroblasts from mouse wound healing strongly resemble human fibroblasts.

We next analysed the trajectories between the human fibroblast clusters. Given that proto-myofibroblasts were present in only half of the conditions, we omitted them to obtain a general trajectory roadmap applicable for all pathologies. In this initial trajectory analysis, we observed stronger connectivities within all homeostatic populations and into the proinflammatory and myofibroblast populations as terminal states (Fig. 5c), indicating variable contributions from the homeostatic populations to the wound fibroblast pools.

We then assessed the conservation of the fascia fibroblast differentiation trajectory to myofibroblasts in each disease. All pathologies showed an initial transition to proinflammatory fibroblasts before transiting to myofibroblasts; scleroderma was the sole exception that showed the proinflammatory fibroblast as the terminal state (Fig. 5d). These analyses indicate that human fascia fibroblasts transition to myofibroblasts through a proinflammatory state, as in mouse wound healing, and that this differentiation process can be disturbed in skin pathologies.

We then investigated the conservation of the two signalling gates, RA and hypoxia. Comparative analysis of the chronology of these signalling pathways with the human datasets showed that RA activity peaked in the proinflammatory state across all diseases with the exception of psoriatic skin, in which RA activity peaked in the fascia fibroblasts (Fig. 5e). Conversely, hypoxia signal activity closely correlated with the TGFβ signalling pathway along the transition from the proinflammatory to myofibroblast state in all human pathologies (Fig. 5e). Furthermore, human fascia explant contraction halts when exposed to exogenous RA or HIF1α inhibitor (Fig. 5f), indicating a conserved role of these two signalling gates during fascia fibroblast differentiation to myofibroblasts.

Together, our analyses indicate that similar to mouse, human fascia fibroblasts undergo a differentiation trajectory to proinflammatory fibroblasts, partially supported by RA signalling, and to later contractile myofibroblasts, mediated by HIF1α-mediated hypoxia signalling. This differentiation trajectory has a major impact in the progression from the inflammatory phase to the tissue contraction phases (proliferation and remodelling) of wound healing.

## Discussion

This study revealed a conserved differentiation process with clinical implications for injury repair. At the root of this process are fascia progenitors that give rise to specialized fibroblast types and two regulatory gates needed for the timely progression of the wound healing phases.

The first transition of fascia progenitors to proinflammatory fibroblasts has an important effect on the initial inflammatory phase. Our data revealed that RA supports this phase by promoting a monocyte recruitment phenotype and by limiting their subsequent conversion into myofibroblasts, accounting for the observed anti-fibrotic effects of exogenous RA[11]. The second differentiation step from proinflammatory fibroblast to proto-myofibroblast, marked by the acquisition of

a contractile phenotype, enables the transition from the inflammatory to the proliferation and remodelling phases of wound healing. Previous studies highlight the importance of HIF1α in fibroblasts for timely wound healing[43]. Besides its classical roles of reactive oxygen species clearance and angiogenesis regulation[44], HIF1α has previously been only indirectly linked to myofibroblast transition via its synergy with the TGFβ signalling in vitro[45]; our data provide direct evidence for upstream regulation of the TGFβ pathway by HIF1α.

An important caveat to consider in this present study, is the limited number of biological replicates in our in vivo experiments. Nonetheless, multiple lines of evidence, including cross-species in silico analyses, novel ex vivo systems and the use of several complementary chemical and genetic mouse models, point towards the conclusion that the conserved differentiation process of fascia fibroblasts to myofibroblasts orchestrates the progression of the wound healing phases during skin repair.

Despite early reports showing a prevalence of myofibroblasts in upper wound regions[46], little attention has been paid to the spatial distribution of specialized fibroblast populations within wounds. Our results draw an important distinction between wound compartments in several ways. CD201$^+$ progenitors are found within the very deep layers of the skin, where they follow spatial bottom-up cues during their differentiation road upon injury. The first spatial cue occurs in the wound bed where progenitors differentiate into proinflammatory fibroblasts and RA signalling is at its highest. In parallel, the second spatial cue appears within the oxygen deprived wound bed in the form of an intrinsic hypoxia response in proinflammatory fibroblasts. The combination of these two spatial cues promotes the controlled transition into proto-myofibroblasts while limiting their terminal differentiation into myofibroblasts due to the activity of RA. The final spatial cue occurs in upper wound regions during the transition to the proliferation phase, where RA is degraded and terminal differentiation into myofibroblasts is granted. Thus, our data indicate that the interplay between RA gradients in hypoxic niches regulate fibroblast differentiation in response to injury.

These results have important clinical implications, as alterations in this differentiation process could result in pathologies such as impaired wound healing and fibrotic scarring. In particular, the manipulation of individual steps along the trajectory opens new roads to modulate different wound fibroblasts to obtain disease-tailored therapies. Treatments of wounds (such as surgical and keloid wounds, and burns) could be modulated to restrain myofibroblasts to prevent excessive scarring and contraction without compromising the proinflammatory role needed during the inflammatory phase, and treatments tailored for modulating the early progenitors would be beneficial for inflammatory skin diseases.

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

# Methods

## Mice and genotyping

C57BL/6 J wild type, En1tm2(cre)Wrst/J (*En1^cre^*), B6.Cg-Gt(ROSA)26So rtm14(CAG-tdTomato)Hze/J (*R26^Ai14^*), B6.129(Cg)-Gt(ROSA)26Sortm 4(ACTB-tdTomato.-EGFP)Luo/J (*R26^mTmG^*), and B6.129S6(Cg)-Gt(ROSA) 26Sortm1(DTA)Jpmb/J (*R26^DTA^*) mouse lines were obtained from Charles River or Jackson laboratories. The *Pdpn^creER^* line was generated in-house and *Cd201^creER^* line was a gift from A. Zeng. B6.129-Hif1atm3Rsjo/J (*Hif1a^flox^*) mice were a gift from the Institute of Diabetes and Obesity, Helmholtz Centre Munich. Animals were housed in the Helmholtz Central Animal Facility; cages were maintained at constant temperature and humidity, with a 12-h light cycle and provided with food and water ad libitum. All animal experiments were reviewed and approved by the Upper Bavarian state government and registered under projects 55.2-1-54-2532-16-61 and 55.2-2532-02-19-23 and strict government and international guidelines. This study complies with all relevant ethical regulations regarding animal research. Animal experiments were performed using 8- to 12-week-old adult mice. Relevant genotypes were identified by standard PCR and gel electrophoresis. For this, genomic DNA from ear clips was extracted using Quick Extract DNA extraction solution (Epicentre) following the manufacturer's guidelines. DNA extract (1 µl) was added to each 19 µl PCR reaction. The reaction mixture was set up using GoTaq PCR core kit (Promega) containing 1× GoTaq green master mix, 0.5 µM forward and reverse primers. The following primers were used: En1-Cre-FW (5′-ATTGCTGTCACTTGGTCGTGGC-3′), En1-Cre-RV (5′-ggaaaatgcttctgtccgtttgc-3′), Procr-Cre-FW (5′-gcggtct ggcagtaaaaactatc-3′), "Procr-Cre-RV" (5′-gtgaaacagcattgctgtcactt-3′), Pdpn-Cre-FW (5′-gatggggaacagggcaagttgg-3′), Pdpn-Cre-RV (5′-ggctctacttcatcgcattccttgc-3′), HIF1a-FW (5′-tgcatgtgtatgggt gtttg-3′) and HIF1a-RV (5′-gaaaactgtctgtaacttcatttcc-3′). All primers were obtained from Sigma Aldrich. PCRs were performed with initial denaturation for 5 min at 94 °C, amplification for 35 cycles (denaturation for 30 s at 94 °C, hybridization for 30 s at 59 °C, and elongation for 1 min at 72 °C) and final elongation for 10 min at 72 °C, and then cooled to 4 °C. In every run, negative controls (non-template and extraction) and positive controls were included. The reactions were carried out in an Eppendorf master cycler. Amplified bands were detected by electrophoresis in agarose gels. Other mouse lines (*mTmG*, *Ai14* and *DTA*) were maintained in a homozygous background.

## In vivo wounds, tamoxifen induction and treatments

Male and female mice, equally distributed between time groups, were used for descriptive experiments. For functional studies, littermates or age-matched mice were randomly assigned into the different experimental groups. To decrease variability, only males were used for the RARγ agonist and HIF1α inhibition treatments (7 dpi), and for the *Hif1α* genetic deletion experiment. Females and males, equally distributed between treatment groups, were used for the PDPN^+^ cell ablation studies and for the 3 dpi HIF1α inhibition experiment. Back-skin full-thickness wounds on adult (8- to 12-week-old) mice were performed as previously[8]. In brief, bilateral 5-mm circular wounds were performed by piercing through a skin fold along the midline of the back using biopsy punches (Stiefel 10005). For the splinted wound model[47], immediately after making the excisional wounds, circular silicon rings were glued (Gracebio 664581) and sutured around the open wounds. For the genetical lineage tracing of CD201^+^ fascia fibroblasts and *Hif1αα* gene deletion, *Cd201^creER^R26^Ai14^*, *Cd201^creER^Hif1a^fl/+^*, *Cd201^creER^Hif1a^fl/fl^*, and *Pdpn^creER^Hif1a^fl/fl^* mice received three intraperitoneal injections of 2 mg (Z)−4-Hydroxytamoxifen (Sigma Aldrich H7904) diluted in 0.2 ml miglyol-812 (Caesar & Lorentz GMBH CSLO3274) two days and one day before wounding, and at the day of wounding. For the genetic lineage tracing and ablation of PDPN^+^ proinflammatory fibroblasts, *Pdpn^creER^R26^mTmG^* or *Pdpn^creER^R26^DTA^* mice received daily intraperitoneal injections of 1 mg (Z)−4-hydroxytamoxifen diluted in 0.1 ml miglyol-812

from day four before wounding to the day of injury. For fascia matrix fate mapping[8,48], 40 µl of 2 mg ml⁻¹ Pacific Blue succinimidyl Ester (Invitrogen P10163) with 0.1 M sodium bicarbonate in saline were subcutaneously injected on the mouse back three- and one day prior to wounding to label the fascia matrix. HIF1α inhibitor (echinomycin, Sigma Aldrich sml0477), at a concentration of 100 µg kg⁻¹ body weight, RARγ agonist (BMS 961, Tocris 3410), at a concentration of 100 µg kg⁻¹ body weight, or vehicle control dimethyl sulfoxide (DMSO, Sigma Aldrich D5879) were subcutaneously injected near and below the wounds immediately after injury and every other day to have a more direct effect on fibroblasts.

## Free-floating ex vivo fascia culture

Whole-back skin was dissected from euthanized mice using scissors and forceps taking care that the fascia beneath remained intact. Human abdominal skin was obtained from the Plastic Surgery department from the Technical University Munich (ethical approval number 496/21-S-KH), patient details remain anonymous due to privacy concerns. For both human and mouse, samples were washed twice in ice-cold PBS (Thermo Fischer) to remove any residual debris and placed with the epidermal side facing down. The soft fascia tissue was then pulled with forceps and cut into approximately four-millimetre pieces and washed in PBS. The fascia explants were then placed in triplicate in 6-well plates supplied with DMEM/F-12 medium containing 10% normal FBS, heat-inactivated FBS (BioConcept, 2-02F110-I; heat-inactivated at 56 °C for 30 min), or 0.5% low serum concentrations, 1× GlutaMAX (Thermo Fisher Scientific 35050038), 1× Penicillin/ streptomycin (Thermo Fisher Scientific 15140122), and 1× MEM non-essential amino acids (Thermo Fisher Scientific 11140035). Media was changed every other day. For ex vivo cell ablation 3 cycles of freezing (1 h at −80 °C) and thawing (30 min at ambient temperature) were performed before culture. For treatment groups, the media was supplemented with additional compounds and refreshed during media change. The compounds used include the HIF1α inhibitor (0.1, 10 µM, echinomycin, Sigma Aldrich sml0477), RA (0.01 and 1 µM, R&D systems 0695), pan-RAR antagonist (0.01 µM, AGN193109, Tocris 5758), RARα agonist (0.01 µM, AM580, Tocris 0760), RARβ agonist (0.01 µM, CD2314, Tocris 3824), RARγ agonist (0.01 µM, BMS 961, Tocris 3410), CYP26B1 inhibitor (0.01 µM, R115866, Sigma Aldrich SML2092), TGFβ1 (10 µM, recombinant mouse TGFβ1 protein, R&D systems 7666-MB-005/CF), and TAZ agonist (10 µM, Kaempferol, R&D systems 3603/50). For the combined treatments, the effective doses for HIF1α inhibitor and RA were 10 and 1 µM, respectively. Samples were maintained in a humidified 37 °C, 5% CO₂ incubator for up to 6 (mouse samples) or 10 days (human samples). The fascia tissue contraction was assessed by daily imaging with a stereomicroscope (Leica M50).

## Fascia fibroblast in vitro culture and qPCR

The dorsal fascia was manually separated from the dermis of adult wild-type mice as before, dissociated and incubated (300 rpm rocking) for 1 h at 37 °C in a digestion solution containing 0.4 mg ml⁻¹ Collagenase A (Sigma 10103586001), 0.1 mg ml⁻¹ Liberase TM (Sigma 5401119001), and 40 U ml⁻¹ DNase (Sigma D4263-5VL) in PBS. Then, the suspension was filtered through 70-µm and then 40-µm sieves. Purification of CD201^+^ fibroblasts was performed by using Magnetic-Activated Cell Sorting (MACS, Miltenyi 130-106-585) according to the manufacturer's instructions (Miltenyi 130-048-801 and 130-042-401). CD201^+^ fibroblasts were cultured in the 10% FBS-containing medium as the fascia explants in the same environmental conditions. All experiments were performed with at least three sets of matched primary fibroblast lines between passages 2 and 6. For the treatments, CD201^+^ fibroblasts were serum-starved in DMEM/F-12 media supplemented with 2% heat- inactivated FBS for 6 h prior to the addition of 1 ng ml⁻¹ mouse recombinant Interleukin (IL)−1β (R&D systems 401-ML-005) or 10 ng ml⁻¹ TGFβ1 (R&D systems 7666-MB-005) in combination with RA at increasing concentrations (0.1 µM, 1 µM and 5 µM). All treatments

were performed in low-serum medium (2% hiFBS). After 48 h of treatment, RNA was extracted using the QIAGEN RNeasy Kit (QIAGEN 74104) according to the manufacturer's instructions. One microgram of RNA was used for cDNA synthesis with the cDNA synthesis kit (LIFE Technologies AB1453A). Quantitative PCR (qPCR) was performed using SYBR Green PCR Master Mix (Thermo Fisher 4309155). Each sample was run in triplicate and initially normalized to the house-keeping gene *Gapdh*. Expression changes were analysed using the $\Delta\Delta C_t$ method. Primers used were: *Gapdh* (forward, 5′-AGGTCGGTGTGAACGGATTTG-3′; reverse, 5′-TGTAGACCATGTAGTTGAGGTCA-3′), *Ccl2* (forward, 5′-TTAAAAACCTGGATCGGAACCAA-3′; reverse, 5′-GCATTAGCTTCAGATTTACGGGT-3′), *Acta2* (forward, 5′- GTCCCAGACATCAGGGGAGTAA-3′; reverse, 5′-TCGGATACTTCAGCGTCAGGA-3′), *Cxcl1* (reverse, 5′-ACTGCACCCAAACCGAAGTC-3′; forward, 5′-TGGGGACACCTTTTAGCATCTT-3′), and *Pdpn* (reverse, 5′-GTTTTGGGGAGCGTTTGGTTC-3′; forward, 5′-CATTAAGCCCTCCAGTAGCAC-3′).

### Histology

At relevant timepoints, samples were fixed overnight with 4% paraformaldehyde (PFA, VWR 43368.9 M), washed thrice with fresh PBS and processed for cryosectioning using OCT (CellPath KMA-0100-00A) in a Cryostat (CryoStar NX70). Sections were rinsed three times with PBS containing 0.05% Tween-20 (PBST) and blocked for 1 h at room temperature with 10% serum in PBST. Then, they were incubated with primary antibody in blocking buffer for 3 hr at ambient temperature or at 4 °C overnight. Sections were then rinsed three times with PBST and incubated with secondary antibody in blocking solution for 60 min at ambient temperature. Finally, they were rinsed three times in PBST and mounted with fluorescent mounting media with 4,6-diamidino-2-phenylindole (DAPI). For whole-mount staining, fixed-fascia pieces (ca. 1 cm$^2$) were incubated in 0.2% Gelatine (Sigma G1393), 0.5% Triton X-100 (Sigma X100), and 0.01% Thimerosal (Sigma T5125) in PBS (PBSGT) for 24 h. Then, they were incubated with the primary antibody in PBSGT for 48 h at ambient temperature followed by the incubation with species-relevant secondary antibody in PBSGT for at least 48 h or until imaged. Primary antibodies used: PDPN (Abcam ab11936, 1:500 dilution), pSTAT3 (Cell Signaling Technology 9145 S, 1:150), RUNX2 (Abcam ab92336, 1:150), GFP (Abcam ab13970, 1:500), PDGFRα (R&D systems AF1062, 1:100), KRT14 (Abcam ab181595, 1:100), PECAM1 (also known as CD31) (Abcam ab56299, 1:50), LYVE1 (Abcam ab14917, 1:100), αSMA (Abcam ab21027, 1:150), YAP1 (Abcam ab205270, 1:100), pSMAD2 (Cell Signalling 18338, 1:100), HIF1A (Novus NB100-479, 1:100), CCL2 (Abcam ab25124, 1:100), CXCL1 (R&D systems MAB453R, 1:100), ALDH1A3 (Novus NBP2-15339, 1:100), CYP26B1 (Elabscience E-AB-36196, 1:100), CD45/PTPRC (Abcam ab23910, 1:100) and PI16 (R&D systems AF4929, 1:100). For some of the primary antibodies (pSTAT3, RUNX2 and HIF1A), antigen retrieval was performed using citrate buffer at 80 °C for 30 min, allowing it to cool down for another 30 min before the blocking step. Alexa Fluor488-, Alexa Fluor568- or Alexa Fluor647-conjugated secondary antibodies (1:500 dilution, Life Technologies) against respective species were used. Masson's trichrome staining was performed using a commercial kit (Sigma Aldrich HT15) following manufacturer's recommendations. Histological sections were imaged in a using a ZEISS AxioImager.Z2m (Carl Zeiss) using the 20× objective or a THUNDER Imager Model Organism (Leica) using a compiled zoom of 20×. Whole-mount samples were imaged using a SP8 Multiphoton microscope (Leica).

### Image Analysis

All image analysis was performed using Fiji (v.1.53c). For assessment of tissue contraction in vitro and wound closure in vivo, the explant or wound area from different timepoints was selected using the 'selection brush' tool and measured using the 'measure' function. Area values were normalized in percentage to the initial (day 0, $A_0$) as 100%. In vitro final contraction ($C$) was defined as the difference between the highest area value of the two initial days after culture (day 0–2, $A_{max}$) minus the lowest area value from the last two days of culture (day 4–6 $A_{min}$); thus, $C = A_{max} - A_{min}$. Wound length was determined as the mean length from three individual measurements in each section by tracing lines covering along (1) the new epidermis, (2) distance between flanking hair follicles, and (3) between the branched *Panniculus carnosus* muscle using the line measurement tool. For cell marker-positive quantifications, individual channels were pre-processed using the functions 'substract background' (rolling=20px), 'median' (radius=1), 'unsharp mask' (radius=1, mask=0.60), 'despeckle', 'enhance contrast' (saturated=0.1 normalize), and 'auto threshold' methods were empirically selected for individual markers: Moments (for DAPI, pSTAT3, YAP1, pSMAD2, CD45, and GFP signal), Intermodes (RUNX2, tdTomato, PDGFRα, and LYVE1), Yen (CCL2), IsoData (PDPN), or RenyiEntropy (αSMA). The wound region of interest (ROI) was manually selected using the 'selection brush' tool taking as reference the breached *P. carnosus* below the dermis and the flanking hair follicles in the epidermis above. Total wound cell numbers (within the ROI) from binary images were measured using the DAPI channel and using the functions 'fill holes', 'watershed', and 'analyze particles' (size=70-1400px) in that order. Marker-positive cell numbers were measured by using the mask of the marker with the DAPI channel using the 'image calculator_AND', 'dilate' twice, 'close', 'fill holes', 'watershed', and 'analyze particles' (size=70-1400px) functions. For double marker-positive cell calculations, an initial mask from two marker channels was created and processed as before. For wound region quantifications, 3 ROIs were manually selected with the 'selection brush' tool. The wound lateral limits were delimited as stated before, the upper wound region extended vertically 200 microns from the wound surface (at 3 dpi) or the wound epidermis (at 7 dpi). The fascia was measured as the area extending below the breached *P. carnosus* muscle down to the dorsal muscles, while the wound core covered the area between the wound fascia and the upper wound. Non-mesenchymal areas (such as hair follicles and wound epidermis) were deleted from the ROIs. Marker channels were pre-processed as before and the mean grey value (MGV) for each ROI was then quantified with the 'measure' function. For visual aid, marker signal intensity was represented with the 'fire' or '16-color' pseudo-colours. ECM area density was determined by converting Masson's trichrome-stained RGB images into CMYK using the RGB-to-CMYK plugin. Then, the cyan channel (ECM) was extracted and the MGV measured before being pre-processed as above. ECM density was defined as the ECM (cyan) area in the wound covering the total wound area. ECM density and ECM MGV in wounds were normalized to the healthy adjacent dermis to avoid batch effects between stained sections. Fractal analysis was performed as before[7]. In brief, the cyan channel was pre-processed as before, and binary images were generated using 'auto threshold' ('Default' method) function. Binary images were then analysed with the FracLac plugin using the same settings as before. All quantification plots were generated using the Python-based packages Matplotlib and Seaborn.

### Flow cytometry

Skin and wounds were collected using a 8 mm biopsy punch (Stiefel 10008) and fascia and dermis were separated as described above. Tissue was defatted, finely minced with scissors, and incubated in digestion solution (0.2 mg–1 mg ml$^{-1}$ liberase, 0.5 mg ml$^{-1}$ collagenase A, 100 U ml$^{-1}$ DNase in serum-free DMEM) for 60 min at 37 °C while rocking at 350 rpm. Digestion was stopped with 10 ml DMEM, samples were vortexed for few seconds, and then strained through 70-μm sieves. For the isolation of bone marrow cells, tibiae and femur of mice were dissected, flushed using a syringe with a G21 needle, and then filtered through 40-μm filter to remove tissue debris. Single-cell suspensions were fixed with 2% PFA to preserve surface markers stability for long periods. Single-cell suspensions were pelleted and suspended in PBS containing fluorophore-conjugated antibodies for 30 min before analysis. All antibodies were used in a 1:200 dilution except for the

CD45 antibody, which was diluted 1:800. Antibodies used: CD45 (also known as PTPRC)–APC and CD45–PE/Cy7 (30-F11, Biolegend), CD31–APC (390, e-Biosciences), TER119–APC (Ter119, Biolegend), CD326 (also known as EPCAM)–AF647 (G8.8, Biolegend), CD11b–AF488 (M1/70, Biolegend), LY6G–PacBlue (1A8, Biolegend), F4/80 (also known as ADGRE1)–APC (BM8, Biolegend), CD3–PE/Cy7 (500A2, Biolegend), CD19–BV510 (6D5, Biolegend), CD140a (also known as PDGFRA)–PE–Cy7 (APA5, e-Biosciences). All samples were run on a BD LSRII cytometer (BD Biosciences) equipped with violet, blue, and red lasers and analysed in the BD FACS Diva analyser suite (BD Biosciences). For all cell types, initial gate from forward scatter (FSC-A) versus side-scatter (SSC-A) plots was used. From this, single-cell gate from FSC-A versus FSC-H plots was used to exclude debris. Strict doublet exclusion was performed prior gating for immune cells (CD45$^+$) and stromal cells (Lin$^-$: CD45$^-$ CD31$^-$ Ter119$^-$ EpCAM$^-$). Monocytes and macrophage cells were gated as ADGRE1$^+$, neutrophils as LY6G$^+$, T cells as CD3$^+$, and B cells as CD19$^+$. Similarly, tdTomato$^+$ cells from $Cd201^{creER}R26^{Ai14}$ mice were gated from Lin$^-$ cells then PDGFRA$^+$, and from CD31$^+$, CD45$^+$ and EPCAM$^+$ cells. Signal compensation was performed on FMO controls using compensation beads (Thermo Fisher 01-3333-42).

### Stromal cell enrichment for scRNA-seq
Eight-millimetre-diameter biopsies of wounds and surrounding skin were taken from $En1^{cre}R26^{mTmG}$ dorsal skin wounds at relevant times after injury. Samples from three mice were pooled for each stage (six wounds per timepoint). Collected tissue was minced with surgical scissors and digested with Liberase TM enzymatic cocktail supplemented with 25 U ml$^{-1}$ DNase I (Sigma Aldrich D4263) at 37 °C for 60 min. The resulted single-cell suspension was filtered and incubated at 4 °C for 30 min with conjugated primary antibodies (dilution 1:200) against specific cell markers of unwanted cell type lineages. Antibodies used: APC-anti-CD31 (eBioscience 17-0311-82), eFluor660-anti-Lyve1 (eBioscience 50-0443-82), APC-anti-Ter119 (Biolegend bld-116212), and APC-anti-CD45 (Biolegend Bld-103112). Cells were then washed and incubated at 4 °C for 20 min with suitable magnetic beads against the conjugated-fluorophores and dead cell markers (Dead Cell removal kit, Miltenyi 130-090-101). Microbeads used: Anti-Cy5/Alexa Fluor647 (Miltenyi 130-090-855) and anti-APC (Miltenyi 130-091-395). Cells were separated using OctoMACS and MS columns (Myltenyi) according to manufacturer's guidelines. Negative selection was then washed, counted, and diluted at 100 cells per microliter in PBS with 0.04% BSA and 300 U ml$^{-1}$ of RNase inhibitor (RiboLock LIFE technologies eo0382). Only samples with >90% living cells were used for Dropseq separation.

### Dropseq libraries preparation and sequencing
Stromal-enriched cell suspensions were separated using the Dropseq[49] microfluidic-based method following the McCaroll lab protocol with a few adaptations. In brief, using a microfluidic PDMS device (Nanoshift), cells were encapsulated in droplets with barcoded beads (120 per µl, ChemGenes Corporation, Wilmington, MA) at a rate of 4,000 µl h$^{-1}$. Droplet emulsions were collected for 15 min each prior to droplet breakage with perfluoro-octanol (Sigma-Aldrich). After breakage, beads were collected, and the hybridized mRNA transcripts were reverse transcribed (Maxima RT, Thermo Fisher). Unused primers were removed by the addition of exonuclease I (New England Biolabs). Beads were then washed, counted, and aliquoted for pre-amplification (2,000 beads per reaction, equal to ~100 cells per reaction) with 12 PCR cycles (with the recommended settings). PCR products were pooled and purified twice by 0.6× clean-up beads (CleanNA). Prior to tagmentation, cDNA samples were loaded on a DNA High Sensitivity Chip on the 2100 Bioanalyzer (Agilent) to ensure transcript integrity, purity, and amount. For each sample, 1 ng of pre-amplified cDNA, from an estimated 1,000 cells, was tagmented by Nextera XT (Illumina) with a custom P5 primer (Integrated DNA Technologies). Single-cell libraries were sequenced in a 100 bp paired-end run on an Illumina

HiSeq4000 using 0.2 nM denatured sample and 5% PhiX spike-in. For priming of the first read, 0.5 µM Read1CustSeqB (primer sequence: 5'-GCCTGTCCGCGGAAGCAGTGGTATCAACGCAGAGTAC-3') was used. STAR (version 2.5.2a) was used for mapping the reads and to align them to the mm10 genome reference (provided by Drop-seq group, GSE63269) that was tailored to include the eGFP cDNA transcript.

### scRNA-seq data analysis
All the analyses were performed using the phyton toolkit Scanpy[50] and complementary tools under its ecosystem. Matrices of individual samples were concatenated, quality checked by filtering cells with fewer than 100 genes. Genes that appeared in less than 50 cells and cells containing less than 0.5% of the total genes in the dataset were also filtered out. To decrease batch- and cell cycle effects, combat and cell cycle regression algorithms were used as recommended by the developers. The UMAP algorithm was used as the preferred dimensional reduction method.

Cluster annotation was selected by increasing iteratively threshold values while preserving a defined cell cluster in the dataset. In the whole mouse dataset, the resolution used was the highest possible that still preserved all the Plp1-expressing cells (Schwann cells) in a single cluster. When clustering fibroblast subsets, the control cluster used was Acta2-expressing myofibroblasts. Louvain and Leiden clustering algorithms were used complementarily. Ranked marker genes for each cluster were calculated using the method 'wilcoxon' and used for GO term statistical overrepresentation test with the PANTHERdb web tool. The fibroblast supercluster was defined by the expression of classical pan-fibroblast markers such as $Col1a1$ and $Pdgfra$. Skin fibroblast expression profiles were calculated using the 'gene score' function using previously reported markers[8,17,51–53]. Papillary fibroblasts markers included: $Sparc$, $Ndufa4l2$, $Cldn10$, $Cpz$, $Col3a1$, $Cgref1$, $Col16a1$, $Mfap4$, $Cd63$, $Mt1$, $Fth1$, $Cyp2f2$, $Ccl19$, $Prdm1$, $Lrig1$, $Ephb2$, $Trps1$, $Tmem140$, $Cd302$, $Maf$, $Adra2a$, $Casp1$, $Dendd2a$, $Ntn1$, $Sepp1$ (also known as $Selenop$), $Sipa1l2$, $Tfap2c$, $Axin2$, $Ctsc$, $Osr1$, $Cmtm3$, $Rgl1$, $Hrsp12$ (also known as $Rida$), $Moxd1$, $Itm2c$, $Steap1$, $Tnfrsf19$, $Cadm1$, $Efhd1$, $Ucp2$, $Creb3$, $Creb3l3$, $Mxd4$ and $Apcdd1$. Reticular fibroblast markers included: $Gpx3$, $Cxcl12$, $Cygb$, $F3$, $Gsn$, $Dpt$, $Myoc$, $Tmeff2$, $Hmcn2$, $Krt19$, $Dact1$, $Fstl3$, $Map1b$, $Nexn$, $Gls$, $Fndc1$, $Vgll4$, $Arpc1b$, $Sulf1$, $Cdh2$, $Dbndd2$, $Tgm2$, $Fgf9$, $Limch1$, $Mgst1$, $Npr3$, $Sox11$, and $Vcam1$. Fascia fibroblast markers included: $Plac8$, $Anxa3$, $Akr1c18$, $Pla1a$, $Ifi27l2a$, $Ifi205$, $Sfrp4$, $Prss23$, $Ackr3$, $Nov$ (also known as $Ccn3$), $Fap$, $Rhoj$, $Klf9$, $Sfrp2$, and $Tgfb2$. For the myofibroblast profile the markers $Postn$, $Acta2$, $P4ha1$, $P4ha2$, $P4ha3$, $Col11a1$, $Tnc$, $Tagln$, $Pdgfrb$, and $Vim$ were used. For the bioinformatic 'sorting' of EPFs, the subset of fibroblast cells with a raw count value over 0.5 of the eGFP transcript was annotated independently.

Human skin diseases datasets were obtained from the GEO repository (GSE163973, GSE162183, GSE156326 and GSE160536), merged, and processed as before. Annotation of human fibroblast clusters was performed using scArches[54]. In brief, after training on the reference dataset (mouse) the node weights were transferred to a new model. Human dataset was then tested on the trained model and cells were annotated accordingly. Parameters were initialized in accordance with the scArches optimization.

Mouse trajectories from all fibroblasts and from fascia-to-myofibroblast trajectory clusters were inferred by PAGA with RNA velocity-directed edges using the scvelo toolkit[55,56]. 'Dynamic modeling' was used under standard settings to calculate velocities. Cells were arranged under the inferred trajectory using the 'velocity pseudotime' function rooting the trajectory origin in the most distant cells within the fascia cluster from the rest of the clusters. Human trajectories (unbiased and from fascia-to-myofibroblast trajectory clusters) were inferred using PAGA without RNA velocity due to the non-matched, monotemporal nature of the human samples. Arragment of the trajectories were performed then using diffusion pseudotime instead of velocity pseudotime.

Regulon analysis was performed by mapping the expression of 1623 mammalian transcription factors from the RIKEN database and statistically correlating (Pearson R) their expression with the fibroblastic state signatures. Programmes were selected from the list of transcription factors whose expression positively correlated with a signature that had validated downstream targets in the TRANSFAC and ENCODE databases. Individual programme scores were calculated according to the expression of the selected transcription factors using the score gene function. To identify the complete regulon, the combined list of downstream targets for each programme was mapped and correlated with the programme scores. Positively correlated genes, whose expression matched the upstream regulating programmes, were used for the regulon score using the gene score function and for the GO term overrepresentation test using the PANTHERdb[57] web tool.

Transcription factor activity was inferred using the DoRothEA toolkit[58]. Signalling pathway activity was predicted by mapping the expression of genes from a manually curated list for relevant factors of the TGFβ, Wnt, mechanotransduction, cell-cell adhesion, chemokine, and RA signalling pathways. Complementarily, hypoxia, Wnt, and TGFβ pathway activity was predicted using the PROGENY tool[58], showing similar results, for TGFβ and Wnt signal activity, to our manual method. The HIF1α regulon positively regulated during hypoxia signal was identified by correlating the validated downstream targets of HIF1α from the ENCODE and DoRothEA databases to the hypoxia signal activity from PROGENY.

## Statistics

Statistical analyses were performed using the Python toolkit Scipy. No methods were used to predetermine sample size. No blinding or randomization was used for data analysis. Two-tailed $t$-tests, one-way or two-way ANOVAs, and Pearson's $R$ were implemented with standard settings. Significant differences were considered with $P$ values below 0.05 and are indicated in the figures. In all plots, error bars depict the standard deviation.

## Reporting summary

Further information on research design is available in the Nature Portfolio Reporting Summary linked to this article.

## Data availability

The generated scRNA-seq data has been deposited in https://doi.org/10.5281/zenodo.10013140. All other data that supports the findings of this study are available from the corresponding author upon reasonable request. Source data are provided with this paper.

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

**Acknowledgements** We thank S. Sabrautzki and H. Adler for their veterinary advice and support; A. Zeng for providing the *Cd201creER* transgenic line; and V. Bergen and A. Wolf for scRNA-seq data analysis support. This project was supported by the European Research Council Consolidator Grant (ERC-CoG 819933), the LEO Foundation (LF-OC-21-000835) and the European Foundation for the Study of Diabetes (EFSD) Anniversary Fund Program.

**Author contributions** Y.R. outlined and supervised the research narrative and experimental design. D.C.-G. analysed the scRNA-seq data, performed image analysis and in vivo wound experiments, and designed figures and illustrations. H.Y. and B.D. performed immunohistochemistry, in vivo wound experiments, ex vivo fascia cultures and image analysis. A.S. generated and validated double transgenic mouse lines and supported in in vivo wound experiments. S.K. performed the transfer learning to annotate the human datasets. R. Kandi performed the flow cytometry. Y.L. and R.D. performed the fascia fibroblast purification and treatments. R. Kopplin and D.S.S. performed ex vivo fascia cultures. J.W. supervised and generated the mouse experimentation protocols. R.I. assisted with histology and immunohistochemistry. D.J. supported in the collection of human material from the clinic. M.S., M.A., I.A. and H.B.S. generated the mouse scRNA-seq libraries and assisted with data analysis. T.V. and H.-G.M. assisted with translational and clinical advice. Y.R. and D.C.-G. wrote the manuscript with support from H.Y. and B.D.

**Funding** Open access funding provided by Helmholtz Zentrum München - Deutsches Forschungszentrum für Gesundheit und Umwelt (GmbH).

**Competing interests** The authors declare no competing interests.

**Additional information**
**Correspondence and requests for materials** should be addressed to Yuval Rinkevich.

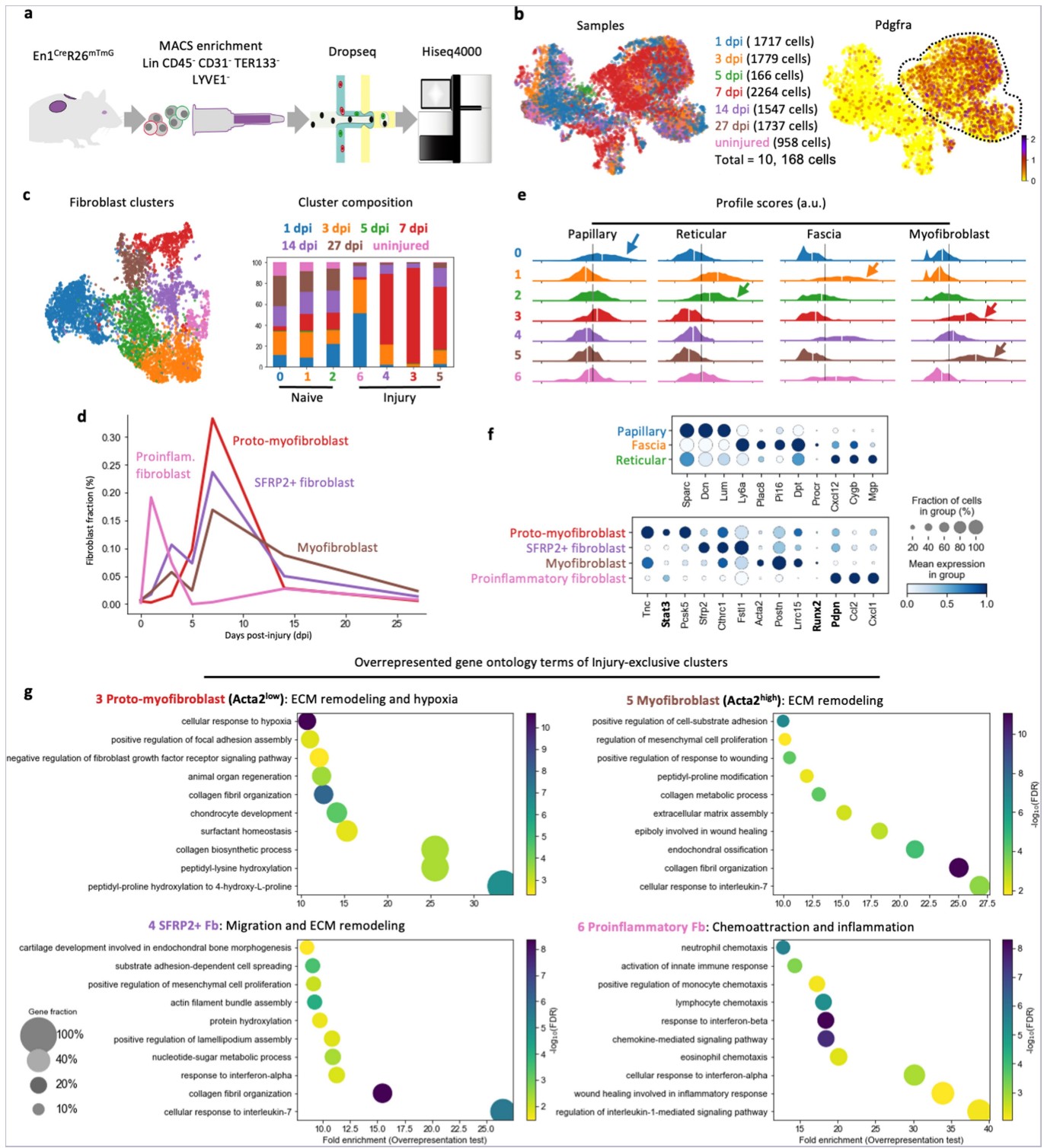

**Extended Data Fig. 1 | Longitudinal single-cell RNA sequencing of stromal cells during murine wound healing. a**. Experimental approach for the Engrailed-1 lineage tracing coupled with stromal cell enrichment and Dropseq method for single-cell sequencing. **b**. UMAP representations of complete dataset color-coded for samples (time points and cell number, left) and *Pdgfra* expression (right) to identify fibroblasts. **c**. UMAP representation of fibroblast clusters (left) and the cluster composition of the different samples (right) revealing homeostatic (always present) and injury-specific populations (absent in uninjured sample). **d**. Fraction in % of each injury-specific population across samples (time-points). **e**. Ridge plots depicting profile scores from indicated fibroblast subtypes in each cluster. Gray lines indicate the overall mean for each profile score and the white lines indicate the cluster mean. Colour arrows indicate clusters with the highest profile for each fibroblast subtype. **f**. Gene expression dot-plots of top markers for homeostatic (left) and injury-related (right) clusters. **g**. Overrepresented GO terms in indicated injury cluster. Scatter plots show the fold enrichment (x-axis) of the term (y-axis), genes expressed/genes in term coverage in % (dot size), and the false discovery rate value (colour gradient).

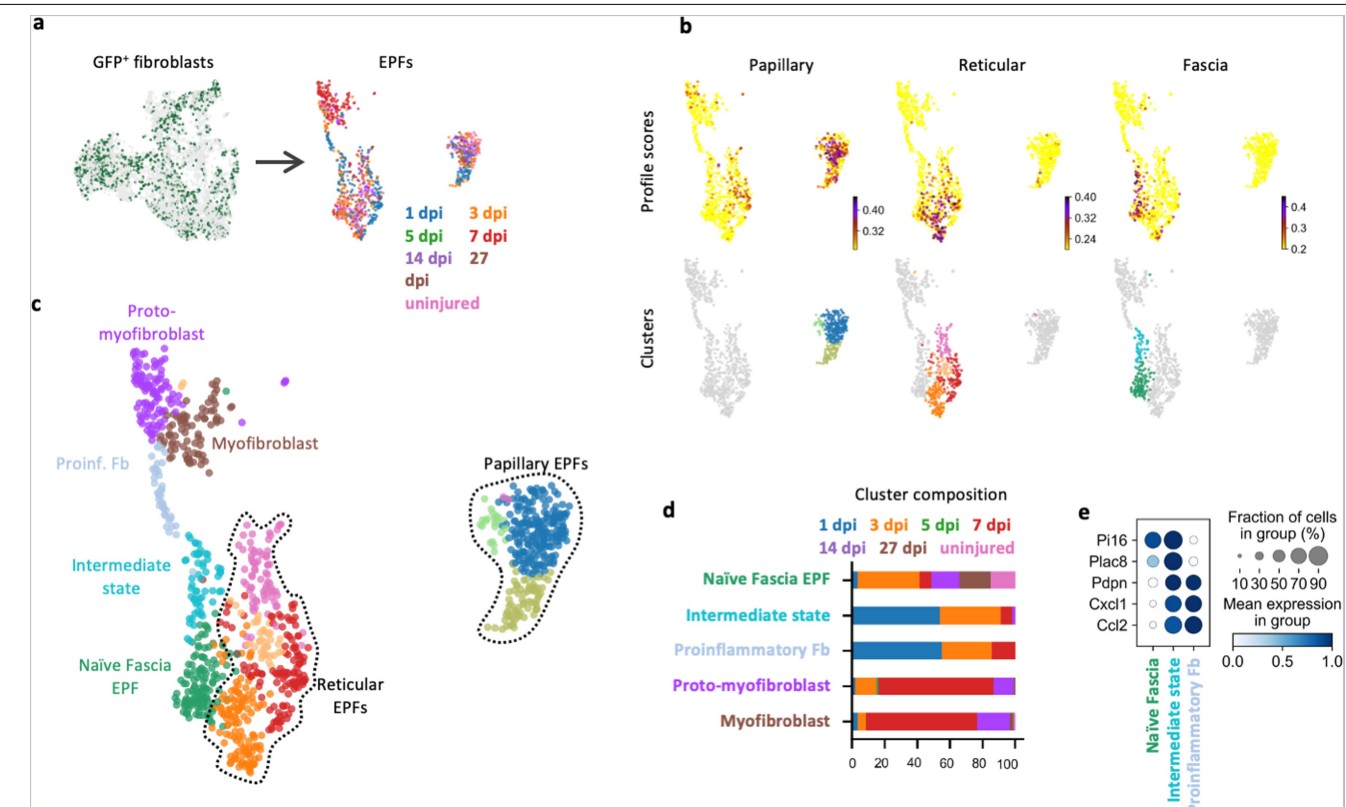

**Extended Data Fig. 2 | Fascia-to-myofibroblast differentiation occurs within the *En1*-lineage. a**. UMAP representation of fibroblast showing GFP expression (EPFs) that were sorted and subclustered. **b**. UMAPs showing the skin fibroblast profile scores (top) and their associated EPF subclusters (bottom). **c**. UMAP of EPFs subclusters. **d**-**e**. An intermediate state with both naïve fascia and proinflammatory profiles supports the transition sprouting from fascia fibroblasts. **d**. Sample composition of EPF subclusters from the fascia-to-myofibroblast trajectory showing that the intermediate state appears in early wounds. **e**. Expression dot-plot of naïve fascia and proinflammatory markers being present in the intermediate state.

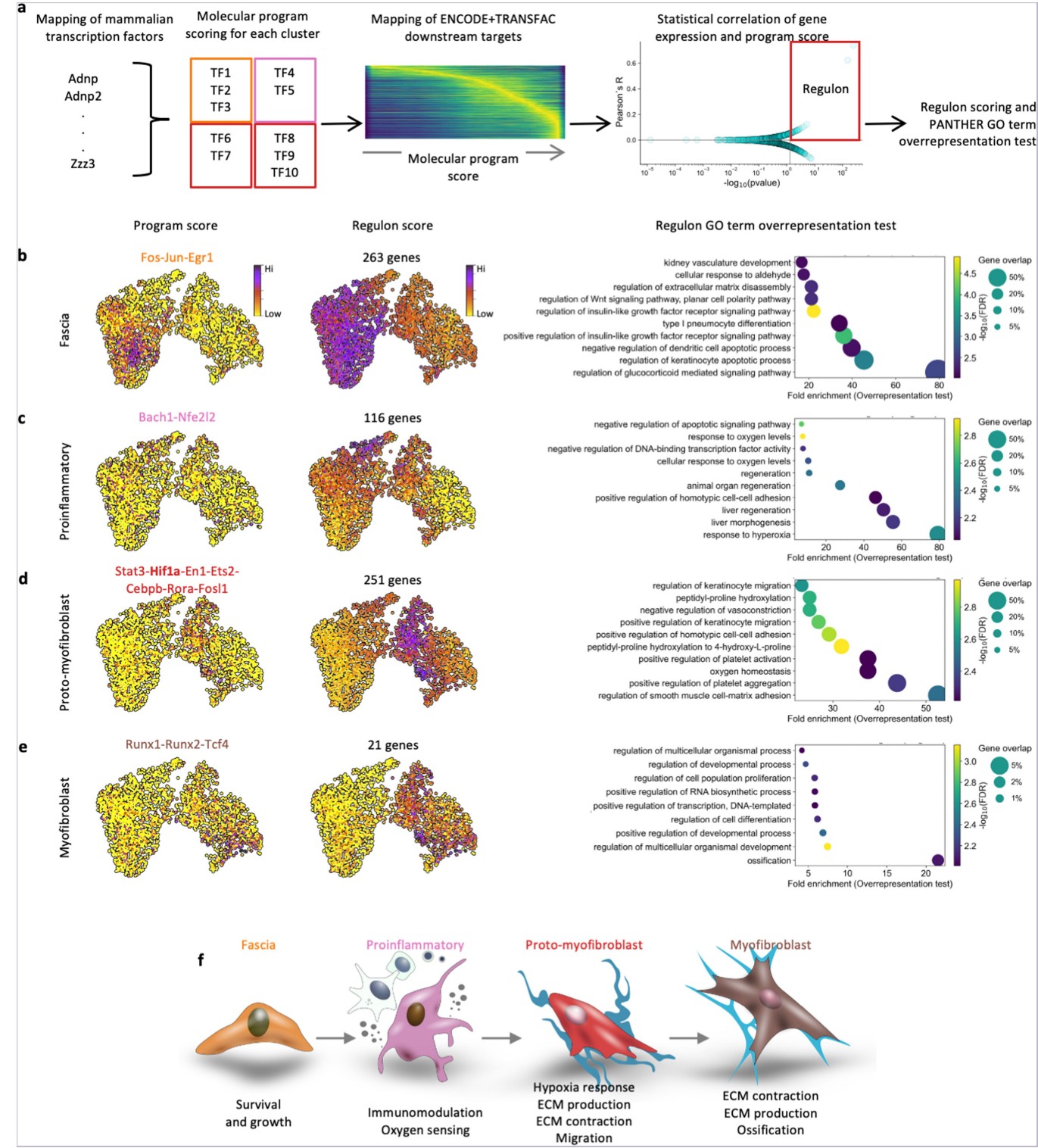

**Extended Data Fig. 3 | State-specific regulons reveal a stepwise differentiation linked to the wound healing phases. a**. Analysis pipeline to determine the collection of transcription factors (programme) and their associated regulons for each fibroblastic state. **b-e**. UMAPs of programme (left) and regulon scores (centre), and their overrepresented GO terms (right) for the fascia (b), proinflammatory (c), proto-myofibroblast (d), and myofibroblast states (e). **f**. Scheme of the fibroblastic states along the trajectory and their phenotype revealed from scRNAseq analysis.

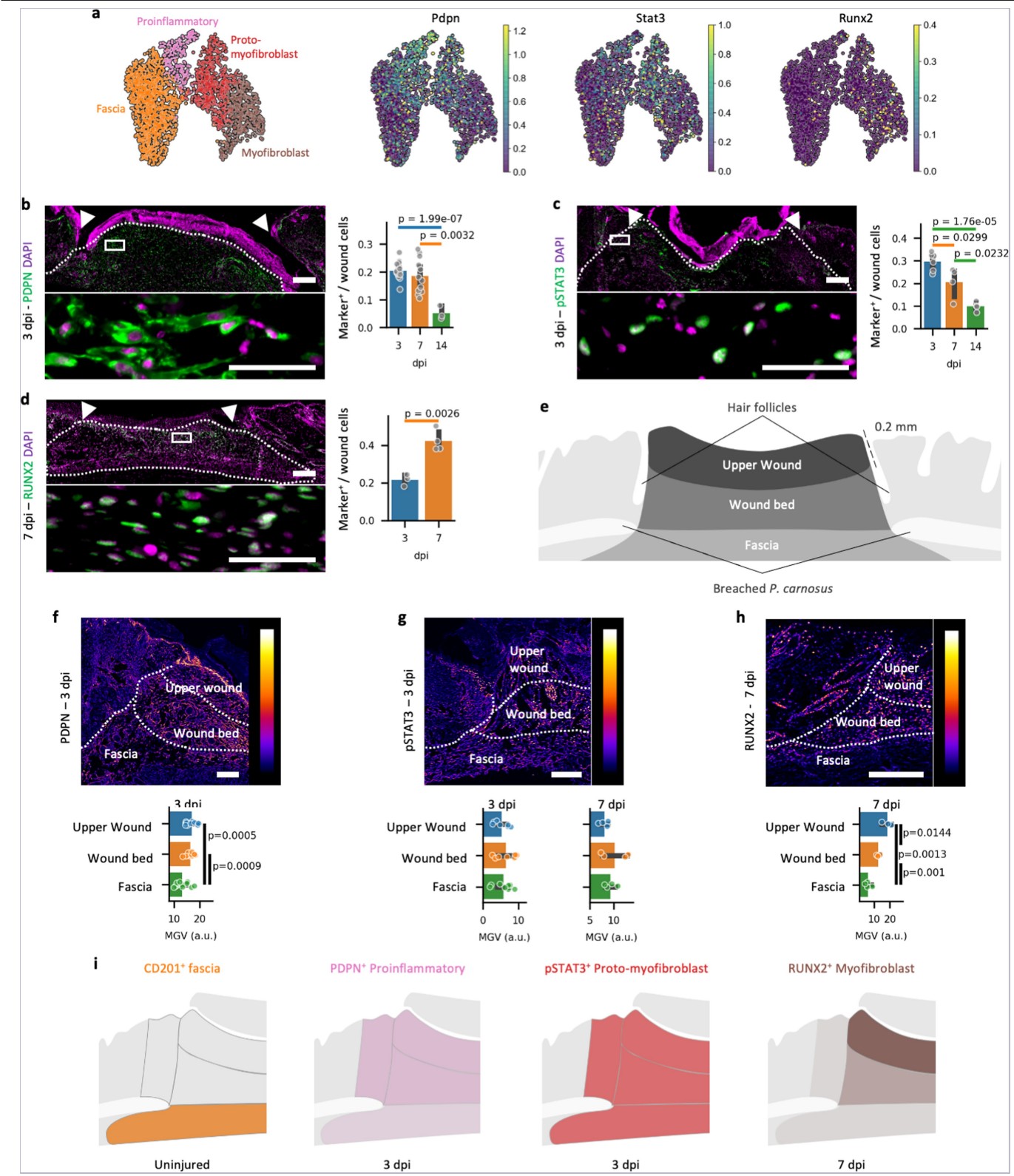

**Extended Data Fig. 4** | See next page for caption.

**Extended Data Fig. 4 | Spatiotemporal patterns during the transition.**
**a**. UMAPs of fibroblast from trajectory, colour-coded for individual clusters (left) and marker genes used in this study (right). **b-d**. Representative immunostainings (left) for PDPN (a), pSTAT3 (b), or RUNX2 (c) from 3- (a-b) or 7-dpi wounds (c) and marker-expressing cell quantifications at different timepoints after injury (right). N = 16 (3 dpi), 12 (7 dpi), and 3 (14 dpi) images from PDPN-stained (a) sections. 6 (3 dpi) and 4 (7 and 14 dpi) images from pSTAT3-stained (b) sections. 3 (3 dpi) and 4 (7 dpi) images from RUNX2-stained (c) sections. All images were obtained from 3 biological replicates for each timepoint. High magnification images (bottom) from rectangle insert in low-power images (top). Dotted line delimits the wound region. Arrowheads indicate the original injury site. **e**. Schematics of the wound regions analysed and the spatial cues that defined them. **f-h**. "Fire" pseudo-colouring representations (top) to depict expression intensity of PDPN (e), pSTAT3 (f), or RUNX2 (g) within the different wound regions at indicated dpi. Mean Gray Value (MGV) expression intensity quantifications (bottom) at indicated dpi in the different wound regions. N = 12 (upper wound and wound core) and 13 (wound fascia) values from PDPN-stained (e) sections. 7 (upper wound and wound core) and 8 (wound fascia) values from pSTAT3-stained (f) sections at 3 dpi. 4 values from pSTAT3-stained (f) sections at 7 dpi. 3 (upper wound) and 4 (wound core and wound fascia) values from RUNX2-stained (g) sections. All images were obtained from 3 biological replicates for each timepoint. Dotted line delimits each wound region. All p-values (p) indicated were obtained from two-tailed T-tests. Scale bars: 500 microns in low magnification (a-c and e-g), and 50 microns in high magnification micrographs (a-c). **i**. Spatial distribution of the fibroblast states in discrete wound regions.

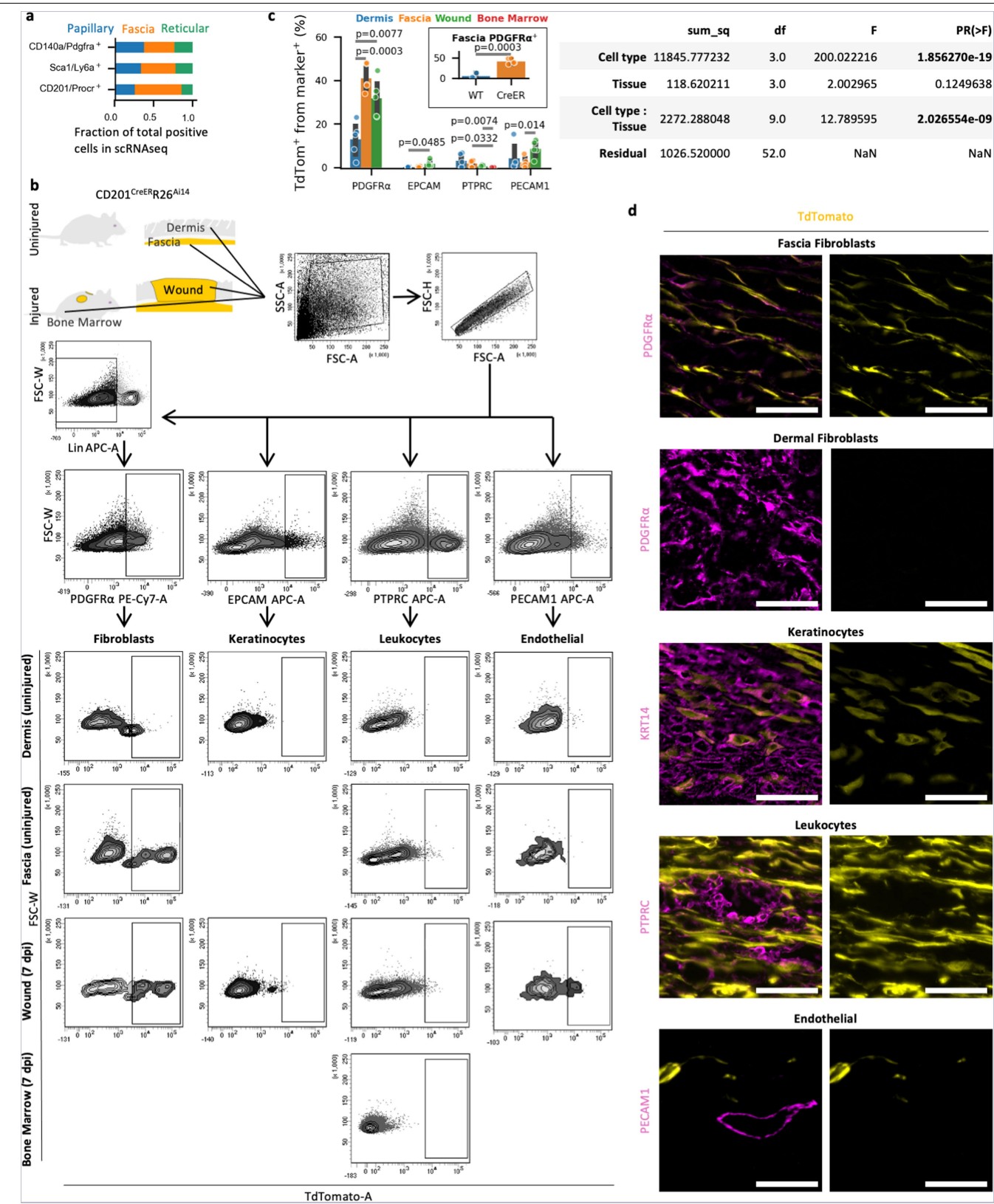

The following table appears within the figure:

| | sum_sq | df | F | PR(>F) |
|---|---|---|---|---|
| Cell type | 11845.777232 | 3.0 | 200.022216 | **1.856270e-19** |
| Tissue | 118.620211 | 3.0 | 2.002965 | 0.1249638 |
| Cell type : Tissue | 2272.288048 | 9.0 | 12.789595 | **2.026554e-09** |
| Residual | 1026.520000 | 52.0 | NaN | NaN |

**Extended Data Fig. 5 | CD201^CreER R26^Ai14 genetic lineage tracing system to fate map fascia fibroblasts. a**. Homeostatic fibroblast fractions in fascia marker-expressing cells within the scRNAseq dataset. **b**. Flow cytometry strategy to identify proportions of different cell types labelled with TdTomato in the uninjured dermis and fascia, as well from wound and bone marrow from 7 dpi CD201^CreER R26^Ai14 mice. **c**. TdTomato-labelled fraction (left) from total fibroblasts (PDGFRα⁺), epithelial (EPCAM⁺), leukocytes (PTPRC⁺), and endothelial cells (PECAM1⁺) present in dermis, fascia, wounds, or bone marrow. Insert: labelled fibroblast fraction in wildtype and CreER positive animals shows minimal leaking of the genetic system. All p-values (p) indicated were obtained from two-tailed T-tests. 2-way ANOVA results from TdTomato-positive fractions comparison (right). N = 5 biological replicates. Representative micrographs immunolabeled for fibroblast (PDGFRα), keratinocytes (KRT14), leukocytes (PTPRC), and endothelial cells (PECAM1) markers on CD201^CreER R26^Ai14 skin sections validating the flow cytometry observations. Scale bars: 50 microns.

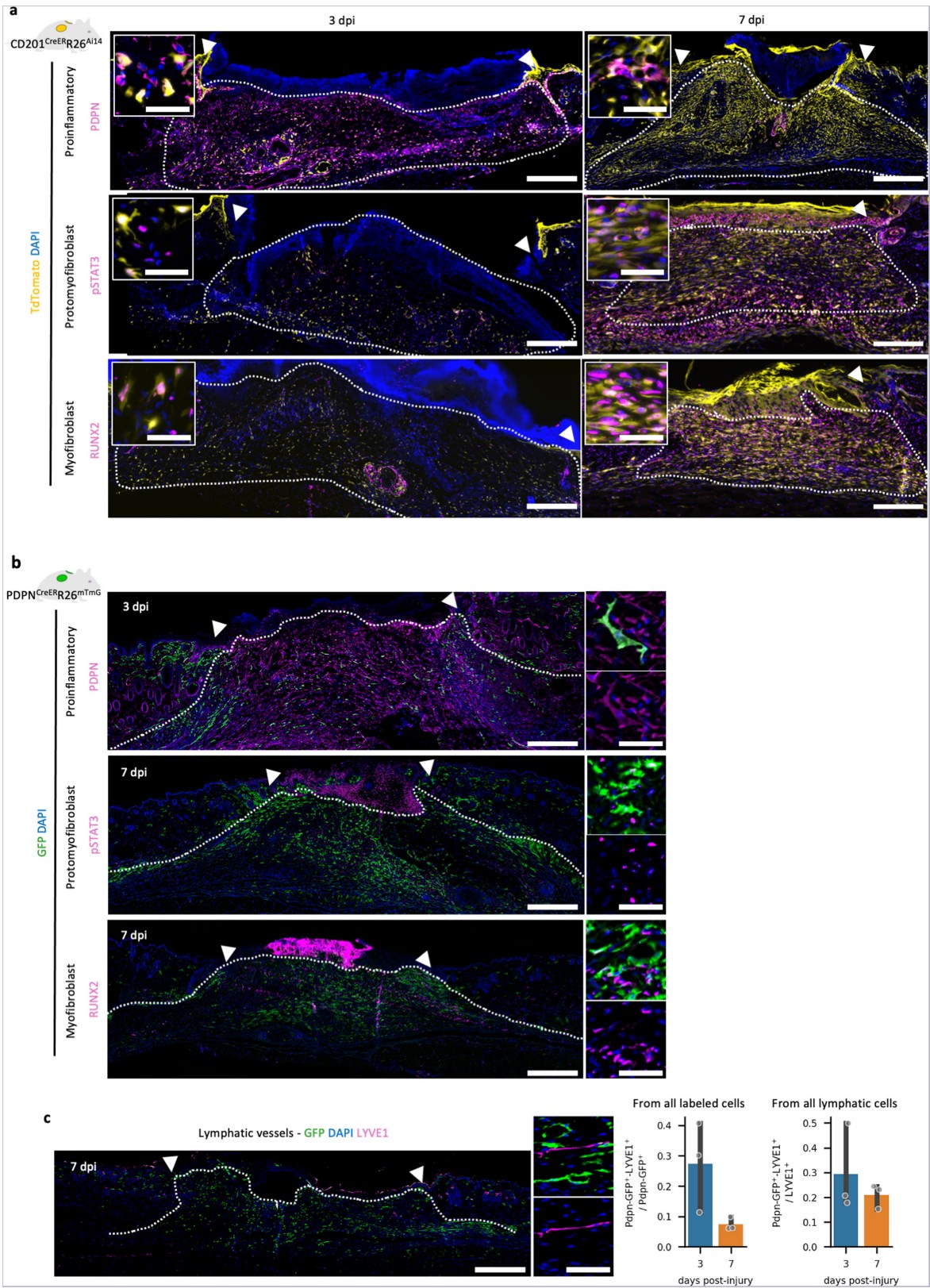

**Extended Data Fig. 6 | Expression of wound fibroblast markers on the CD201^CreER R26^Ai14 and Pdpn^CreER R26^mTmG genetic lineage tracing systems.**
**a**. Representative low and high magnification (inserts) images showing the expression of PDPN (top), pSTAT3 (middle), and RUNX2 (bottom) in TdTomato-labelled fascia-derived cells on 3 (left) and 7 dpi (right) wounds. **b**. Representative low (left) and high magnification (right) images showing the expression of PDPN (top), pSTAT3 (middle), and RUNX2 (bottom) in GFP-labelled proinflammatory-derived cells on wounds at indicated timepoints. **c**. Representative low (left)

and high magnification (middle) images showing the residual labelling with GFP in lymphatic vessels (LYVE1⁺) in wounds at indicated timepoints. Cell fractions of labelled-lymphatic endothelial cells from total traced and from total lymphatic vessels in wounds at indicated timepoints. N = 3 biological replicates. Dotted line delimits the wound region. Arrowheads indicate the original injury site. Scale bars: 500 microns in low magnification, and 50 microns in high magnification micrographs.

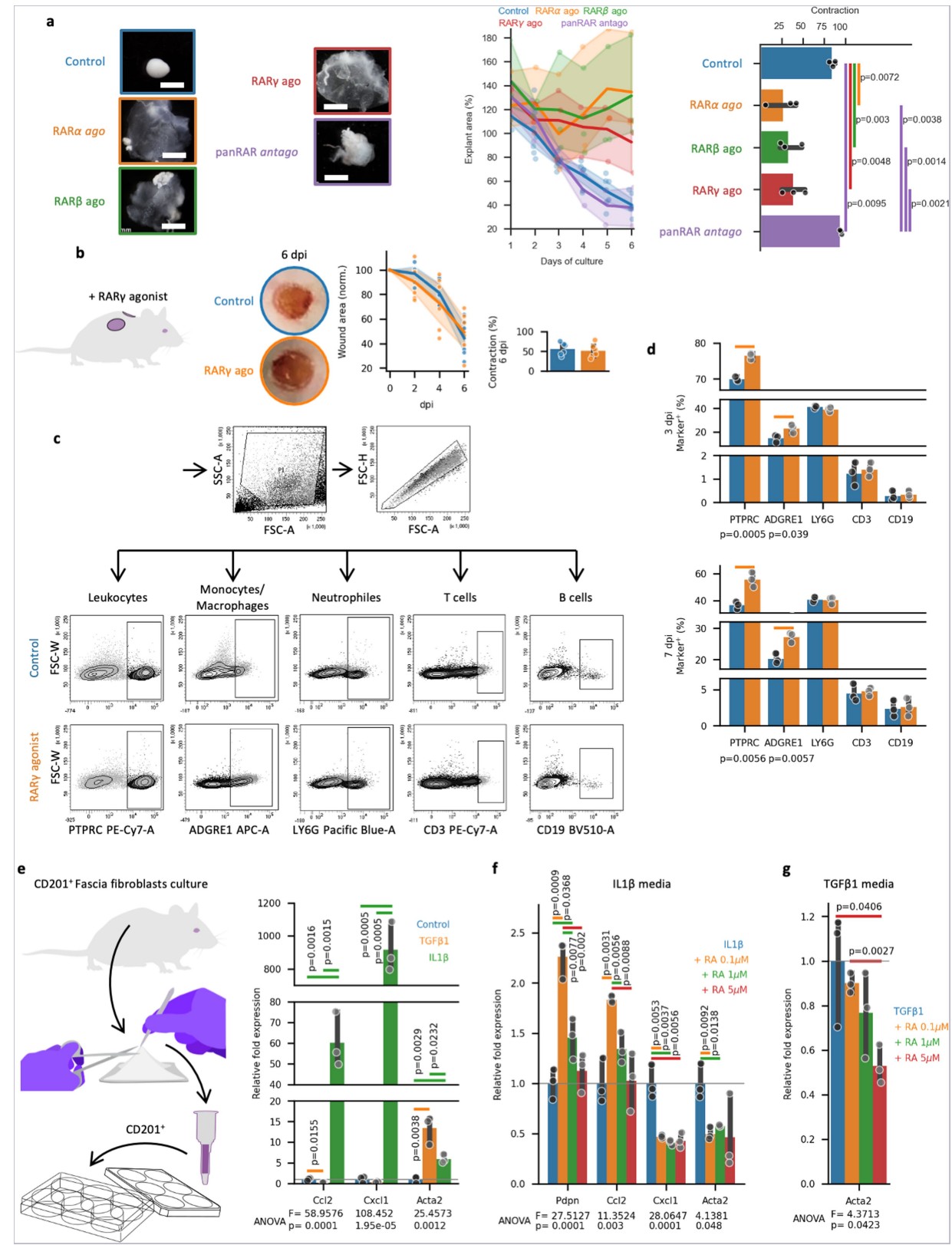

**Extended Data Fig. 7** | See next page for caption.

**Extended Data Fig. 7 | RA supports the proinflammatory state by inducing a monocyte recruitment phenotype. a**. Representative brightfield images at day 6 of culture (left), explant area quantifications over time (middle), and overall contraction quantifications (right) from fascia explants treated with RAR-selective agonists and pan-antagonist. N = 4 (area vs time) and 3 (total contraction) biological replicates. Two-tailed T-tests. Scale bars: 2 mm. **b**. Representative photos of control or RARγ agonist-treated wounds at 6 dpi (left), wound closure measurements (middle), and total wound contraction by 6 dpi (right). N = 6 wounds from 3 biological replicates. **c.** Flow cytometry strategy to identify treatment-induced changes in the recruitment of different immune cell types in wounds. **d**. Total cell type fractions at 3 (top) and 7 dpi (bottom) of general leukocytes (PTPRC$^+$), monocytes/macrophages (ADGRE1$^+$), neutrophiles (LY6G$^+$), T (CD3$^+$), and B lymphocytes (CD19$^+$). N = 3 biological replicates. Two-tailed T-tests. **e.** Strategy for fascia fibroblast purification and culture (left). Expression changes (right) of proinflammatory (*Ccl2* and *Cxcl1*) and myofibroblast markers (*Acta2*) in IL1β- (inflammation-inducing) or TGFβ1-containing media (myofibroblast-inducing). N = 3 technical replicates. Expression changes normalized to control medium. p values on bars from two-tailed T-tests (top) and 1-way ANOVA comparisons between treatments (bottom). **f-g.** Expression changes of indicated markers in inflammation- (f) and myofibroblast-inducing media (g) exposed to exogenous RA at indicated concentrations. N = 3 technical replicates. Expression changes normalized to control IL1β- (f) or TGFβ1-containing medium (g). p values on bars from two-tailed T-tests (top) and 1-way ANOVA comparisons between treatments (bottom).

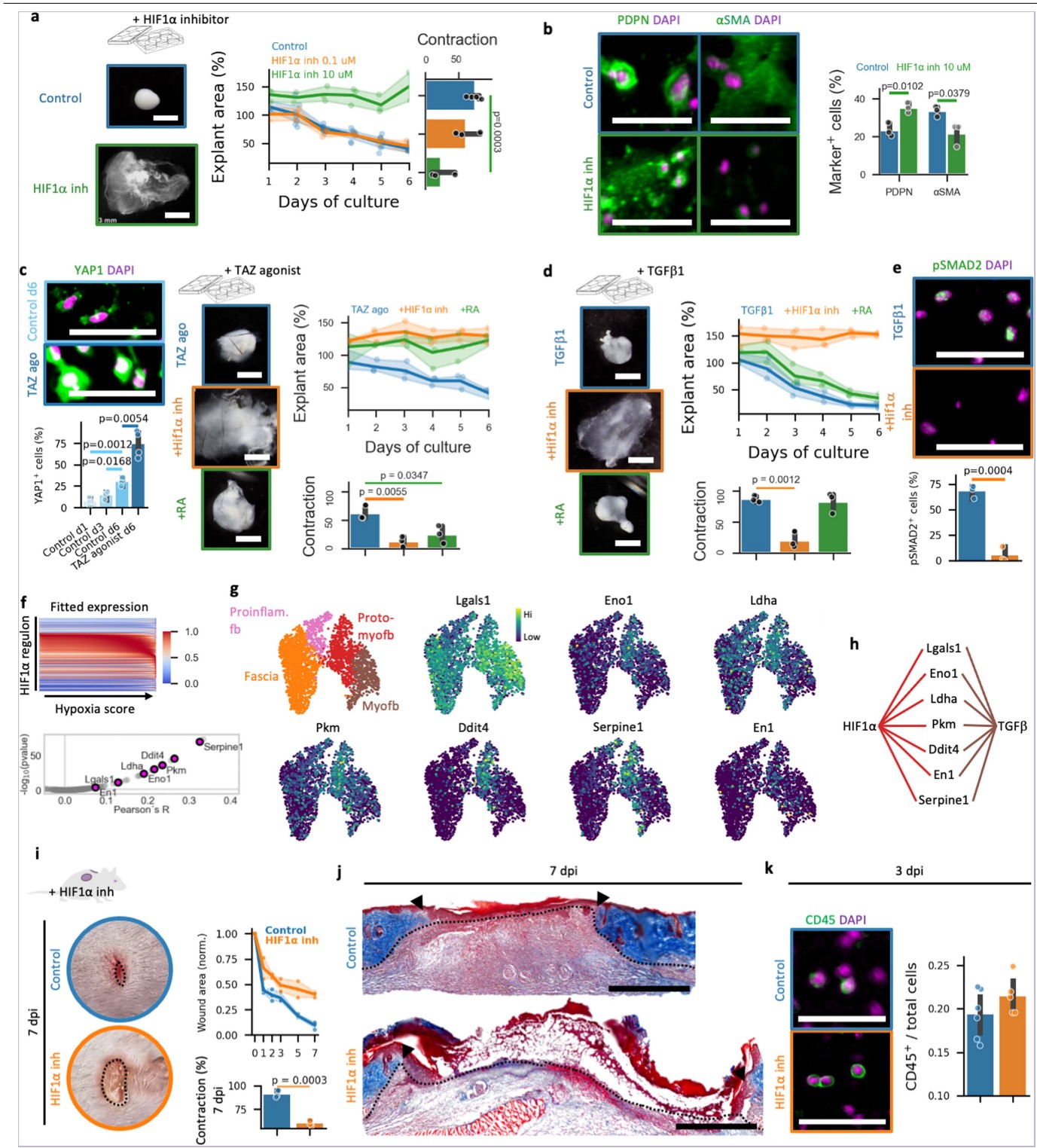

**Extended Data Fig. 8** | See next page for caption.

**Extended Data Fig. 8 | HIF1α instructs the transition into (proto) myofibroblasts. a**. Control (top) or HIF1α inhibitor-treated (bottom) explants after 6 days of culture (left), area over time (middle), and contraction (right) measurements. N = 6 (control) and 3 (inhibitor) replicates. **b**. Immunolabeling (left) and marker-positive cell ratios (right) of proinflammatory fibroblast (PDPN⁺) or myofibroblasts (αSMA⁺) in control (top) or inhibitor-treated (bottom) explants. N = 3 biological replicates. **c**. Immunolabeling (top left) and YAP1⁺ cell ratios (bottom left) upon chemical TAZ agonism. N = 3 (controls) and 4 (agonist) biological replicates. TAZ agonist-treated (centre top) and combined treatments with the inhibitor (centre middle) or RA (centre bottom) explants after 6 days of culture. Area over time (top right), and contraction (bottom right) measurements. N = 3 biological replicates. **d**. TGFβ1-treated (top left) and combined treatments with the inhibitor (centre left) or RA (bottom left) explants after 6 days of culture. Area over time (top right), and contraction (bottom right) measurements. N = 3 biological replicates. **e**. Immunolabeling (top) and pSMAD2⁺ cell ratios (bottom). N = 3 biological replicates. **f**. Correlation of HIF1α regulon vs hypoxia activity, positively regulated genes involved in TGFβ pathway (bottom). **g-h**. UMAPs of gene expression and schematic representation of HIF1α-regulated TGFβ modulators. **i**. Representative photographs (left) of control (top) or inhibitor-treated wounds (bottom) at 7 dpi. Wound area measurements over time (top right) and overall contraction at 7 dpi (bottom right). N = 3 biological replicates. **j**. Trichrome-stained micrographs of control (top) and treated wounds (bottom) at 7 dpi. **k**. Micrographs of control (top left) and treated 3 dpi wounds (bottom left) immunolabeled for CD45/PTPRC and positive cell ratios (right). N = 6 (control) and 5 (treated) wounds from 3 biological replicates. All p-values from two-tailed T-tests. Scale bars: 2 mm in explants photographs (a, c-d), 500 microns (g), and 50 microns (b-c, e, and h).

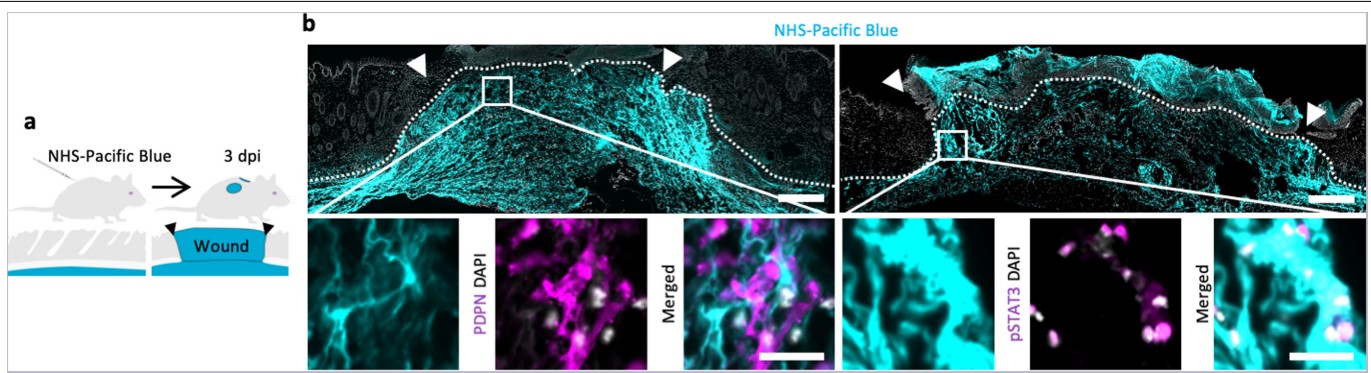

**Extended Data Fig. 9 | Fate mapping of fascia extracellular matrix.**
**a**. Schematics of fascia-matrix labelling using NHS-Pacific Blue. **b**. Representative low magnification (top) and high magnification (bottom) images of control wounds immunolabeled for PDPN or pSTAT3, showing association with traced extracellular matrix. Scale bars: 500 microns in low magnification and 50 microns in high magnification micrographs.

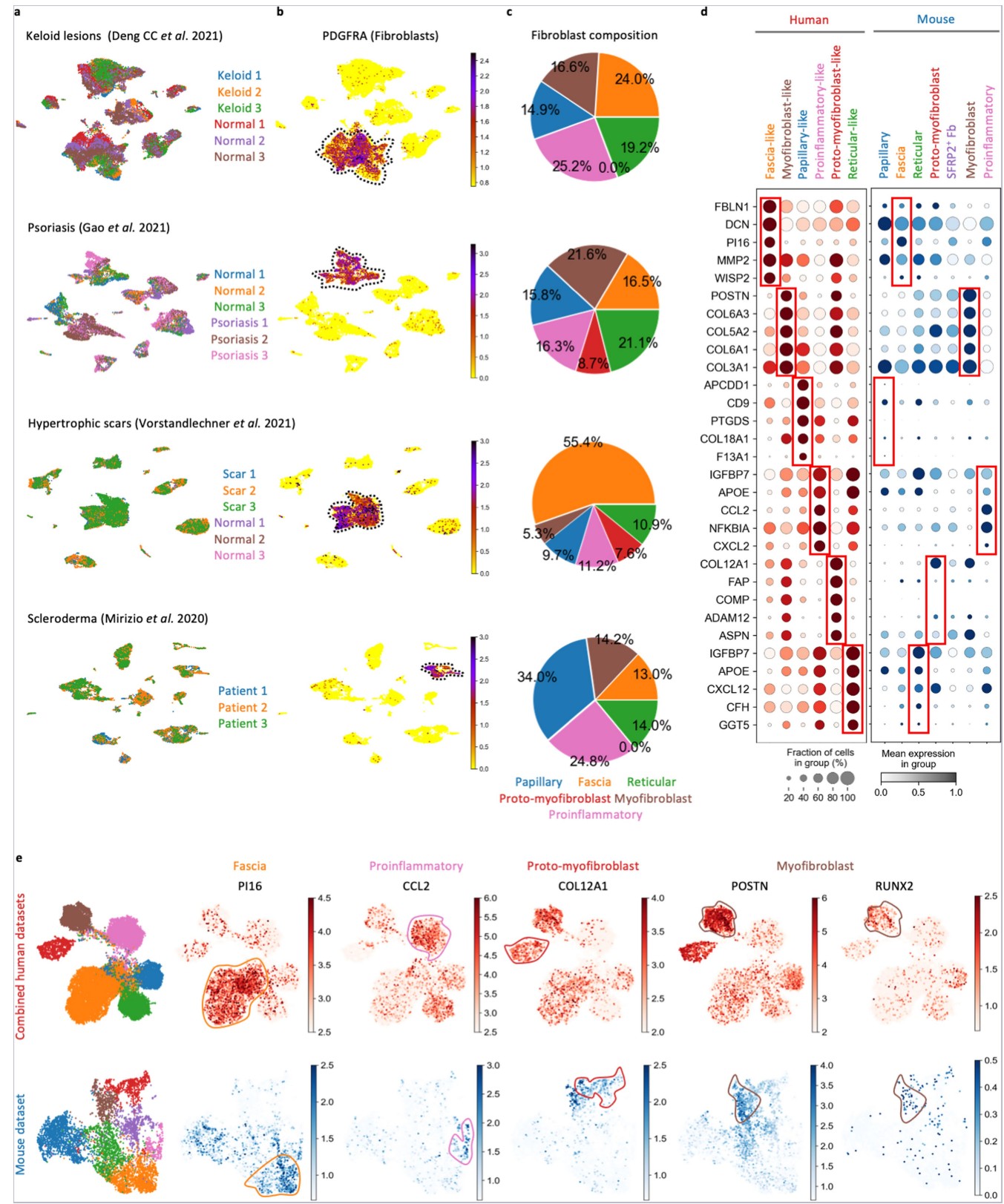

**Extended Data Fig. 10 | Comparative analysis of human and mouse fibroblasts subtypes. a-b**. UMAP representation of scRNA-seq datasets of denoted human skin pathologies color-coded for sample of origin (a) and the pan-fibroblast marker *PDGFRA* expression levels (b). **c**. Composition charts depicting the proportion of the different fibroblast subtypes for each dataset.

**d**. Expression comparison between merged human (red) and mouse datasets (blue) of top marker genes for the human fibroblast clusters. Expression of analogous fibroblast clusters is denoted in red rectangles. **e**. UMAP representations the expression levels of highly conserved markers in the merged human (top) and mouse (bottom) datasets.

| | |
|---|---|

# Reporting Summary

## Statistics

For all statistical analyses, confirm that the following items are present in the figure legend, table legend, main text, or Methods section.

| n/a | Confirmed | |
|---|---|---|
| ☐ | ☒ | The exact sample size (*n*) for each experimental group/condition, given as a discrete number and unit of measurement |
| ☐ | ☒ | A statement on whether measurements were taken from distinct samples or whether the same sample was measured repeatedly |
| ☐ | ☒ | The statistical test(s) used AND whether they are one- or two-sided<br>*Only common tests should be described solely by name; describe more complex techniques in the Methods section.* |
| ☒ | ☐ | A description of all covariates tested |
| ☐ | ☒ | A description of any assumptions or corrections, such as tests of normality and adjustment for multiple comparisons |
| ☐ | ☒ | A full description of the statistical parameters including central tendency (e.g. means) or other basic estimates (e.g. regression coefficient) AND variation (e.g. standard deviation) or associated estimates of uncertainty (e.g. confidence intervals) |
| ☐ | ☒ | For null hypothesis testing, the test statistic (e.g. *F*, *t*, *r*) with confidence intervals, effect sizes, degrees of freedom and *P* value noted<br>*Give P values as exact values whenever suitable.* |
| ☒ | ☐ | For Bayesian analysis, information on the choice of priors and Markov chain Monte Carlo settings |
| ☒ | ☐ | For hierarchical and complex designs, identification of the appropriate level for tests and full reporting of outcomes |
| ☐ | ☒ | Estimates of effect sizes (e.g. Cohen's *d*, Pearson's *r*), indicating how they were calculated |

*Our web collection on statistics for biologists contains articles on many of the points above.*

## Software and code

Policy information about availability of computer code

| Data collection | FACS: BD FACS Diva (BD Bioscience 6.4.1)<br>Microscopy: LAS X (Leica) and ZEN black (Carl Zeiss) |
|---|---|
| Data analysis | Image analysis: Fiji (v1.53c)<br>FACS: BD FACS Diva (BD Bioscience 6.4.1)<br>Statistics and plotting: python (3.8), scipy (1.5.3), pandas (1.3.5), numpy (1.23.5), matplotlib (3.7.1), and seaborn (0.12.2).<br>scRNAseq: scanpy (1.9.3), decoupler (PROGENY and DOROTHEA 1.4.0), scArches (0.5.9), scvelo (0.2.5), PANTHERdb (14.1) |

For manuscripts utilizing custom algorithms or software that are central to the research but not yet described in published literature, software must be made available to editors and reviewers. We strongly encourage code deposition in a community repository (e.g. GitHub). See the Nature Portfolio guidelines for submitting code & software for further information.

## Data

Policy information about availability of data

All manuscripts must include a data availability statement. This statement should provide the following information, where applicable:

- Accession codes, unique identifiers, or web links for publicly available datasets
- A description of any restrictions on data availability
- For clinical datasets or third party data, please ensure that the statement adheres to our policy

The generated scRNAseq data has been deposited in the Gene Expression Omnibus under the accession number (GEO ID to be confirmed). All other data that support the findings of this study are available from the corresponding author upon reasonable request.

## Research involving human participants, their data, or biological material

Policy information about studies with human participants or human data. See also policy information about sex, gender (identity/presentation), and sexual orientation and race, ethnicity and racism.

| | |
|---|---|
| Reporting on sex and gender | *Use the terms sex (biological attribute) and gender (shaped by social and cultural circumstances) carefully in order to avoid confusing both terms. Indicate if findings apply to only one sex or gender; describe whether sex and gender were considered in study design; whether sex and/or gender was determined based on self-reporting or assigned and methods used.* *Provide in the source data disaggregated sex and gender data, where this information has been collected, and if consent has been obtained for sharing of individual-level data; provide overall numbers in this Reporting Summary. Please state if this information has not been collected.* *Report sex- and gender-based analyses where performed, justify reasons for lack of sex- and gender-based analysis.* |
| Reporting on race, ethnicity, or other socially relevant groupings | *Please specify the socially constructed or socially relevant categorization variable(s) used in your manuscript and explain why they were used. Please note that such variables should not be used as proxies for other socially constructed/relevant variables (for example, race or ethnicity should not be used as a proxy for socioeconomic status).* *Provide clear definitions of the relevant terms used, how they were provided (by the participants/respondents, the researchers, or third parties), and the method(s) used to classify people into the different categories (e.g. self-report, census or administrative data, social media data, etc.)* *Please provide details about how you controlled for confounding variables in your analyses.* |
| Population characteristics | *Describe the covariate-relevant population characteristics of the human research participants (e.g. age, genotypic information, past and current diagnosis and treatment categories). If you filled out the behavioural & social sciences study design questions and have nothing to add here, write "See above."* |
| Recruitment | *Describe how participants were recruited. Outline any potential self-selection bias or other biases that may be present and how these are likely to impact results.* |
| Ethics oversight | *Identify the organization(s) that approved the study protocol.* |

Note that full information on the approval of the study protocol must also be provided in the manuscript.

# Field-specific reporting

Please select the one below that is the best fit for your research. If you are not sure, read the appropriate sections before making your selection.

☒ Life sciences          ☐ Behavioural & social sciences          ☐ Ecological, evolutionary & environmental sciences

For a reference copy of the document with all sections, see nature.com/documents/nr-reporting-summary-flat.pdf

# Life sciences study design

All studies must disclose on these points even when the disclosure is negative.

| | |
|---|---|
| Sample size | Required experimental sample sizes were estimated based on previous established protocols in the field. The sample sizes were adequate as the differences between experimental groups were reproducible. All n values are clearly indicated within the figure legends. |
| Data exclusions | No data was excluded from the analysis. |
| Replication | All animal experiments were performed by at least two independent researchers showing similar results. Ex vivo and in vitro experiments were replicated at least three times. |
| Randomization | Age- and weight-matched animals were randomly divided into treatment groups. Experiments that included both male and female mice were divided as to have same number of animals of the same sex in each group. |
| Blinding | No experiments presented in this study required blinding. |

# Reporting for specific materials, systems and methods

We require information from authors about some types of materials, experimental systems and methods used in many studies. Here, indicate whether each material, system or method listed is relevant to your study. If you are not sure if a list item applies to your research, read the appropriate section before selecting a response.

## Materials & experimental systems

| n/a | Involved in the study |
|---|---|
| ☐ | ☒ Antibodies |
| ☒ | ☐ Eukaryotic cell lines |
| ☒ | ☐ Palaeontology and archaeology |
| ☐ | ☒ Animals and other organisms |
| ☒ | ☐ Clinical data |
| ☒ | ☐ Dual use research of concern |
| ☒ | ☐ Plants |

## Methods

| n/a | Involved in the study |
|---|---|
| ☒ | ☐ ChIP-seq |
| ☐ | ☒ Flow cytometry |
| ☒ | ☐ MRI-based neuroimaging |

## Antibodies

| | |
|---|---|
| Antibodies used | Histology:<br>anti-PDPN (Abcam ab11936, 1:500 dilution)<br>anti-pSTAT3 (Cell Signaling Technology 9145S, 1:150)<br>anti-RUNX2 (Abcam ab92336, 1:150)<br>anti-GFP (Abcam ab13970, 1:500)<br>anti-PDGFRα (R&D systems AF1062, 1:100)<br>anti-KRT14 (Abcam ab181595, 1:100)<br>anti-PECAM1/CD31 (Abcam ab56299, 1:50)<br>anti-LYVE1 (Abcam ab14917, 1:100)<br>anti-αSMA (Abcam ab21027, 1:150)<br>anti-YAP1 (Abcam ab205270, 1:100)<br>anti-pSMAD2 (Cell Signalling 18338, 1:100)<br>anti-HIF1A (Novus NB100-479, 1:100)<br>anti-CCL2 (Abcam ab25124, 1:100)<br>anti-CXCL1 (R&D systems MAB453R, 1:100)<br>anti-ALDH1A3 (Novus NBP2-15339, 1:100)<br>anti-CYP26B1 (Elabscience E-AB-36196, 1:100)<br>anti-CD45/PTPRC (Abcam ab23910, 1:100)<br>anti-PI16 (R&D systems AF4929, 1:100)<br>FACS:<br>All antibodies were used in a 1:200 dilution except for the CD45 antibody, which was diluted 1:800.<br>anti-CD45(PTPRC)-APC and anti-CD45-PE/Cy7 (30-F11, Bio legend)<br>anti-CD31(PECAM1)-APC (390, e-Biosciences)<br>anti-TER119-APC (Ter119, Bio legend)<br>anti-CD326(EPCAM)-AF647 (G8.8, Bio legend)<br>anti-CD11b-AF488 (M1/70, Bio legend)<br>anti-LY6G-PacBlue (1A8, Bio legend)<br>anti-F4/80(ADGRE1)-APC (BM8, Bio legend)<br>anti-CD3-PE/Cy7 (500A2, Bio legend)<br>anti-CD19-BV510 (6D5, Bio legend)<br>anti-CD140a(PDGFRA)-PE-Cy7 (APA5, e-Biosciences). |
| Validation | See manufacturers' notes. Antibodies were additionally validated using respective isotype antibodies in immunofluorescence assays. |

## Animals and other research organisms

Policy information about studies involving animals; ARRIVE guidelines recommended for reporting animal research, and Sex and Gender in Research

| | |
|---|---|
| Laboratory animals | Animal experiments were performed using 8- to 12-week-old adult mice.<br>C57BL/6J wildtype<br>En1tm2(cre)Wrst/J (En1Cre)<br>B6.Cg-Gt(ROSA)26Sortm14(CAG-tdTomato)Hze/J (R26Ai14)<br>B6.129(Cg)-Gt(ROSA)26Sortm4(ACTB-tdTomato.-EGFP)Luo/J (R26mTmG)<br>B6.129S6(Cg)-Gt(ROSA)26Sortm1(DTA)Jpmb/J (R26DTA)<br>PDPNCreER<br>CD201/ProcrCreER (Dr. Ariel Zeng, SIBCB) |

| | B6.129-Hif1atm3Rsjo/J (Hif1aflox) |
|---|---|
| Wild animals | The study did not involve wild animals. |
| Reporting on sex | Male and female mice, equally distributed between time groups, were used for descriptive experiments. For functional studies, littermates or age-matched animals were randomly assigned into the different experimental groups. To decrease variability, only males were used for the RARγ agonist and Hif1α inhibition treatments (7 dpi), and for the Hif1α genetic deletion experiment. Females and males, equally distributed between treatment groups, were used for the PDPN+ cell ablation studies and for the 3 dpi Hif1α inhibition experiment. |
| Field-collected samples | The study did not involve samples collected from field. |
| Ethics oversight | Government of Upper Bavaria , Germany |

Note that full information on the approval of the study protocol must also be provided in the manuscript.

# Flow Cytometry

## Plots

Confirm that:

☒ The axis labels state the marker and fluorochrome used (e.g. CD4-FITC).

☒ The axis scales are clearly visible. Include numbers along axes only for bottom left plot of group (a 'group' is an analysis of identical markers).

☒ All plots are contour plots with outliers or pseudocolor plots.

☒ A numerical value for number of cells or percentage (with statistics) is provided.

## Methodology

| | |
|---|---|
| Sample preparation | Skin and wounds were collected using 8 mm biopsy punch (Stiefel 10008) and fascia and dermis were separated as stated in the manuscript. Tissue was defatted, finely minced with scissors, and incubated in digestion solution (0.2 mg-1mg/ mL liberase, 0.5 mg/mL collagenase-A, 100 U/mL DNase in serum-free DMEM) for 60 minutes at 37℃ while rocking at 350 rpm. Digestion was stopped with 10 mL DMEM, samples were vortexed for few seconds, and then strained through 70 μm sieves. For the isolation of bone marrow cells, tibiae and femur of mice were dissected, flushed using a syringe with a G21 needle, and then filtered through 40μm filter to remove tissue debris. Single-cell suspensions were fixed with 2 % PFA to preserve surface markers stability for long periods. Single-cell suspensions were pelleted and suspended in PBS containing fluorophore-conjugated antibodies for 30 min before analysis. |
| Instrument | All samples were run on a BD LSRII cytometer (BD Biosciences) equipped with violet, blue, and red lasers and analysed in the BD FACS Diva analyser suite (BD Biosciences). |
| Software | BD FACS Diva (BD Bioscience 6.4.1) |
| Cell population abundance | The purity of sorted cells were determined by flow cytometric analysis of the sorted cells with the same gating strategy as during sorting |
| Gating strategy | For all cell types, initial gate from forward scatter (FSC-A) vs. side-scatter (SSC-A) plots was used. From this, single-cell gate from FSC-A vs FSC-H plots was used to exclude debris.  Strict doublet exclusion was performed prior gating for immune cells (CD45+) and stromal cells (Lin-: CD45- CD31- Ter119- EpCAM-). Monocytes and macrophage cells were gated as ADGRE1+, Neutrophils as LY6G+, T-cells as CD3+, and B-cells as CD19+. Similarly, TdTomato+ cells from CD201CreERR26Ai14 mice were gated from Lin- cells then PDGFRA+, and from CD31+, CD45+, and EPCAM+ cells. Signal compensation was performed on FMO controls using compensation beads (Thermo Fisher 01-3333-42). |

☒ Tick this box to confirm that a figure exemplifying the gating strategy is provided in the Supplementary Information.

