## [Peer Review File · Nature]

Manuscript Title: CD201+ 1 fascia progenitors choreograph injury repair

Reviewer Comments & Author Rebuttals

Reviewer Reports on the Initial Version:

Referees' comments:

Referee #1 (Remarks to the Author):

Correa-Gallegos identified CD201+ cells (mainly cells from the fascia) as progenitors of wound fibroblasts, and they show their transition into pro-inflammatory fibroblasts and then into myofibroblasts. They further identify retinoic acid (RA) and HIF-1a signaling as key regulators of the different transition steps.

This is an interesting wound healing study with a large number of scRNA-seq and fate mapping data, which provide further insight into the roles of different fibroblast subpopulations in wound healing. It is a nice extension of the previous work from the authors, which identified an important role of fascia fibroblasts in the healing of deep wounds. While some results were predictable based on previous work of the authors and many others (incompletely cited - see below), the identification of CD201+ cells as progenitors of different wound fibroblasts is novel. However, it is not clear if all CD201+ cells are really fascia fibroblasts (see below). The role of RA and HIF-1a in the transition stages is also novel and interesting, but there are some technical limitations (see below), and other factors may also be important for this transition. Some of the results are overinterpreted.

Major points:

1. Page 6, end of first paragraph: CD201 had previously been shown to be expressed by mesenchymal cells/fibroblasts. For example, CD201+ cells were found to be important for spleen regeneration (Tan and Watanabe, 2017), and PDGFRa+/CD201+ cells are mesenchymal progenitors in the skeletal muscle (Uezumi et al., 2016). These papers should be cited – otherwise one gets the impression that expression of CD201 in stromal cells and a role of CD201+ stromal cells as progenitors is entirely novel (which is not the case).
2. The data shown in Fig. S1h-l are highly relevant and should be moved to the main figure. The legend should include more information (e.g. based on x sequenced cells from y wounds and z mice). These data indeed suggest that fascia fibroblasts can give rise to the other fibroblast subpopulations, but it is not functionally proven that “they present a single lineage that emerges exclusively from homeostatic fascia fibroblasts”. This should be expressed more carefully.
3. Fig. S3 and other figures showing wound healing data: Please mention the number of wounds that were analyzed and the number of mice. It seems that some of the data were only based on 3 wounds. Given the high variability of the wound healing process, this is not sufficient. Please also add magnification bars to all wounds – they are missing in all photomicrographs.. T-test is not suitable for in vivo samples, which usually do not show a Gaussian distribution.

4. The TAT-Cre experiment is elegant. However, it is not convincingly demonstrated that only fascia fibroblasts were targets. I can see green cells in the dermis of non-injured skin (Fig. S4b).. The high magnification picture (again no magnification provided) shows only the fascia and not the dermis – please also show a high magnification of the dermis. The specificity of the deletion is crucial for the interpretation of the data. It is also not clear which percentage of fascia fibroblasts was targeted by TAT-Cre. It is also possible that the TAT-Cre protein remains in the skin for some time and then deletes in many wound fibroblasts (and other cells) upon injury.

5. The 1.7-fold enrichment of CD201 in fascia fibroblasts is good, but CD201 is obviously not specific for fascia fibroblasts. Fig. 1d suggests that TdTomato is indeed mainly expressed by fascia fibroblasts. However, there seem to be a few positive cells in the dermis (Fig. 1e and S5b). It is still possible that a few dermal cells that are TdTomato positive contribute to the pool of positive cells in 7-day wounds. Are there still positive cells in the skin when you remove the fascia? CD201 is also expressed by hematopoietic stem/progenitor cells, which are known to contribute to wound healing. Some of the positive cells in the granulation tissue could come from the haematopoietic system (could be tested using bone marrow chimeric mice).

6. Page 11: The transcription factor analysis is interesting, but it should be validated at the protein level by staining for the active forms of the relevant transcription factors. For example, is there indeed more active Nfe2l2 in pro-inflammatory fibroblasts (or more of its antagonist Bach1)? And is there more HIF1a in the nucleus of proto-myofibroblasts and more Tcf4 in the myofibroblasts? Without such a validation these results are not very meaningful.

7. The use of ex vivo fascia cultures is an interesting approach. However, it has clear limitations. First, the influence of other cells (in particular immune cells) and of endocrine factors cannot be studied. Most importantly, the fascia cultures were maintained in medium with 10% FCS, which includes a lot of TGF- β (and other growth factors and hormones). Therefore, it is not surprising that the tissue contracts, because TGF- β promotes the contractility of fibroblasts. Serum-free conditions (or at least low serum) would be more relevant. Do the cultures indeed express pro-inflammatory cytokines at day 1-3 and other myofibroblast markers at day 6? What happens to CD201 expression in these cultures? The results of Fig. 1k-l certainly do not allow the conclusion that “the classic phases of wound healing... are directly regulated by the differentiation of CD201+ progenitors”.

8. Fig. 2g shows contraction, but not myofibroblast differentiation – please stain for α -SMA. Is there a reduction in α -SMA and/or reduced stress fiber formation?

9. In the fascia explant experiment, RA was added at pharmacological concentrations (1 μ M, this is a very high concentration that is not within the physiological range..) – this experiment does not allow a conclusion about the role of endogenous RA. There is retinol in serum, which is converted to RA in fibroblasts and other cells. Therefore, it would be more important to determine if culture in retinol-free medium promotes myofibroblast differentiation. This experiment would also be important to support the statement that “RA signaling downregulation grants the exit and progression into myofibroblasts” (page 14, first paragraph), for which no functional evidence is provided so far. Overall, the RA experiments are interesting, but they do not clarify if endogenous RA is really the

driving factor for the differentiation of CD201+ fibroblasts into pro-inflammatory fibroblasts.

10. Did the authors observe any toxicity of the RA at higher concentration? How about toxicity of any of the compounds used for the experiments in Fig. S7?

11. The in vivo wound healing experiments with the RARg activator are important. However, the authors do not show that there is prolonged wound inflammation in the treated mice. They only show more CD45+ cells at day 3. This could also be the result of a delayed influx. To make this claim, a time course experiment would be required. In addition, the different immune cells have very different functions in wound healing – which immune cells are affected by the activator?

12. How did the authors determine the relevant dose of the RARg activator for the wound healing studies and how many injections were done around the wound? Did the authors consider a topical treatment? What is the effect of the RARg activator on the speed of wound healing? The reduced scarring may also be a consequence of faster re-epithelialisation. Enhanced healing would be consistent with previous studies showing a healing promoting activity of RA (e.g. Lateef et al., 2005). Please cite previous publications that demonstrated effects of RA on wound healing.

13. The anti-fibrotic effect of RA is also not novel. The authors cite papers that show this effect in other organs, but skin data are not mentioned. For example, Okarinen et al., already showed in 1985 that RA inhibits collagen production in skin fibroblasts and it has been suggested as a treatment for systemic sclerosis (RM Thomas et al., 2016). Please cite relevant papers about RA effects on skin fibrosis.

14. The effect of depletion of pro-inflammatory fibroblasts on wound healing is not surprising – in these experiments, a large percentage of the granulation tissue fibroblasts was depleted and therefore, an important source of ECM proteins and growth factors was removed. It also seems likely that the massive cell death that occurs in these mice upon DT treatment causes strong inflammation, which impairs the healing process. In the opinion of this reviewer, this experiment does not confirm that podoplanin-positive fibroblasts are the progenitors of myofibroblasts as stated on page 15.

15. The HIF-1 inhibitor experiments are interesting, but it is not shown that HIF1a protein levels are indeed higher in podoplanin-positive wound fibroblasts compared to other fibroblasts. In addition, one would like to know if there is HIF1a in the nucleus of the fibroblasts from ex vivo fascia cultures. Does ex vivo culture of fascia under hypoxic conditions affect myofibroblast differentiation?

16. The authors show that wound healing is impaired upon treatment with the HIF1 inhibitor. This is nicely demonstrated, but not at all surprising. HIF1a activity is required in many cell types in the wound, including macrophages, and the important role of HIF1a in wound healing has previously been demonstrated (see for example the review of WX Hong et al., 2014). There is even a study where HIF1a was knocked out in fibroblasts – the authors observed impaired wound angiogenesis and impaired wound closure in these mice (Duscher et al., 2015). Knockout of HIF1a in other cell types also affected the wound healing process. Unfortunately, none of these studies has been cited, which gives the impression that the role of HIF1a described in the present study is novel

(which is not the case). The most interesting experiment is the one shown in Fig. 3g, but it is not clear when the treatment with the inhibitor was started. Ideally, it should be started at around day 3-5 when inflammatory fibroblasts are present, but myofibroblasts still largely absent.

17. The bioinformatics analysis of the human keloid and psoriasis samples is interesting, but the data are overinterpreted. They do not provide clear evidence that “fascia progenitors differentiate into myofibroblasts directly through the pro-inflammatory state”. In particular, it is not clear if the pro-inflammatory fibroblasts are derived from fascia. It should also be considered (and mentioned) that fascia is very different in human skin compared to mouse skin.

18. It is interesting that CD201 is also a fibroblast (potentially fibroblast progenitor) marker in keloids and psoriatic skin. The authors should determine the site of expression of CD201 in normal human skin and keloids (and ideally also human skin wounds) by immunostaining. Is it also mainly expressed in the fascia?

Other points:

19. Introduction, first paragraph: Pro-inflammatory fibroblasts indeed contribute to the attraction of leukocytes. However, other signals, e.g. factors released upon blood clotting, ROS, or bacterial products, provide initial signals for the influx of leukocytes, which are more important (at least at the beginning) than fibroblasts. This should be clarified, otherwise the reader gets the impression that fibroblasts are the only regulators of wound inflammation.

20. Introduction, end of last paragraph: Myofibroblasts are already relevant in the second phase of wound healing – they are the major producers of ECM and clearly present during the phase of granulation tissue formation.

21. Introduction, first paragraph of the second page: Ref. 26 is not appropriate- it is a paper about arthritis and not a wound healing paper.

22. Results, first paragraph: The previous data from the authors indeed show that scar-forming fibroblasts are present in the fascia. However, publications from others (in particular from the Longaker lab) show that scar-forming fibroblasts are also present in other layers. Please clarify.

23. It is important to clearly mention the wound healing model that was used. Were these splinted wounds? This is suggested from some of the figures, but it is not clearly mentioned. What was the size of the wounds and the sex and age of the mice? In the method section the authors refer to previous papers, but it is also not clearly defined there. The type of wound model has strong influence on the behaviour of fibroblasts. The contribution of fascia fibroblasts may be different in wounds that are not splinted because of different biomechanical properties.

24. The authors should provide more details about the scRNA-seq experiments. It is mentioned that 29,383 cells from uninjured skin were sequenced – how many cells were sequenced from wounds at the different stages? From how many wounds and how many mice were these cells? Since wound healing is highly variable, it would be important to have at least N=2 (at least from an important time

point).

25. Page 6, first paragraph: Dermopontin should be replaced by Dermatopontin.

26. Fig. S3: It is not clear if the authors counted all cells in the wound (including epidermis) or only cells in the granulation tissue – please clarify. I suggest to show percentage of wound cells (change y-axis to %). How did the authors define the border between wound bed and fascia – there is no clear border and this seems a bit arbitrary.

27. It is surprising that no cells in uninjured skin are labeled in the Pdpn-CreER reporter mice, because podoplanin is also expressed in lymphatic endothelial cells. Please clarify.

28. Fig. 1k and l lacks information about the statistics. How many cultures were analyzed at what time point and from how many mice? (same for other figures with fascia cultures, e.g. Fig. 2).

29. Page 12, third paragraph (The pro-inflammatory transition is gated by retinoic acid) includes some references that are not appropriate. Ref. 48, for example, is a paper on cardiac fibrosis. There are many reviews on growth factors and cytokines in skin wounds, which could be cited here. There are many interleukins that have very different functions, and Ref 49 is a paper on IL-22 in scleroderma, and it is not clear why it was selected. For cell-cell interactions the author only cite their own manuscripts, but there are many more cell-cell interactions, and a review on this topic should be cited. Ref. 4 is generally on myofibroblasts – please also cite a review or original article on mechano-transduction in wound healing.

30. The potential role of RA in the transition and in scar formation is interesting, but there is insufficient evidence that RA really induces a pro-inflammatory state. CCL2 immunostaining is shown, but immunostaining for secreted factors is not very accurate. I suggest to perform qPCR for the pro-inflammatory cytokines that are expressed in pro-inflammatory fibroblasts in wounds in vivo.

31. Fig. S7b: Please confirm that YAP signaling is really activated in these cells upon addition of the agonist.

32. Fig. S7c: It is surprising that exogenous TGF- β 1 still has an effect in 10% serum. How much TGF- β 1 was used - Probably a rather high concentration?.

33. Fig. 7e: There is no concentration-dependent effect of the HIF inhibitor, because the low concentration had no effect. How specific is the inhibitor at the high concentration? Ideally, one would like to know the effect of HIF-1 α knock-down, which should be possible in the explant culture. Does the HIF inhibitor affect TGF- β receptor levels?

34. Page numbers are missing.

Referee #2 (Remarks to the Author):

In their article entitled, "CD201+ fascia progenitors choreograph injury repair", Correa-Gallegos, Rinkevich and colleagues use single cell RNA sequencing (scRNAseq) and bioinformatic analyses to describe fibroblast heterogeneity during wound healing using a mouse skin injury model. Their manuscript presents three central findings: (1) Their analyses suggest that fibroblast activation and differentiation in healing wounds occurs in a stepwise process, beginning with a fascia resident CD201+ fibroblast that differentiates through an inflammatory cell state into a proto- and terminal-myofibroblast. They confirm this trajectory using fate-tracking mouse models. (2) Their analyses also suggest that retinoic acid (RA) and hypoxia-inducible factor 1-alpha signaling (HIF-1a) are signals responsible for the transitions from resting tissue fibroblast to inflammatory fibroblast and from inflammatory fibroblast to protomyofibroblast, respectively. They confirm the requirement for these signaling pathways using in vitro culture models and in vivo pharmacological inhibition. (3) Their analyses of published scRNAseq data are suggestive that the differentiation trajectories and signaling pathways they identify in mouse are conserved to human and relevant to fibrotic diseases of the skin. While the subject of the manuscript is of great interest to the field, significant experimental, analytical, and editorial improvements are required before this manuscript can be considered for publication at Nature. Importantly, more analysis and experimental evidence is required to support the claims that: (1) Bioinformatic analyses and experimental evidence indicate that proto- and myofibroblasts differentiate from inflammatory fibroblasts and (2) RA and HIF-1a are cell intrinsic regulators of fibroblast cell state transition.

Major Points:

1. The analyses presented in Extended Data Figure 1h and 1i are central to the conclusions reached by the authors. These data, along with complete UMAP representations of all fibroblasts sequenced from the tissue (Extended Data Figure 1c) need to be presented in the main figures and the analysis needs to be explained more thoroughly. How confident are the authors that these data prove that Sfrp2+ fibroblasts do not differentiate into proto- or terminal-myofibroblasts? The authors should present more than one trajectory analysis method that supports their claims and should consider generating experimental evidence that demonstrates that myofibroblasts in their model do not differentiate from papillary or reticular progenitors or through an Sfrp2+ intermediate state.
2. In Figure 1a, the authors present a UMAP representation of a subset of fibroblasts from their data without adequate explanation. It is unclear why, for example, Sfrp2+ fibroblasts have been removed from this plot. These cells are clearly transcriptionally similar to the myofibroblasts and proto-myofibroblasts. How does this UMAP plot and pseudotime analysis compare if the Sfrp2+ fibroblasts are included in this analysis?
3. The authors clearly identify the Cd201.CreERT2 mouse model as effective in marking fascia fibroblasts in the skin. The authors should better characterize and describe the anatomical location of these cells in the fascia. The transcriptional profile described for these cells is reminiscent of that of a perivascular adventitial fibroblast and the fascia is highly vascularized; are these cells located near or associated with blood vessels?

4. To adequately support the claims that RA and HIF-1a signaling are required in a cell-intrinsic manner for the progressive differentiation of inflammatory fibroblasts and proto myofibroblasts, the authors must generate conditional knockout mouse models. The in vitro experiments presented do not adequately model disease physiology and the readouts of tissue contraction and single gene expression are not sufficient to define cell state transitions. The in vivo evidence presented is very encouraging, however small molecule agonism of RA signaling and inhibition of HIF-1a will likely influence multiple cell compartments so any effects of treatment cannot be attributed directly to pathway activity in the fibroblasts. scRNAseq will likely be a necessary readout for these conditional KO experiments.

5. In the human datasets presented, it is unclear why authors subset fibroblast clusters in extended data fig 9g and 9h. This would seem to bias their downstream analyses. Do other fibroblast populations give rise to myofibroblasts in the human skin based on bioinformatic analyses of the unsubsetted data set?

Other Points:

1. The authors state that cluster 4 – proto-fibroblasts uniquely express the gene *Cthrc1*, but the data presented in Extended Data Fig 1f appears to suggest that all myofibroblast clusters express this marker.

2. The text contains a number of incorrect figure references.

Referee #3 (Remarks to the Author):

Fascia is an interesting system since it covers entire organs to support 'structural continuity'. The fascia of skin is important because it harbors mesenchymal populations that contribute to wound repair and disease. The study in this manuscript builds on the labs previous work to understand how fascia responds to during wound healing so that skins continuity can be re-established. The authors of this manuscript present a novel roadmap for wound healing that is regulated by a two-step mechanism through the differentiation of fascia fibroblasts. The first step is regulated by retinoic acid signaling, which inhibits pro-inflammatory differentiation. The second step is regulated by hypoxia. The findings of this manuscript have the potential to bring new light to different approaches to treat wound healing and skin diseases and it will shine a spotlight on an important but poorly studied region of organs.

Major:

1) In figure 4. The authors link their findings to human skin utilizing scRNA-seq analysis. The authors should perform skin explant experiments to validate the human relevance.

2) The CD201 mouse model does not seem entirely specific. The authors should more clearly validate the Cd201 mouse model. In order to do this the authors should labeled the mice before injury and then to perform flow cytometry analysis. The critical foundational pictures in Figure 1 and

Extended Fig5 are not convincing to show that only fascia are labeled. This is because it appears in Figure 1e that the tdTomato cells in the dermis of the skin are out of focus and not bright. I don't think that this detracts from the story/message, just that readers should not be led to utilize this mouse model for only modulating fascia fibroblasts. Similar advise for the PdpnCreERt model should also be performed.

Minor

The authors should note in the text the size and shape of the wound performed in each experiment throughout the manuscript.

Author Rebuttals to Initial Comments:

Point-by-point response to the reviewer's comments and suggestions

We deeply appreciate the constructive comments and suggestions from the three reviewers, which have now been addressed in a fully revised manuscript. Please find below a detailed point-by-point response to the reviewer's comments/suggestions.

Referee #1 (Remarks to the Author):

Correa-Gallegos identified CD201+ cells (mainly cells from the fascia) as progenitors of wound fibroblasts, and they show their transition into pro-inflammatory fibroblasts and then into myofibroblasts. They further identify retinoic acid (RA) and HIF-1a signaling as key regulators of the different transition steps.

This is an interesting wound healing study with a large number of scRNA-seq and fate mapping data, which provide further insight into the roles of different fibroblast subpopulations in wound healing. It is a nice extension of the previous work from the authors, which identified an important role of fascia fibroblasts in the healing of deep wounds. While some results were predictable based on previous work of the authors and many others (incompletely cited - see below), the identification of CD201+ cells as progenitors of different wound fibroblasts is novel. However, it is not clear if all CD201+ cells are really fascia fibroblasts (see below). The role of RA and HIF-1a in the transition stages is also novel and interesting, but there are some technical limitations (see below), and other factors may also be important for this transition. Some of the results are overinterpreted.

We thank the reviewer for his/her comments. In this revised version we extensively expanded our experimental datasets and observations as well as made significant textual changes in order to respond to the reviewer's concerns (see below).

Major points:

1. Page 6, end of first paragraph: CD201 had previously been shown to be expressed by mesenchymal cells/fibroblasts. For example, CD201+ cells were found to be important for spleen regeneration (Tan and Watanabe, 2017), and PDGFRa+/CD201+ cells are mesenchymal progenitors in the skeletal muscle (Uezumi et al., 2016). These papers should be cited – otherwise one gets the impression that expression of CD201 in stromal cells and a role of CD201+ stromal cells as progenitors is entirely novel (which is not the case).

We have now included these papers into the revision and made textual changes to reference the above citations.

2. The data shown in Fig. S1h-I are highly relevant and should be moved to the main figure.

The legend should include more information (e.g. based on x sequenced cells from y wounds and z mice). These data indeed suggest that fascia fibroblasts can give rise to the other fibroblast subpopulations, but it is not functionally proven that “they present a single lineage that emerges exclusively from homeostatic fascia fibroblasts”. This should be expressed more carefully.

As requested, the data was included into a new Figure 1a-b. We also modified the Result section to better explain the complex array of interconnectivities between the cell clusters.

3. Fig. S3 and other figures showing wound healing data: Please mention the number of wounds that were analyzed and the number of mice. It seems that some of the data were only based on 3 wounds. Given the high variability of the wound healing process, this is not sufficient. Please also add magnification bars to all wounds – they are missing in all photomicrographs.. T-test is not suitable for in vivo samples, which usually do not show a Gaussian distribution.

We share the reviewer’s notion of the correct use of statistics in biological systems. In this particular case, statisticians were consulted when writing our animal experimental plans and they suggested a minimal “n” of three animals is sufficient based on previous literature using similar animal models. These in turn underwent severe scrutiny, also in the statistical aspect, by the Bavarian’s veterinarian officials and scientific experts. Furthermore, most, if not all, leading works in the field of skin wound healing systematically use parametrical methods such as Student’s t-test and ANOVA, as in PMID:30467144, PMID:24336287, PMID:35077667, PMID:32755548, and more recently PMID:35614212. Therefore, we don’t believe that non-parametric methods, which provide lower statistical certainty, would be more fitting in our manuscript.

As requested by the reviewer, we included the scale bars in all micrographs and indicated the biological and technical “n” in the figure legends.

4. The TAT-Cre experiment is elegant. However, it is not convincingly demonstrated that only fascia fibroblasts were targets. I can see green cells in the dermis of non-injured skin (Fig. S4b).. The high magnification picture (again no magnification provided) shows only the fascia and not the dermis – please also show a high magnification of the dermis. The specificity of the deletion is crucial for the interpretation of the data. It is also not clear which percentage of fascia fibroblasts was targeted by TAT-Cre. It is also possible that the TAT-Cre protein remains in the skin for some time and then deletes in many wound fibroblasts (and other cells) upon injury.

To ensure specificity of fascia fibroblasts, we are now replacing this supplementary data and including a much deeper expanded description of fascia fibroblasts using genetic lineage tracing approaches such as the CD201CreER genetic system. This approach showed

superior specificity as compared to TAT-Cre when tagging fascia fibroblasts (see point below and Extended data figure 4).

5. The 1.7-fold enrichment of CD201 in fascia fibroblasts is good, but CD201 is obviously not specific for fascia fibroblasts. Fig. 1d suggests that TdTomato is indeed mainly expressed by fascia fibroblasts. However, there seem to be a few positive cells in the dermis (Fig. 1e and S5b). It is still possible that a few dermal cells that are TdTomato positive contribute to the pool of positive cells in 7-day wounds. Are there still positive cells in the skin when you remove the fascia? CD201 is also expressed by hematopoietic stem/progenitor cells, which are known to contribute to wound healing. Some of the positive cells in the granulation tissue could come from the haematopoietic system (could be tested using bone marrow chimeric mice).

To address the reviewer's concerns, we significantly expanded our characterization of the CD201 transgenic system showing virtually exclusive labeling in fascia fibroblasts. We also observed no labeling of HSCs in the bone marrow and no expression of CD45/PTPRC in TdTomato-labeled cells in skin wounds (Extended data figure 3b-d), making unlikely that other sources giving rise to the observed cell pool in wounds. Compared to existing lineage tracing approaches, our data shows that the CD201CreER represents the most specific system to discriminate a fascia population of fibroblasts from its dermal fibroblast counterparts.

6. Page 11: The transcription factor analysis is interesting, but it should be validated at the protein level by staining for the active forms of the relevant transcription factors. For example, is there indeed more active Nfe2l2 in pro-inflammatory fibroblasts (or more of its antagonist Bach1)? And is there more HIF1a in the nucleus of proto-myofibroblasts and more Tcf4 in the myofibroblasts? Without such a validation these results are not very meaningful.

We agree that transcription factor analysis at protein levels would be interesting. However, the extensive characterization of all these TFs is out of the scope of this manuscript which focuses on the main signaling pathways that regulate the in-out from each fibroblast stage, namely RA and Hypoxia. Nonetheless, we have indeed validated the expression of STAT3, RUNX2 at protein level for the proto-, and myofibroblast programs. We have included new extensive characterization of the functions of another TF in the proto-myofibroblast, HIF1A, in Figure 4 and extended data figure 7. We have now also changed the aforementioned section to highlight the phenotype generated by these programs for each fibroblast state and their relation to the wound healing process.

7. The use of ex vivo fascia cultures is an interesting approach. However, it has clear limitations. First, the influence of other cells (in particular immune cells) and of endocrine factors cannot be studied. Most importantly, the fascia cultures were maintained in medium with 10% FCS, which includes a lot of TGF- β (and other growth factors and hormones). Therefore, it is not surprising that the tissue contracts, because TGF- β promotes the contractility of fibroblasts. Serum-free conditions (or at least low serum) would be more relevant. Do the cultures indeed express pro-inflammatory cytokines at day 1-3 and other

myofibroblast markers at day 6? What happens to CD201 expression in these cultures? The results of Fig. 1k-l certainly do not allow the conclusion that "the classic phases of wound healing... are directly regulated by the differentiation of CD201+ progenitors".

We appreciate the reviewer's suggestions. We included now cultures using low and heat-inactivated serum to prevent the action of any potential serum factors, such as TGF β . These new experiments show that fascia contraction persists under these conditions. Moreover, we now include new data confirming expression of cytokine CCL2 (Figure 3h) and α SMA (Figure 2c) within fascia cultures. Nonetheless, we rephrased our conclusions to better fit our observations highlighting their role on tissue contraction.

8. Fig. 2g shows contraction, but not myofibroblast differentiation – please stain for α -SMA. Is there a reduction in α -SMA and/or reduced stress fiber formation?

α SMA immunostaining and quantifications were included now (Figure 3h)

9. In the fascia explant experiment, RA was added at pharmacological concentrations (1 μ M, this is a very high concentration that is not within the physiological range..) – this experiment does not allow a conclusion about the role of endogenous RA. There is retinol in serum, which is converted to RA in fibroblasts and other cells. Therefore, it would be more important to determine if culture in retinol-free medium promotes myofibroblast differentiation. This experiment would also be important to support the statement that "RA signaling downregulation grants the exit and progression into myofibroblasts" (page 14, first paragraph), for which no functional evidence is provided so far. Overall, the RA experiments are interesting, but they do not clarify if endogenous RA is really the driving factor for the differentiation of CD201+ fibroblasts into pro-inflammatory fibroblasts.

We now expanded our set of experiments to better understand the endogenous role of RA in the proinflammatory state. These include:

- 1) Treatments with a ten-fold lower concentration of RA (0.1 μ M), still preventing fascia contraction, as well as a pan-RAR antagonist (enhancing contraction, Figure 3g-h, Extended data figure 7a). We would also like to point out that published studies exposing fibroblasts and organ cultures to RA normally use higher concentrations than those we presented in the previous manuscript (e.g. PMID:2354920 used up to 10-fold higher concentrations, PMID:8424454 used 3.3-fold higher, and more recently PMID:35902913 used the same 1 μ M concentration).
- 2) We now show that RA activation specifically promotes a monocyte-recruitment phenotype via inducing the expression of Ccl2 (Extended data figure 7b-g).

We now made textual changes to better fit the supportive role of RA in promoting the proinflammatory phenotype (specifically inducing monocyte recruitment).

10. Did the authors observe any toxicity of the RA at higher concentration? How about toxicity of any of the compounds used for the experiments in Fig. S7?

As pointed in point 9, similar and even higher working concentrations of RA have been previously published. The same goes for Hif1a inhibitors which have also been previously published in similar ranges of working concentrations (see PMID:35560016), making unlikely that the observed effects are due to toxic effects. For the reviewer's eyes, we directly explored potential toxicity by measuring the expression of Caspase 3, a marker of apoptosis showing no significant increase in the amount of Cas3+ cells in any of the treatments using the TAZ activator (Kaem), TGFb1, or the Hif1a inhibitor (EQM, see below).

ANOVA table	SS	DF	MS	F (DFn, DFd)	P value
Treatment (between columns)	178.4	5	35.68	F (5, 9) = 1.196	P=0.3832
Residual (within columns)	268.5	9	29.83		
Total	446.9	14			

11. The in vivo wound healing experiments with the RARg activator are important. However, the authors do not show that there is prolonged wound inflammation in the treated mice. They only show more CD45+ cells at day 3. This could also be the result of a delayed influx. To make this claim, a time course experiment would be required. In addition, the different immune cells have very different functions in wound healing – which immune cells are affected by the activator?

We appreciate the reviewer's suggestions. We now explored in more detail the prolonged inflammation phase caused by our treatment, showing that, specifically, monocyte/macrophage recruitment is boosted for up to a week after injury (please see new Extended data figure 7b-c)

12. How did the authors determine the relevant dose of the RARg activator for the wound healing studies and how many injections were done around the wound? Did the authors consider a topical treatment? What is the effect of the RARg activator on the speed of wound healing? The reduced scarring may also be a consequence of faster re-epithelialisation. Enhanced healing would be consistent with previous studies showing a healing promoting activity of RA (e.g. Lateef et al., 2005). Please cite previous publications that demonstrated effects of RA on wound healing.

Working concentration of the RARg agonist was determined via literature search (PMID:28642153) and the reason behind not using a topical treatment was to mitigate any direct effect of the agonist on the epidermis itself. As the reviewer also points out, topical RA exerts a beneficial effect in slowly healing wounds of genetically diabetic mice, particularly by boosting keratinocyte activity (PMID:11820728). As we are interested in the effects of RA signal activation within the core and deep layers of the wound, where the fascia and proinflammatory fibroblasts reside, we opted for subcutaneous injections as a more direct way of administration. The third paragraph of the discussion section is solely dedicated to the regenerative effects reported with retinoic acid treatments including now relevant skin references (please see point below).

13. The anti-fibrotic effect of RA is also not novel. The authors cite papers that show this effect in other organs, but skin data are not mentioned. For example, Okarinen et al., already showed in 1985 that RA inhibits collagen production in skin fibroblasts and it has been suggested as a treatment for systemic sclerosis (RM Thomas et al., 2016). Please cite relevant papers about RA effects on skin fibrosis.

We thank the reviewer for his/her comments, and we have now included the relevant references into the discussion as well as in the reference list.

14. The effect of depletion of pro-inflammatory fibroblasts on wound healing is not surprising – in these experiments, a large percentage of the granulation tissue fibroblasts was depleted and therefore, an important source of ECM proteins and growth factors was removed. It also seems likely that the massive cell death that occurs in these mice upon DT treatment causes strong inflammation, which impairs the healing process. In the opinion of this reviewer, this experiment does not confirm that podoplanin-positive fibroblasts are the progenitors of myofibroblasts as stated on page 15.

We appreciate the reviewer's comment but respectfully disagree with their assumptions due to the following points:

1) no DT treatment was involved in the experiment as the toxin is internally produced by the cells upon Cre recombination. Therefore, no inflammation due to any treatment with external toxin is possible.

2) A strong inflammation would cause a notable leukocyte infiltration which is obviously not the case as seen in the Trichrome staining images (Figure 2f), arguing again against a strong inflammation.

3) We have included several lines of evidence to prove that proinflammatory fibroblasts give rise to myofibroblasts, including bioinformatic trajectory/connectivity analysis (Figure 1a-c, extended data figures 1-3), complementary genetic lineage tracing methods (Figure 1f-k, extended data figure 6), and functional validation via chemical activation/inhibition (RA and HIF1a) showing that sustaining the proinflammatory or blocking the later states results in a negative effect on myofibroblasts numbers. Furthermore, the same genetic ablation experiment directly shows a dramatic decrease in the amount of myofibroblasts.

Nonetheless and in response to the reviewer's concerns, we have now repurposed this data to highlight the importance of the differentiation trajectory in the wound healing process as a whole when this gets blocked (Figure 2e-f).

15. The HIF-1 inhibitor experiments are interesting, but it is not shown that HIF1a protein levels are indeed higher in podoplanin-positive wound fibroblasts compared to other fibroblasts. In addition, one would like to know if there is HIF1a in the nucleus of the fibroblasts from ex vivo fascia cultures. Does ex vivo culture of fascia under hypoxic conditions affect myofibroblast differentiation?

We have now addressed the reviewer's comments, by validating the HIF1a protein activity *in silico*, and by HIF1a protein expression *in vivo* and *in vitro* at relevant times during wound healing/tissue contraction, mainly during the transition period from proinflammatory to proto-myofibroblasts (please see new Figure 4a-b)

16. The authors show that wound healing is impaired upon treatment with the HIF1 inhibitor. This is nicely demonstrated, but not at all surprising. HIF1a activity is required in many cell types in the wound, including macrophages, and the important role of HIF1a in wound in wound healing has previously been demonstrated (see for example the review of WX Hong et al., 2014). There is even a study where HIF1a was knocked out in fibroblasts – the authors observed impaired wound angiogenesis and impaired wound closure in these mice (Duscher et al., 2015). Knockout of HIF1a in other cell types also affected the wound healing process. Unfortunately, none of these studies has been cited, which gives the impression that the role of HIF1a described in the present study is novel (which is not the case). The most interesting experiment is the one shown in Fig. 3g, but it is not clear when the treatment with the inhibitor was started. Ideally, it should be started at around day 3-5 when inflammatory fibroblasts are present, but myofibroblasts still largely absent.

To confirm the Hif1a role specifically in fascia and proinflammatory fibroblasts we now included cell-specific gene knockout experiments showing the same effects as the chemical inhibitor treatment (Figure 4d-e). We have also included additional references highlighting previous findings on HIF1a in wound healing, all based on the reviewer's request.

17. The bioinformatics analysis of the human keloid and psoriasis samples is interesting, but the data are overinterpreted. They do not provide clear evidence that "fascia progenitors differentiate into myofibroblasts directly through the pro-inflammatory state". In particular, it is not clear if the pro-inflammatory fibroblasts are derived from fascia. It should also be considered (and mentioned) that fascia is very different in human skin compared to mouse skin.

Based on the reviewer's suggestion, we have now expanded our analysis of human skin samples, and included the following new approaches:

- 1) We now used a machine learning method to map the mouse fibroblast equivalents in the merged human datasets, giving higher confidence of the detection of the different fibroblast states in the human datasets (Figure 5a and extended data figure 10a).
- 2) We performed a gene expression comparison to describe the transcriptional differences between mouse and human subpopulations (Figure 5a and extended data figure 10b-c).
- 3) We provide new RNA velocity-assisted connectivity analysis, just as with mouse, in which we show a direct transition from fascia into proinflammatory fibroblasts (Figure 5c), and later into myofibroblasts.
- 4) We validated the presence of fascia fibroblasts (via PI16 expression, point below) in healthy human skin and the presence of (RUNX2⁺) myofibroblasts in keloid lesions (Figure 5b).

18. It is interesting that CD201 is also a fibroblast (potentially fibroblast progenitor) marker in keloids and psoriatic skin. The authors should determine the site of expression of CD201 in normal human skin and keloids (and ideally also human skin wounds) by immunostaining. Is it also mainly expressed in the fascia?

Despite testing several commercial antibodies, we were unable to find any working anti-CD201 antibody suitable for immunolabeling on human sections. Instead, we have now validated the presence of fascia fibroblasts in healthy human skin via expression of PI16, one of the best conserved markers for fascia fibroblasts between mice and human datasets (please see new Figure 5b)

Other points:

19. Introduction, first paragraph: Pro-inflammatory fibroblasts indeed contribute to the attraction of leukocytes. However, other signals, e.g. factors released upon blood clotting, ROS, or bacterial products, provide initial signals for the influx of leukocytes, which are more important (at least at the beginning) than fibroblasts. This should be clarified, otherwise the reader gets the impression that fibroblasts are the only regulators of wound inflammation.

We rephrased the text to better describe the role of proinflammatory fibroblasts.

20. Introduction, end of last paragraph: Myofibroblasts are already relevant in the second phase of wound healing – they are the major producers of ECM and clearly present during the phase of granulation tissue formation.

As per the point above, we rephrased the text according to the reviewer's comment.

21. Introduction, first paragraph of the second page: Ref. 26 is not appropriate- it is a paper about arthritis and not a wound healing paper.

We apologize for the confusion, the reference is intended to highlight the significance of spatial distribution of fibroblast subsets to injury repair, regardless of the biological system (this aspect is mostly neglected in wound healing studies). Based on the reviewer's comment, we moved the reference into a more fitting location.

22. Results, first paragraph: The previous data from the authors indeed show that scar-forming fibroblasts are present in the fascia. However, publications from others (in particular from the Longaker lab) show that scar-forming fibroblasts are also present in other layers. Please clarify.

We now have rephrased this paragraph to clarify the scar-forming capacity of other fibroblast subsets.

23. It is important to clearly mention the wound healing model that was used. Were these splinted wounds? This is suggested from some of the figures, but it is not clearly mentioned. What was the size of the wounds and the sex and age of the mice? In the method section the authors refer to previous papers, but it is also not clearly defined there. The type of wound model has strong influence on the behaviour of fibroblasts. The contribution of fascia fibroblasts may be different in wounds that are not splinted because of different biomechanical properties.

We have now included a more detailed description of the type of injuries performed: 1st paragraph of the first result subsection and 4th paragraph of the 4th subsection. We also specified in the M&M section, the mice sex, age, and wound size details.

24. The authors should provide more details about the scRNA-seq experiments. It is mentioned that 29,383 cells from uninjured skin were sequenced – how many cells were sequenced from wounds at the different stages? From how many wounds and how many mice were these cells? Since wound healing is highly variable, it would be important to have at least N=2 (at least from an important time point).

As requested by the reviewer, details on the number of analyzed cells in the scRNAseq dataset for each timepoint has been included in the extended data figure 1b (note that in this case only the numbers of post-processed high-quality cells is provided). Wounds and mouse numbers have been disclosed in the materials and methods section “Samples from three mice were pooled for each stage (6 wounds per time-point)”.

25. Page 6, first paragraph: Dermopontin should be replaced by Dermatopontin.

We thank the reviewer and have made the necessary textual changes.

26. Fig. S3: It is not clear if the authors counted all cells in the wound (including epidermis) or only cells in the granulation tissue – please clarify. I suggest to show percentage of wound cells (change y-axis to %). How did the authors define the border between wound bed and fascia – there is no clear border and this seems a bit arbitrary.

We thank the reviewer and would like to highlight this data is available in the image analysis subsection of the materials and methods:

“The wound lateral limits were delimited as stated before [The wound region of interest (ROI) was manually selected ... taking as reference the breached *Panniculus carnosus* below the dermis and the flanking hair follicles in the epidermis above], the upper wound region extended vertically 200 microns from the wound surface (at 3 dpi) or the wound epidermis (at 7 dpi). The fascia was measured as the area extending below the breached *P. carnosus* muscle down to the dorsal muscles, while the wound core covered the area between the wound fascia and the upper wound. Non-mesenchymal areas (e.g, hair follicles, wound epidermis) were deleted from the ROIs.”

Thus,

- 1) no epidermis was included in the analysis.
- 2) border between fascia and wound bed is defined using the panniculus carnosus as a spatial reference.
- 3) Cell ratio (positive cells/total cells) is equivalent to cell percentage, in which 1 is the same as 100%.

To make the area selection criteria clearer, we have now added these spatial cues in the scheme of the extended data figure 4d.

27. It is surprising that no cells in uninjured skin are labeled in the Pdpn-CreER reporter mice, because podoplanin is also expressed in lymphatic endothelial cells. Please clarify.

We indeed see lymphatic vessels being labeled with GFP, and we refer only to the lack of expression in skin fibroblasts, we have now added additional data regarding LYVE1 expression in new Extended data figure 6c.

28. Fig. 1k and l lacks information about the statistics. How many cultures were analyzed at what time point and from how many mice? (same for other figures with fascia cultures, e.g. Fig. 2).

We now have included in all figure legends details about the biological and technical replicates for all quantified experiments, based on the reviewer's comment.

29. Page 12, third paragraph (The pro-inflammatory transition is gated by retinoic acid) includes some references that are not appropriate. Ref. 48, for example, is a paper on cardiac fibrosis. There are many reviews on growth factors and cytokines in skin wounds, which could be cited here. There are many interleukins that have very different functions, and Ref 49 is a paper on IL-22 in scleroderma, and it is not clear why it was selected. For cell-cell interactions the author only cite their own manuscripts, but there are many more cell-cell interactions, and a review on this

topic should be cited. Ref. 4 is generally on myofibroblasts – please also cite a review or original article on mechano-transduction in wound healing.

Appropriate skin references have now been added to this section.

30. The potential role of RA in the transition and in scar formation is interesting, but there is insufficient evidence that RA really induces a pro-inflammatory state. CCL2 immunostaining is shown, but immunostaining for secreted factors is not very accurate. I suggest to perform qPCR for the pro-inflammatory cytokines that are expressed in pro-inflammatory fibroblasts in wounds in vivo.

We have now provided evidence proving that RA promotes the expression of Ccl2 via qPCR in Extended data figure 7 e-f

31. Fig. S7b: Please confirm that YAP signaling is really activated in these cells upon addition of the agonist.

As requested, we show increased expression of YAP1 in treated samples (Extended data 8c)

32. Fig. S7c: It is surprising that exogenous TGF-b1 still has an effect in 10% serum. How much TGF-b1 was used - Probably a rather high concentration?.

The concentration used was 10uM of recombinant TGFb1 protein (stated in the material and methods). Indeed, a high concentration compared to cells-on-dish culture systems. As we presented a more complex ex vivo system, higher proteins concentration was required. The intent and purpose of the experiment was to test the relevance of HIF1a activity while competing with other inductive signals (e.g. TGFb) and not study TGFb itself, as its role as inductor of myofibroblast conversion is widely known. The idea was to test whether inhibiting HIF1a activity would still prevent tissue contraction even in the presence of an excess of TGFb, which was the case.

33. Fig. 7e: There is no concentration-dependent effect of the HIF inhibitor, because the low concentration had no effect. How specific is the inhibitor at the high concentration? Ideally, one would like to know the effect of HIF-1a knock-down, which should be possible in the explant culture. Does the HIF inhibitor affect TGF-b receptor levels?

We have now included additional data showing that HIF1a inhibition significantly reduces pSMAD2 expression, indicating a downregulation of the TGFb pathway, even in the presence of an excess of TGFb1 (Extended data 8d). We have also rephrased the result section related to the HIF1a inhibition experiments to better describe our observations. Furthermore, we have performed KO experiments in vivo validating our observations with the inhibitor (Figure 4d-e).

34. Page numbers are missing.

Page numbers have now been added

Referee #2 (Remarks to the Author):

In their article entitled, "CD201+ fascia progenitors choreograph injury repair", Correa-Gallegos, Rinkevich and colleagues use single cell RNA sequencing (scRNAseq) and bioinformatic analyses to describe fibroblast heterogeneity during wound healing using a mouse skin injury model. Their manuscript presents three central findings: (1) Their analyses suggest that fibroblast activation and differentiation in healing wounds occurs in a stepwise process, beginning with a fascia resident CD201+ fibroblast that differentiates through an inflammatory cell state into a proto- and terminal- myofibroblast. They confirm this trajectory using fate-tracking mouse models. (2) Their analyses also suggest that retinoic acid (RA) and hypoxia-inducible factor 1-alpha signaling (HIF-1a) are signals responsible for the transitions from resting tissue fibroblast to inflammatory fibroblast and from inflammatory fibroblast to protomyofibroblast, respectively. They confirm the requirement for these signaling pathways using in vitro culture models and in vivo pharmacological inhibition. (3) Their analyses of published scRNAseq data are suggestive that the differentiation trajectories and signaling pathways they identify in mouse are conserved to human and relevant to fibrotic diseases of the skin. While the subject of the manuscript is of great interest to the field, significant experimental, analytical, and editorial improvements are required before this manuscript can be considered for publication at Nature. Importantly, more analysis and experimental evidence is required to support the claims that: (1) Bioinformatic analyses and experimental evidence indicate that proto- and myofibroblasts differentiate from inflammatory fibroblasts and (2) RA and HIF-1a are cell intrinsic regulators of fibroblast cell state transition.

We appreciate the reviewer's comments and we have now included significant changes in our revised manuscript to address this reviewer's concerns.

Major Points:

1. The analyses presented in Extended Data Figure 1h and 1i are central to the conclusions reached by the authors. These data, along with complete UMAP representations of all fibroblasts sequenced from the tissue (Extended Data Figure 1c) need to be presented in the main figures and the analysis needs to be explained more thoroughly. How confident are the authors that these data prove that Sfrp2+ fibroblasts do not differentiate into proto- or terminal-myofibroblasts? The authors should present more than one trajectory analysis method that supports their claims and should consider generating experimental evidence that demonstrates that myofibroblasts in their model do not differentiate from papillary or reticular progenitors or through an Sfrp2+ intermediate state.

We have now shifted the related image panels into new Figure 1a-b and included a better description of our rationale to focus our efforts in studying the fascia-to-myofibroblast trajectory for the following reasons:

1) Our bioinformatic analysis suggests that our trajectory is the most likely source giving rise to myofibroblasts, and substantially corroborated by several lineage-specific genetic lineage tracing approaches.

2) Although our analysis also indicate that reticular fibroblasts could directly give rise to myofibroblasts (which has been already proven PMID:24336287), our previous studies indicate that, in full-thickness skin wounds, fascia fibroblasts represent the most prominent source for myofibroblasts (PMID:31776510). This study is primarily focused on this neglected fibroblast subtype instead of the vastly well studied reticular population.

3) Regarding the Sfrp2+ cluster, please refer to the point below.

2. In Figure 1a, the authors present a UMAP representation of a subset of fibroblasts from their data without adequate explanation. It is unclear why, for example, Sfrp2+ fibroblasts have been removed from this plot. These cells are clearly transcriptionally similar to the myofibroblasts and proto-myofibroblasts. How does this UMAP plot and pseudotime analysis compare if the Sfrp2+ fibroblasts are included in this analysis?

We thank the reviewer for this point and would like to clarify. Our trajectory inference (Figure 1a-b) suggests that Sfrp2+ cluster does not transit into any further fate and its most likely source are reticular fibroblasts (0.25 compared to other potential sources connectivity values <0.066). Therefore, this state is not present in the trajectory sprouting from the fascia fibroblasts in their way to myofibroblasts and further characterization of this state remains out of the scope of this work.

3. The authors clearly identify the Cd201.CreERT2 mouse model as effective in marking fascia fibroblasts in the skin. The authors should better characterize and describe the anatomical location of these cells in the fascia. The transcriptional profile described for these cells is reminiscent of that of a perivascular adventitial fibroblast and the fascia is highly vascularized; are these cells located near or associated with blood vessels?

We now extensively expanded our characterization of the CD201CreER system (Extended data figure 5b-d) and, particularly, we provide evidence showing that labeled fascia fibroblasts are present along the entire fascia connective tissue and are not associated with blood vessels (Figure 1h)

4. To adequately support the claims that RA and HIF-1a signaling are required in a cell-intrinsic manner for the progressive differentiation of inflammatory fibroblasts and proto myofibroblasts, the authors must generate conditional knockout mouse models. The in vitro experiments presented do not adequately model disease physiology and the readouts of tissue contraction and single gene expression are not sufficient to define cell state transitions. The in vivo evidence presented is very encouraging, however small molecule agonism of RA signaling and inhibition of HIF-1a will likely influence multiple cell

compartments so any effects of treatment cannot be attributed directly to pathway activity in the fibroblasts. scRNAseq will likely be a necessary readout for these conditional KO experiments.

We have now included new data that strongly supports our hypothesis regarding the intrinsic activity of RA and HIF1A along our differentiation trajectory. Particularly the following experiments:

- 1) RA treatments on purified CD201+ fascia fibroblasts cause direct gene expression changes towards proinflammatory states, such as CCL2 and PDPN expression upregulation as well as reduction of myofibroblast markers such as α SMA (Extended data figure 7e-g).
- 2) We now show in silico activity and in vivo expression patterns of Hif1a in CD201- and in PDPN-derived fibroblasts as well as in whole fascia explants (Figure 4a-b).
- 3) We have now included an induced Hif1a gene knockout using our complementary genetic systems (CD201^{CreER} and Pdpn^{CreER}), which replicate our chemical treatments (Figure 4d-e).

5. In the human datasets presented, it is unclear why authors subset fibroblast clusters in extended data fig 9g and 9h. This would seem to bias their downstream analyses. Do other fibroblast populations give rise to myofibroblasts in the human skin based on bioinformatic analyses of the unsubsetted data set?

We have now provided an unbiased method for detection of all fibroblasts in the human datasets using a machine learning algorithm and RNA velocity assisted PAGA analysis (Extended data figure 10a and Figure 5a/c). For the reviewer's eyes, we are providing below new PAGA velocity graphs that represents the complete unbiased lineage trajectories in the human psoriatic (top) and keloid datasets (bottom), both of which showing the fibroblast differentiation subsets as originating from the fascia fibroblasts.

paga velocity-graph (clusters)

paga velocity-graph (clusters)

When performing the RNA velocity calculations, we noticed that the interconnectivities between clusters were more limited compared to our mouse dataset. We suspect that the poor interconnectivity shown in the human datasets might be the result of the biological and temporal limitations of the experimental set-up (e.g., limited variable biological numbers of usually 3 patients and only one late not-matched time-point in the disease). Models for trajectory inference indeed are more robust when dealing with longitudinal (multiple timepoints) datasets. Yet, both show indirect or direct transition of fascia fibroblasts into myofibroblasts, suggesting that indeed fascia represents an important source for myofibroblasts in human as well.

As downstream analyses were intended to test the conserved chronicity of the signaling pathways (RA, hypoxia, and TGF β as reference) between human and mice along specifically the trajectory of fascia into myofibroblasts, it became irrelevant the inclusion of other clusters. As stated before, the scope of this work is focused on this particular trajectory and not on the contribution between different sources.

Other Points:

1. The authors state that cluster 4 – proto-fibroblasts uniquely express the gene *Cthrc1*, but the data presented in Extended Data Fig 1f appears to suggest that all myofibroblast clusters express this marker.

We have now rephrased the text to indicate it is highly expressed in cluster 4 (SFRP2⁺ not proto-myofibroblasts) and have removed the wording “uniquely expressed”.

2. The text contains a number of incorrect figure references.

We have now corrected the figure references in the revised manuscript.

Referee #3 (Remarks to the Author):

Fascia is an interesting system since it covers entire organs to support ‘structural

continuity'. The fascia of skin is important because it harbors mesenchymal populations that contribute to wound repair and disease. The study in this manuscript builds on the labs previous work to understand how fascia responds to during wound healing so that skins continuity can be re-established. The authors of this manuscript present a novel roadmap for wound healing that is regulated by a two-step mechanism through the differentiation of fascia fibroblasts. The first step is regulated by retinoic acid signaling, which inhibits pro-inflammatory differentiation. The second step is regulated by hypoxia. The findings of this manuscript have the potential to bring new light to different approaches to treat wound healing and skin diseases and it will shine a spotlight on an important but poorly studied region of organs.

We thank the reviewer for his/her comments.

Major:

1) In figure 4. The authors link their findings to human skin utilizing scRNA-seq analysis. The authors should perform skin explant experiments to validate the human relevance.

We now provide human fascia explant data as suggested by the reviewer (see new Figure 5e).

2) The CD201 mouse model does not seem entirely specific. The authors should more clearly validate the Cd201 mouse model. In order to do this the authors should labeled the mice before injury and then to perform flow cytometry analysis. The critical foundational pictures in Figure 1 and Extended Fig5 are not convincing to show that only fascia are labeled. This is because it appears in Figure 1e that the tdTomato cells in the dermis of the skin are out of focus and not bright. I don't think that this detracts from the story/message, just that readers should not be led to utilize this mouse model for only modulating fascia fibroblasts. Similar advise for the PdpnCreERt model should also be performed.

We appreciate the reviewer's comment, and have now expanded our characterization of the CD201 reporter system using flow cytometry to show its specificity to fascia fibroblasts in skin studies (see new Extended data figure 5b-c). Similar expanded characterization was also now performed with the PDPN system and both analyses have been included into the revision (see new Extended data figure 6b-c).

Minor

The authors should note in the text the size and shape of the wound performed in each experiment throughout the manuscript.

PDPN-ablation models were performed with a splinted wound model (and mentioned in paragraph 4 of the "Interrupting the differentiation trajectory impairs wound healing" result subsection). All remaining skin injuries were performed without splints, as stated in the M&M section. We have now added an initial description of the type of injury in the second paragraph of the Result section.

Reviewer Reports on the First Revision:

Referees' comments:

Referee #1 (Remarks to the Author):

The authors have performed a large number of additional experiments and bioinformatics analyses, and the revised manuscript is clearly improved. As recommended by the reviewers, important previous publications have now been cited. This allows the reader to judge which of the presented data are really novel. Nevertheless, this reviewer believes that some of the data are still overinterpreted. For example, the experiments with the *PdgnCreER/R26DTA* mice clearly show that the loss of *Pdgn*-positive fibroblasts (which are abundant!) and also of *Pdgn*-positive lymphatic endothelial cells (not mentioned here) causes impaired wound healing. However, this does not allow the conclusion that the “transition from pro-inflammatory to contractile states actively promotes tissue-level architectural changes”. It is very likely that pro-inflammatory fibroblasts already contribute to wound healing – independent of their future phenotype. Furthermore, the results shown in Fig. 4 show that HIF-1alpha is important for wound healing (as previously described by others), and the new conditional knockout mouse data are very important. However, this does not allow the conclusion that the effect of HIF-1alpha on myofibroblast differentiation is entirely responsible for the impaired healing. HIF-1alpha may also regulate various genes that directly have an effect on wound healing (e.g. VEGF, which promotes angiogenesis). This should be discussed more carefully.

This reviewer is still concerned about the low number of mice used for functional wound healing studies (N=3). The authors mention that they performed a biometric analysis, but it is not clear on which data this analysis is based. Wound healing is unfortunately a highly variable process, and N=3 does not provide reliable data in functional wound healing experiments. This is particularly problematic since the authors obviously used male and female mice for their studies. There are well known differences in the speed of repair between male and female mice (see for example Ashcroft and Mills, 2002, and other studies). With N=3 they may have compared three male with three female mice..

The data with human fascia are important, but they were obviously only obtained with fascia from one donor. The authors should at least mention the source of the fascia in Material and Methods (which part of the skin?) and provide the information about the ethical approval.

The inclusion of data from psoriasis is confusing. Psoriasis is not a fibrotic skin disease - scleroderma data would be more relevant.

Minor points:

It would have been helpful if the authors had marked their changes in the revised manuscript.

The distribution of α -SMA in Fig. 1k is strange – α -SMA usually clusters at the wound edge. In

general, the aSMA staining is not well visible.

Please refer to Fig. 5e at the end of the first paragraph of page 22.

Line 682: RA signalling is not “degraded” – only molecules are degraded. Please rephrase.

Materials and Methods: Please replace Hydrotamoxifen by Hydroxytamoxifen

Referee #3 (Remarks to the Author):

In this resubmission by Correa-Gallegos et al. the authors have made progress in addressing one concern, while not fully addressing another.

1) The first concern regarding performing skin explant experiments using human tissue in skin explant assays was performed and has addressed my concerns.

2) Unfortunately, the request that the validation of the labeling specificity of the CD201CreERT2 and PdpnCreERT2 mouse lines (also noted by other reviewers) to be fully characterized in regard to the fascia in homeostatic conditions (before wounding) has not been performed. The experiments that were performed by the authors in Extended Fig 5 are from wounded tissue. And it is impossible to interpret the origin of the cells without characterizing labeling of CD201 and Pdpn mice from homeostatic conditions. Without this foundational knowledge it becomes difficult to understand the downstream experiments in this manuscript appropriately.

Author Rebuttals to First Revision:

Point-By-Point response to editor

Referees' comments:

Referee #1 (Remarks to the Author):

The authors have performed a large number of additional experiments and bioinformatics analyses, and the revised manuscript is clearly improved. As recommended by the reviewers, important previous publications have now been cited. This allows the reader to judge which of the presented data are really novel. Nevertheless, this reviewer believes that some of the data are still overinterpreted. For example, the experiments with the *PdgnCreER/R26DTA* mice clearly show that the loss of *Pdgn*-positive fibroblasts (which are abundant!) and also of *Pdgn*-positive lymphatic endothelial cells (not mentioned here) causes impaired wound healing. However, this does not allow the conclusion that the “transition from pro-inflammatory to contractile states actively promotes tissue-level architectural changes”. It is very likely that pro-inflammatory fibroblasts already contribute to wound healing – independent of their future phenotype. Furthermore, the results shown in Fig. 4 show that HIF-1alpha is important for wound healing (as previously described by others), and the new conditional knockout mouse data are very important. However, this does not allow the conclusion that the effect of HIF-1alpha on myofibroblast differentiation is entirely responsible for the impaired healing. HIF-1alpha may also regulate various genes that directly have an effect on wound healing (e.g. VEGF, which promotes angiogenesis). This should be discussed more carefully.

We thank referee #1 for these comments and have now discussed more carefully the limitations of our models.

In particular:

1) We rephrased the ablation model results as suggested:

[lines: 361-4, , also highlighted in the current manuscript] *“Given that our $PDPN^{CreER}$ reporter line showed a preference to label proinflammatory fibroblast over lymphatic vessels (Extended data figure 6c), we then tested in vivo the relevance of the proinflammatory fibroblast state by performing genetic cell ablation of $PDPN^+$ proinflammatory fibroblasts in vivo.”*

[lines: 381-2] *“Altogether, our results indicate that the fascia-derived proinflammatory fibroblasts may play a pivotal role for the proper progression of the wound healing phases.”*

2) Attending to the referee's concern about the link of Hif1a with the myofibroblast differentiation, we also rephrased our interpretation of the results and discuss the role of Hif1a with TGFb (see point 3 below):

Result section:

[lines: 482-4] *"This data strongly suggests that hypoxia, through Hif1 α , plays an important role in lineage differentiation of fascia progenitor into proto- and myofibroblasts."*

[lines: 574-6] *"Our data shows that the Hif1 α driven transition into proto- and myofibroblast states is pivotal for connective tissue contraction of wounded skin."*

Discussion:

[lines: 686-701] *"Our ex vivo and in vivo studies indicate that hypoxia signal via Hif1 α plays a key role in the acquisition of the proto- and myofibroblast state from fascia progenitors. Supporting this notion, Hif1 α downregulation in pericytes⁸⁰ or fibroblasts⁸¹⁻⁸² prevents their transition into myofibroblasts in vitro and their pharmacological inhibition⁸³⁻⁸⁴ attenuates lung fibrosis. Furthermore, Hif1 α deletion in all skin fibroblasts has previously been shown to impair wound closure⁸⁵ without a biological basis for these observations. Classical roles of Hif1 α include preventing reactive oxygen species accumulation in immune cells during the inflammatory phase and promoting angiogenesis of endothelial cells during the proliferation phase⁸⁶. In contrast, Hif1 α has only been indirectly linked to myofibroblast transition via its synergy with the TGF β signalling⁸⁶, hypoxia enhances TGF β response in dermal fibroblasts in-vitro⁸⁷, and Hif1 α downregulation prevents TGF β -induced myofibroblasts transition in lung fibroblasts⁸³. Our in silico and ex vivo experiments demonstrate a direct proof for a dominant effect of Hif1 α inhibition over the activation of the TGF β pathway. Altogether, our observations indicate that Hif1 α plays a key role during wound healing by licensing the differentiation of fascia progenitors into myofibroblast by modulating the TGF β pathway."*

3) To further address the reviewer's concerns on the effect of HIF1a and myofibroblast differentiation, we have generated new in silico evidence showing that Hif1a directly regulates genes that modulate the TGFb pathway [lines: 500-508] and Extended data figure 8f-h:

[NEW] Extended data Fig. 8 | HIF1 α instructs the transition into contractile (proto)myofibroblasts. f. Correlation analysis of Hif1 α regulon genes (DoRoThEA and ENCODE) vs hypoxia signal activity score (PROGENY) to identify positively regulated genes (bottom) involved in TGF β signalling (marked in magenta). **g.** UMAPs of Hif1 α -regulated TGF β modulators expression levels. **h.** Schematic representation of the Hif1 α targets that influence TGF β signal activity.

This reviewer is still concerned about the low number of mice used for functional wound healing studies (N=3). The authors mention that they performed a biometric analysis, but it is not clear on which data this analysis is based. Wound healing is unfortunately a highly variable process, and N=3 does not provide reliable data in functional wound healing experiments. This is particularly problematic since the authors obviously used male and female mice for their studies. There are well known differences in the speed of repair between male and female mice (see for example Ashcroft and Mills, 2002, and other studies). With N=3 they may have compared three male with three female mice..

We understand the referee's concern about the difference in wound closure rate between female and male mice and how that could give unreliable results with relatively small groups.

Although we used indistinctly males and females for descriptive experiments (as mentioned in the materials and methods section), all functional studies in this paper employed mostly male mice and, whenever we used females, we split them in equal numbers between the groups.

To address the referee's concerns, we provide a table describing the groups used for the functional in vivo studies:

Experiment	Figures	Groups	n (per group)	N (total wounds)	Females	Males	Used for closure rate analysis
PDPN ⁺ ablation d14	Figure 2e	3 (R26 ^{WT} , R26 ^{mTmG} , and R26 ^{DTA})	3	18	1/1/1	2/2/2	YES
PDPN ⁺ ablation d7	Figure 2f	2 (R26 ^{WT} and R26 ^{DTA})	3	12	1/1	2/2	X
RARg agonist treatment 1 d3	Figure 3i	2 (treated and control)	3	12	X	3/3	X
RARg agonist treatment 1 d14	Figure 3j	2 (treated and control)	3	12	X	3/3	X
RARg agonist treatment 2 d3	EDF 7d	2 (treated and control)	3	12	X	3/3	X
RARg agonist treatment 2 d7	EDF 7b-d	2 (treated and control)	3	12	X	3/3	YES
HIF1a inh. Treatment d3	EDF 8h	2 (treated and control)	3	12	1/1	2/2	X

HIF1a inh. Treatment d7	EDF 8f-g, Figure 4c and f-g	2 (treated and control)	3	12	X	3/3	YES
HIF1a genetic deletion	Figure 4d-e	3 (Control, fascia-null, and proinf-null)	3	18	X	3/3/3	YES

As the referee can see, all key functional experiments assessing wound closure rate (Hif1a inhibition and genetic deletion, and RARg agonist treatment) were performed only in male mice, and only in one experiment females were included in same numbers between the groups.

It is also worth mentioning, that the referenced paper by the referee (Ashcroft and Mills PMID: 12208862) shows clear differences in the closure rate between normal and castrated males but not between females versus males. Other studies comparing males and females show minimal to no significant differences in the wound closure rate (PMID: 23527289, PMID: 23240590, PMID: 17244326, and PMID: 36536184), we therefore believe that the gender had no influence in the differences observed in the closure rate in the one of our functional experiments that include female mice.

To address the referee's comment, we now also amended the Materials and Methods section to clearly indicate the biometric criteria for group selection that decreased the variability:

[lines: 769-774] "Male and female mice, equally distributed between time groups, were used for descriptive experiments. For functional studies, littermates or age-matched animals were randomly assigned into the different experimental groups. To decrease variability, only males were used for the RAR γ agonist and Hif1 α inhibition treatments (7 dpi), and for the Hif1 α genetic deletion experiment. Females and males, equally distributed between treatment groups, were used for the PDPN+ cell ablation studies and for the 3 dpi Hif1 α inhibition experiment."

The data with human fascia are important, but they were obviously only obtained with fascia from one donor. The authors should at least mention the source of the fascia in Material and Methods (which part of the skin?) and provide the information about the ethical approval.

We thank the reviewer for his comment. Although the experiment shown is from a single sample, repeated control experiments in multiple samples from independent patients show reproducibility of the *ex vivo* contraction model.

For the referee's eyes, we provide a table showing the number of human samples we have obtained over the last 2 years, detailing the times we have repeated the *ex vivo* culture system. In every instance we observe the same behaviour (tissue contraction) as seen across mouse fascia in same culture conditions. Marked in bold is the sample presented in the manuscript.

Sample	Sample	Location	Gender	Age	Date of surgery	Experiment / Storage
AAA11	skin with fat removed: from transition zone of necrotic skin	Leg	M	49	12/3/2021	Histology
AAA12	skin with fat removed: from infected wound and normal part	Abdomen			1/4/2022	Histology and free-floating fascia culture
AAA13	chronic wounds	Hip	M	53	1/18/2022	Histology
AAA14	Abdominal wall		F	40	1/26/2022	Free-floating fascia culture
-	skin, large piece				6/22/2022	Free-floating fascia culture
-	skin & wound				6/29/2022	Histology and free-floating fascia culture
-	skin	Thigh		55	8/10/2022	Free-floating fascia culture
-	skin	Back			9/28/2022	Free-floating fascia culture
-	skin	Abdomen			10/12/2022	Free-floating fascia culture
-	skin	Abdomen		51	12/7/2022	Free-floating fascia culture
-	skin	Abdomen		28	12/8/2022	Free-floating fascia culture: Control, Echinomycin, and RA.
-	skin			32	12/9/2022	Free-floating fascia culture
CAA28	skin				1/27/2023	Free-floating fascia culture
-	skin	Abdomen	F	44	3/31/2023	Free-floating fascia culture
AAB12	skin	Breast		28	4/3/2023	Free-floating fascia culture

Attending the referee's point, we have now included the details for the human fascia experiment in the materials and methods section.

[lines: 797-799] *"Human abdominal skin was obtained from the Plastic Surgery department from the Technical University Munich (ethical approval number 496/21-S-KH), patient details remain anonymous due to privacy concerns."*

The inclusion of data from psoriasis is confusing. Psoriasis is not a fibrotic skin disease - scleroderma data would be more relevant.

The selection of a psoriatic skin dataset was intended to include an inflammatory condition, showing proinflammatory fibroblast states originate from fascia. Attending the reviewer's concern, we have now included in our comparative analysis relevant datasets from human hypertrophic scars and scleroderma patients.

The results section [lines: 583-657] has been amended to include the description of these new datasets.

Data has been included in the new figure 5a, 5c-e, and extended data figure 10:

[NEW] Extended data Fig. 10 | Comparative analysis of human and mouse fibroblasts subtypes. a-b. UMAP representation of scRNAseq datasets of denoted human skin pathologies color-coded for sample of origin (a) and the pan-fibroblast marker PDGFRA expression levels (b). **c.** Composition charts depicting the proportion of the different fibroblast subtypes for each dataset. **d.** Expression comparison between merged human (red) and mouse datasets (blue) of top marker genes for the human fibroblast clusters. Expression of analogous fibroblast clusters is denoted in red rectangles. **e.** UMAP representations the expression levels of highly conserved markers in the merged human (top) and mouse (bottom) datasets.

[NEW] Fig. 5 | Alterations of the fascia-to-myfibroblast trajectory in human skin pathologies. **a.** “Transfer learning” for the mapping of human fibroblasts using the mouse atlas as a reference. UMAP representation of mouse (left), human fibroblasts (right), and river plot (middle) denoting the clusters’ similarities. **b.** Photograph (top left) and low magnification of trichrome-stained section (bottom left) of a human keloid lesion. High magnification micrographs of trichrome-stained, or immunolabeled for the fascia marker PI16,

or for the myofibroblast markers α SMA and RUNX2 in indicated regions of healthy and affected tissue (right). **c.** PAGA connectivities embedded onto UMAP of human fibroblast supercluster (left) and diffusion pseudotime arrangement (right). **d.** UMAPs of subclusters involved in the fascia to myofibroblast trajectories in mouse wound healing and denoted human skin pathologies (top) and their pseudotime ordering (bottom), showing that fascia fibroblast require transit to proinflammatory fibroblasts before becoming myofibroblasts. **e.** RA, hypoxia, and TGF β signalling pathways activity across the inferred trajectories in all datasets showing the conserved connections of RA to the proinflammatory and hypoxia to the myofibroblast state. **f.** Hif1 α inhibitor and RA treatments in human fascia explants show same effects as in mouse explants. Representative brightfield photographs of control, RA-, or Hif1 α inhibitor-treated fascia explants at indicated timepoints after culture (left), area vs time measurements (top right), and total contraction (bottom right). N=4 technical replicates for each condition. Dotted line delimits the keloid lesion. All p-values (p) indicated were obtained from two-tailed T-tests. Scale bars: 2 mm in explants photographs (e), 500 microns in low magnification (b), and 50 microns in high magnification micrographs (b).

Minor points:

It would have been helpful if the authors had marked their changes in the revised manuscript.

We now highlighted the changes made in the most recent manuscript.

The distribution of α -SMA in Fig. 1k is strange – α -SMA usually clusters at the wound edge. In general, the α SMA staining is not well visible.

We changed the image to show the α SMA staining more clearly.

Please refer to Fig. 5e at the end of the first paragraph of page 22.

We made the corresponding changes.

Line 682: RA signalling is not “degraded” – only molecules are degraded. Please rephrase.

Textual changes were done:

[lines: 709-710] *“The second and third spatial directives occurs in more upper wound regions where RA is degraded and is at its lowest.”*

Materials and Methods: Please replace Hydrotamoxifen by Hydroxytamoxifen

Corrected

Referee #3 (Remarks to the Author):

In this resubmission by Correa-Gallegos et al. the authors have made progress in addressing one concern, while not fully addressing another.

1) The first concern regarding performing skin explant experiments using human tissue in skin explant assays was performed and has addressed my concerns.

We thank the referee #3 for his/her comments.

2) Unfortunately, the request that the validation of the labeling specificity of the CD201CreErt2 and PpdnCreErt2 mouse lines (also noted by other reviewers) to be fully characterized in regard to the fascia in homeostatic conditions (before wounding) has not been performed. The experiments that were performed by the authors in Extended Fig 5 are from wounded tissue. And it is impossible to interpret the origin of the cells without characterizing labeling of CD201 and Pdpn mice from

homeostatic conditions. Without this foundational knowledge it becomes difficult to understand the downstream experiments in this manuscript appropriately.

The characterization of the $CD201^{CreER}R26^{Ai14}$ system to label fascia fibroblasts using FACS in Extended Figure 5 was performed with uninjured skin (fascia and dermis samples were from uninjured animals) as well as providing histological data of homeostatic conditions in Figure 1K for the $Pdpn^{CreER}R26^{mTmG}$ system. We realized that the scheme in Figure 5b and the figure legend doesn't explicitly indicate that uninjured homeostatic skin conditions was used for dermis and fascia samples, and we have therefore now amended the text to avoid confusion as shown below.

[NEW] Extended data Fig. 5 | CD201^{CreER}R26^{Ai14} genetic lineage tracing system to fate map fascia fibroblasts. ... b. Flow cytometry strategy to identify proportions of different cell types labelled with TdTomato in the uninjured dermis and fascia, as well from wound and bone marrow from 7 dpi CD201^{CreER}R26^{Ai14} mice.

Major points brought up by referee #2 [Round 1]

Referee #2 (Remarks to the Author):

1. The analyses presented in Extended Data Figure 1h and 1i are central to the conclusions reached by the authors. These data, along with complete UMAP representations of all fibroblasts sequenced from the tissue (Extended Data Figure 1c) need to be presented in the main figures and the analysis needs to be explained more thoroughly. How confident are the authors that these data prove that Sfrp2+ fibroblasts do not differentiate into proto- or terminal-myofibroblasts? The authors should present more than one trajectory analysis method that supports their claims and should consider generating experimental evidence that demonstrates that myofibroblasts in their model do not differentiate from papillary or reticular progenitors or through an Sfrp2+ intermediate state.

Following the referee's recommendations, we shifted the related image panels (Ex. Data Figure 1c & h-i) into new Figure 1a-b (see below) and included a better description of our rationale to focus our efforts in studying the fascia-to-myofibroblast trajectory under the following reasons:

1) The PAGA analysis suggests that our trajectory (Fascia->Proinflammatory->ProtoMyofb->Myofibroblast) is the most likely source for myofibroblasts, and it was substantially corroborated with two independent lineage-specific genetic tracing approaches. Particularly, the CD201^{CreER}R26^{Ai14} system shows a contribution of ca. 80% of the total myofibroblasts in advanced wounds. Assuming an unlikely 100% efficiency of the transgenic system to label fascia fibroblasts, this means that at most, in the context of these type of injuries, other sources (like reticular fibroblasts, see point 2) only give rise to 20% of the myofibroblast pool.

2) Although our analysis also indicate that reticular fibroblasts could directly give rise to myofibroblasts (which has been already proven in PMID:24336287), our previous studies (PMID:31776510) indicate that, in full-thickness skin wounds, fascia fibroblasts represent the most prominent source for myofibroblasts compared to dermal fibroblasts. This study is, thus, primarily focused on this neglected population instead of the vastly well studied reticular fibroblast.

3) Regarding the Sfrp2+ cluster, please refer to the point below.

Expanded rationale:

[Lines: 179-85] “Connectivity values also revealed that the most likely fate of reticular fibroblasts is to the SFRP2+ fibroblast population (Figure 1b). Backtracking the myofibroblast cluster source indicated a preferred origin in the proto-myofibroblast cluster, which in turn likely sprouts out of the proinflammatory fibroblast. Strikingly, the proinflammatory fibroblast cluster derived exclusively from the fascia cluster, indicating that proinflammatory, proto-, and myofibroblasts follow a sequential lineage trajectory that emerges from the fascia fibroblasts (Figure 1b).

[Lines: 200-3] “Based on our trajectory analysis and previous evidence of the contribution of fascia fibroblast into the myofibroblast pool²⁸, we focused our efforts on the sequential trajectory sprouting from the fascia cluster, passing through the proinflammatory, proto-, and ending in the myofibroblast state (Figure 1c).”

Current Fig. 1 a. UMAP representation of all fibroblast clusters embedded with partition-based graph abstraction (PAGA) velocity connectivities. **b.** PAGA connectivity values (left) arranged in potential sources (rows) and fates (column) colour-coded for most (green), mid (yellow), and less probable (orange). Schematics of the potential trajectories highlighting the fascia-to-myofibroblast trajectory (right). **c.** UMAP representations of fibroblast from selected trajectory, colour-coded for individual clusters (left), velocity pseudotime score (middle), and expression of defined markers (right).

[Note to the editor] This issue was also brought up by referee #1 who mentioned this issue has been resolved in the revision.

2. In Figure 1a, the authors present a UMAP representation of a subset of fibroblasts from their data without adequate explanation. It is unclear why, for example, Sfrp2+ fibroblasts have been removed from this plot. These cells are clearly transcriptionally similar to the myofibroblasts and proto-myofibroblasts. How does this UMAP plot and pseudotime analysis compare if the Sfrp2+ fibroblasts are included in this analysis?

Our trajectory inference (Figure 1a-b) suggests that *Sfrp2*⁺ cluster does not transit into any further cell state and its most likely sourced from reticular fibroblasts (0.25 compared to other potential sources connectivity values <0.066). Therefore, this state is not present in the trajectory sprouting from the fascia fibroblasts in their way to myofibroblasts and further characterization of this state remains out of the scope of this work.

An expanded explanation:

[Lines: 179-80] *“Connectivity values also revealed that the most likely fate of reticular fibroblasts is to the SFRP2+ fibroblast population (Figure 1b [see above]).”*

[Note to the editor] This issue was also brought up by referee #1 who mentioned this issue has been resolved in the revision.

3. The authors clearly identify the *Cd201.CreERT2* mouse model as effective in marking fascia fibroblasts in the skin. The authors should better characterize and describe the anatomical location of these cells in the fascia. The transcriptional profile described for these cells is reminiscent of that of a perivascular adventitial fibroblast and the fascia is highly vascularized; are these cells located near or associated with blood vessels?

We have now extensively expanded our characterization of the *CD201CreER* system (Extended data figure 5b-d) and, particularly, we provide evidence showing that labelled fascia fibroblasts are present along the entire fascia connective tissue and are not associated with blood vessels and therefore remain separate from perivascular adventitial fibroblasts (Figure 1h)

[Lines: 295-7] *“We also observed that *TdTomato*⁺ fibroblasts in the fascia were spread along the fascia connective tissue and were not restricted to the perivascular space (Figure 1h).”*

Current Fig. 1 h. Representative whole-mount immunostaining for the endothelial marker PECAM1 in *Procr^{CreER}R26^{Ai1}* fascia. Arrows indicate labelled cells dispersed away the adventitial space.

[Note to the editor] A better characterization of the fascia fibroblast reporter mouse line has been requested by the 3 referees [Point 5 of referee #1 in the previous round and points 2 in both rounds for referee #3]. This issue now seems to be resolved for referee #1. Anatomical description in relation to the adventitial space seem to be a very specific request of referee #2, but we believe that the provided evidence showing labelled fibroblasts unconstrained to the adventitial space is very convincing (Figure 1h).

4. To adequately support the claims that RA and HIF-1a signaling are required in a cell-intrinsic manner for the progressive differentiation of inflammatory fibroblasts and proto myofibroblasts, the authors must generate conditional knockout mouse models. The in vitro experiments presented do not adequately model disease physiology and the readouts of tissue contraction and single gene expression are not sufficient to define cell state transitions. The in vivo evidence presented is very encouraging, however small molecule agonism of RA signaling and inhibition of HIF-1a will likely influence multiple cell compartments so any effects of treatment cannot be attributed directly to pathway activity in the fibroblasts. scRNAseq will likely be a necessary readout for these conditional KO experiments.

We have included new experiments that strongly supports our hypothesis regarding the intrinsic activity of RA and HIF1A on fascia fibroblast differentiation. Particularly with the following experiments:

- 4) RA treatments on purified CD201⁺ fascia fibroblasts cause direct gene expression changes towards proinflammatory states, such as CCL2 and PDPN expression upregulation as well as reduction of myofibroblast markers such as aSMA (Extended data figure 7e-g).

Current Extended data Fig. 7 e. Strategy for fascia fibroblast purification and culture (left). Expression changes (right) of proinflammatory (Ccl2 and Cxcl1) and myofibroblast markers (Acta2) in IL1β- (inflammation-inducing) or TGFβ1-containing media (myofibroblast-inducing). N=3 technical replicates. Expression changes normalized to control medium. p values on bars from two-tailed T-tests (top) and 1-way ANOVA comparisons between treatments (bottom). **f-g.** Expression changes of indicated markers in inflammation- (f) and myofibroblast-inducing media (g) exposed to exogenous RA at indicated concentrations. N=3 technical replicates. Expression changes normalized to control IL1β- (f) or TGFβ1-containing medium (g). p values on bars from two-tailed T-tests (top) and 1-way ANOVA comparisons between treatments (bottom).

[Note to editor] Due to the redundancy of activity among retinoic acid receptors and synthesizing enzymes, a knockout model for retinoic acid signalling would require multiple Knockouts in the same animal and mouse lines that are not commercially obtainable at the moment. Therefore, we opted for an in vitro approach to validate the direct effect of RA on fascia fibroblasts.

5) We now show *in silico* activity and *in vivo* expression patterns of Hif1a in CD201- and in PDPN-derived fibroblasts as well as in whole fascia explants (Figure 4a-b).

Current Fig. 4 a. HIF1α activity vs expression correlations in the different fibroblast along the trajectory from the scRNAseq data. Pearson's R coefficient. **b.** Representative high magnification images of

CD201^{CreER}R26^{Ai14}, PDPN^{CreER}R26^{mTmG} wounds, and fascia explant at indicated time points after injury or culture showing the expression of HIF1α in traced and cultured cells.

6) We have now also included an induced Hif1a gene knockout using our complementary genetic systems (CD201^{CreER} and Pdpn^{CreER}), which replicate our chemical treatments with the HIF1a inhibitor (Figure 4d-e).

Current Fig. 4 d. Representative photographs of control, fascia, or proinflammatory null wounds at 9 dpi (left). Wound area quantifications over time (middle), contraction at 7 dpi (right top) and at 9 dpi (right bottom). N=6 wounds from 3 biological replicates for each genotype. **e.** Masson's trichrome staining of 9 dpi wounds from indicated genotypes (top). Wound maturation-related measurements (bottom). N=12 images from 3 biological replicates per each genotype.

[Note to editor] experiments to validate the roles of RA and HIF1A on fascia fibroblasts, such as conditional knockouts, have been requests from referee #1 [point 9 and 16 from first round] and referee #2. Referee #1 has recognized this issue has been resolved in the current manuscript.

5. In the human datasets presented, it is unclear why authors subset fibroblast clusters in extended data fig 9g and 9h. This would seem to bias their downstream analyses. Do other fibroblast populations give rise to myofibroblasts in the human skin based on bioinformatic analyses of the unsubsetted data set?

We have provided an unbiased method for detection of all fibroblasts in the human datasets using a machine learning algorithm and PAGA trajectory inference (Figure 5a and c).

[NEW] Fig. 5 | Alterations of the fascia-to-myofibroblast trajectory in human skin pathologies. a. “Transfer learning” for the mapping of human fibroblasts using the mouse atlas as a reference. UMAP representation of mouse (left), human fibroblasts (right), and river plot (middle) denoting the clusters’ similarities. ... **c.** PAGA connectivities embedded onto UMAP of human fibroblast supercluster (left) and diffusion pseudotime arrangement (right).

[Note to editor] Similar issue was also pointed out by referee #1 [point 17 of the first round], which has been resolved in referee #1’s opinion for the current manuscript.

Reviewer Reports on the Second Revision:

Referees' comments:

Referee #1 (Remarks to the Author):

The authors have performed additional computational analyses to address some of my questions and they have rephrased some statements. These modifications further improve the quality of the manuscript. This reviewer is still concerned about the very low sample number for wound healing experiments (N=3 in most cases), which is problematic because of the high variability of such experiments. However, at least the issue with the sex of the experiments has been addressed.

Referee #2 (Remarks to the Author):

The authors have adequately addressed most of the comments though it remains unclear why in the analysis of single cell RNA sequencing from mouse and human the authors insist on subsetting the sequenced cells before running trajectory/pseudotime analysis (Figures 1 and 5). This approach is vulnerable to overcorrection/data-fitting.

Referee #3 (Remarks to the Author):

The authors provide an explanation for characterizing the CD201CreER mouse line in unwounded conditions. While the characterization could have been much better it is sufficient for the current manuscript.

This reviewer has no further concerns.

Author Rebuttals to Second Revision:

Referees' comments:

Referee #1 (Remarks to the Author):

The authors have performed additional computational analyses to address some of my questions and they have rephrased some statements. These modifications further improve the quality of the manuscript. This reviewer is still concerned about the very low sample number for wound healing experiments (N=3 in most cases), which is problematic because of the high variability of such experiments. However, at least the issue with the sex of the experiments has been addressed.

We appreciate the contribution of Referee #1 for the improvement of our manuscript. To address the concern of low sample number, we have included a "limitations in our study" statement in the discussion section (highlighted in the current manuscript) as follows:

[Lines: 462-467] *"An important caveat to consider in this present study, is the limited number of biological replicates in our in vivo experiments. Nonetheless, multiple evidence fronts, including cross-species in silico analyses, novel ex vivo systems, and the use of several complementary chemical and genetic mouse models, point towards the same conclusion that the conserved differentiation process of fascia fibroblasts into myofibroblasts orchestrates the progression of the wound healing phases during skin repair."*

Referee #2 (Remarks to the Author):

The authors have adequately addressed most of the comments though it remains unclear why in the analysis of single cell RNA sequencing from mouse and human the authors insist on subsetting the sequenced cells before running trajectory/pseudotime analysis (Figures 1 and 5). This approach is vulnerable to overcorrection/data-fitting.

We appreciate the comment from referee #2. We believe this concern might arise from an inadequate description in the result section when describing the trajectories analyses.

We have indeed parted from an initial unbiased (all fibroblast clusters included) trajectory inference analyses in both mouse (Figure 1a-b) and human datasets (Figure 5c).

Fig. 1 | CD201⁺ fascia fibroblasts differentiate into specialized states during wound healing. a. UMAP plots of all fibroblast clusters embedded with partition-based graph abstraction (PAGA) connectivities for trajectory inference. **b.** PAGA connectivity values (left) arrange in potential sources (rows) and fates (column) colour-coded for most (green), mid (yellow), and less probable (orange). Schematics of the potential trajectories highlighting the fascia-to-myofibroblast trajectory (right).

Fig. 5 | Fascia-to-myofibroblast trajectory in human skin pathologies. c. PAGA connectivities embedded onto UMAP of human fibroblast supercluster (left) and diffusion pseudotime arrangement (right).

We then recalculate the trajectories when sub-setting the fascia-to-myofibroblast trajectory agents (Figure 1c, and Figure 5d-e) for downstream analyses. The reason behind this, as mentioned in the result section, is to look in detail the molecular programs that dictate the activation and differentiation of fascia fibroblast into myofibroblasts, as we have identified this population to be pivotal for skin wound healing.

Fig. 1 | CD201⁺ fascia fibroblasts differentiate into specialized states during wound healing. c. UMAPs of fibroblast from selected trajectory, colour-coded for individual clusters (left), velocity pseudotime score (middle), and expression of defined markers (right).

Fig. 5 | Fascia-to-myofibroblast trajectory in human skin pathologies. d. UMAPs of subclusters involved in the fascia-to-myofibroblast trajectories in mouse wound healing and indicated human skin pathologies (top) and their pseudotime ordering (bottom), showing that fascia fibroblast require transit to proinflammatory fibroblasts before becoming myofibroblasts.

Other trajectories (e.g. reticular fibroblasts into SFRP2⁺ or myofibroblasts) likely follow a different molecular program that, even we consider extremely relevant, are outside the scope of this study; thus they were omitted for the downstream analyses and experiments.

To make clearer that trajectories were calculated twice (unbiased and sub-setting), we made textual changes (highlighted in the current manuscript) in both the results and methods sections as follows:

[Lines: 108-109] *"This initial unbiased analysis revealed a complex interconnection between all fibroblasts..."* **(When describing unbiased trajectories in mouse dataset)**

[Lines: 127-130] *"Based on our initial trajectory analysis... we recalculated the sequential trajectory sprouting from the fascia cluster, passing through the proinflammatory, proto-, and ending in the myofibroblast state (Figure 1c) for further study."* **(When describing the subset reanalysis in mouse)**

[Lines: 422-425] *"In this initial trajectory analysis... indicating variable contributions from the homeostatic populations into the wound fibroblast pools."* **(Human unbiased trajectories)**

[Lines: 426-427] *"We then assessed the conservation of the fascia fibroblast differentiation trajectory into myofibroblasts in each disease."* **(Human subset trajectories)**

[Lines: 875-877 & 880-881] *"Mouse trajectories from all fibroblasts and from fascia-to-myofibroblast trajectory clusters were inferred by... Human trajectories (unbiased and from fascia-to-myofibroblast trajectory clusters) were inferred..."* **(Methods section)**

Referee #3 (Remarks to the Author):

The authors provide an explanation for characterizing the CD201CreER mouse line in unwounded conditions. While the characterization could have been much better it is sufficient for the current manuscript.

This reviewer has no further concerns.

We appreciate the invaluable comments from referee #3 that helped improve substantially our manuscript along the revision process.